# When Do Transformers Outperform Feedforward and Recurrent Networks? A Statistical Perspective

**Alireza Mousavi-Hosseini**[1], **Clayton Sanford**[2], **Denny Wu**[3], **Murat A. Erdogdu**[1]

[1]University of Toronto and Vector Institute,  [2]Google Research,
[3]New York University and Flatiron Institute

{mousavi,erdogdu}@cs.toronto.edu, chsanford@google.com, dennywu@nyu.edu

## Abstract

Theoretical efforts to prove advantages of Transformers in comparison with classical architectures such as feedforward and recurrent neural networks have mostly focused on representational power. In this work, we take an alternative perspective and prove that even with infinite compute, feedforward and recurrent networks may suffer from larger sample complexity compared to Transformers, as the latter can adapt to a form of *dynamic sparsity*. Specifically, we consider a sequence-to-sequence data generating model on sequences of length $N$, where the output at each position only depends on $q \ll N$ relevant tokens, and the positions of these tokens are described in the input prompt. We prove that a single-layer Transformer can learn this model if and only if its number of attention heads is at least $q$, in which case it achieves a sample complexity almost independent of $N$, while recurrent networks require $N^{\Omega(1)}$ samples on the same problem. If we simplify this model, recurrent networks may achieve a complexity almost independent of $N$, while feedforward networks still require $N$ samples. Our proposed sparse retrieval model illustrates a natural hierarchy in sample complexity across these architectures.

## 1 Introduction

The Transformer [VSP$^+$17], a neural network architecture that combines attention and feedforward blocks, forms the backbone of large language models and machine learning approaches across many domains [RNSS18, DBK$^+$20, BMR$^+$20]. The theoretical efforts surrounding the success of Transformers have so far demonstrated various capabilities like in-context learning [ASA$^+$23, VONR$^+$23, BCW$^+$23, ZFB24, KNS24, and others] and chain-of-thought prompting along with its benefits [FZG$^+$23, MS24, LLZM24, KS24, and others] in various settings. There are fewer works that provide specific benefits of Transformers in comparison with feedforward and recurrent architectures. On the approximation side, there are tasks that Transformers can solve with size logarithmic in the input, while alternative architectures require polynomial size [SHT23, SHT24]. Based on these results, [WWHL24] showed a separation between Transformers and feedforward networks by providing further optimization guarantees for gradient-based training of Transformers on a sparse token selection task.

While most prior works focused on the approximation separation between Transformers and feedforward networks (FFNs), in this work we focus on a purely statistical separation, and ask:

> *What function class can Transformers learn with fewer samples compared to feedforward and recurrent networks, even with infinite computational resources?*

[FGBM23] approached the above problem with random features, where the query-key matrix for the attention and the first layer weights for the two-layer feedforward network were fixed at random

39th Conference on Neural Information Processing Systems (NeurIPS 2025).

| Statistical Model | Feedforward | RNN | Transformer |
|---|---|---|---|
| Simple-$q$STR | ✗ (Theorem 9) | ✓ (Theorem 5) | ✓ (Theorem 3) |
| $q$STR | ✗ (Theorem 9) | ✗ (Theorem 7) | ✓ (Theorem 3) |

Table 1: Summary of main contributions (see Theorem 1). ✓ indicates a sample complexity upper bound that is almost sequence length-free (up to polylogarithmic factors). ✗ indicates a lower bound of order $N^{\Omega(1)}$.

initialization. However, this only presents a partial picture, as neural networks can learn a significantly larger class of functions once "feature learning" is allowed, i.e., parameters are trained to adapt to the structure of the underlying task [Bac17, BES$^+$22, DLS22, BBSS22, DKL$^+$23, AAM23, MHWE24].

We evaluate the statistical efficiency of Transformers and alternative architectures by characterizing how the sample complexity depends on the input sequence length. A benign length dependence (e.g., sublinear) signifies the ability to achieve low test error in longer sequences, which intuitively connects to the *length generalization* capability [AWA$^+$22]. While Transformers have demonstrated this ability in certain structured logical tasks, they fail in other simple settings [ZBL$^+$23, LAG$^+$23]. Our generalization bounds for bounded-norm Transformers — along with our contrasts to RNNs and feedforward neural networks — provide theoretical insights into the statistical advantages of Transformers and lay the foundation for future rigorous investigations of length generalization.

## 1.1 Our Contributions

We study the $q$-Sparse Token Regression ($q$STR) data generating model, a sequence-to-sequence model where the output at every position depends on a sparse subset of the input tokens. Importantly, this dependence is dynamic, i.e., changes from prompt to prompt, and is described in the input itself. We prove that by employing the attention layer to retrieve relevant tokens at each position, single-layer Transformers can adapt to this dynamic sparsity, and learn $q$STR with a sample complexity almost independent of the length of input sequence $N$, as long as the number of attention heads is at least $q$. On the other hand, we develop a new metric-entropy-based argument to derive norm and parameter-count lower bounds for RNNs approximating the $q$STR model. Thanks to lower bounds on weight norm, we also obtain a sample complexity lower bound of order $N^{\Omega(1)}$ for RNNs. Further, we show that RNNs can learn a subset of $q$STR where the output is a constant sequence, which we call simple-$q$STR, with a sample complexity polylogarithmic in $N$. Finally, we develop a lower bound technique for feedforward networks (FFNs) that takes advantage of the fully connected projection of the first layer to obtain a sample complexity lower bound linear in $N$, even when learning simple-$q$STR models. The following theorem and Table 1 summarize our main contributions.

**Theorem 1** (Informal). *We have the following hierarchy of statistical efficiency for learning $q$STR.*

- *A single-layer Transformer with $H \geq q$ heads can learn $q$STR with sample complexity almost independent of $N$, and cannot learn $q$STR when $H < q$ even with infinitely many samples.*

- *RNNs can learn* simple-$q$STR *with sample size almost independent of $N$, but require at least $\Omega(N^c)$ samples for some constant $c > 0$ to learn a generic $q$STR model, regardless of their size.*

- *Feedforward neural networks, regardless of their size, require $\Omega(Nd)$ samples to learn even* simple-$q$STR *models, where $d$ is input token dimension.*

We empirically validate the intuitions from Theorem 1 in Figure 1. Observe that on a 1STR task, both FFNs and RNNs suffer from a large sample complexity for larger $N$. However, for a simple-1STR model, RNNs perform closer to Transformers with a much milder dependence on $N$ than FFNs.

## 1.2 Related Work

While generalization is a fundamental area of study in machine learning theory, theoretical work on the generalization capabilities of Transformers remains relatively sparse. Some works analyze the inductive biases of self-attention through connections to max-margin SVM classifiers [VDT24]. Others quantify complexity in terms of the simplest programs in a formal language (such as the RASP model of [YCA23]) that solve the task and relate that to Transformer generalization [ZBL$^+$23, CS24]. The most relevant works to our own are [EGKZ22, TT23, Tru24], which employ covering numbers to bound the sample complexity of deep Transformers with bounded weights. They demonstrate a logarithmic scaling in the sequence length, depth, and width and apply their bounds

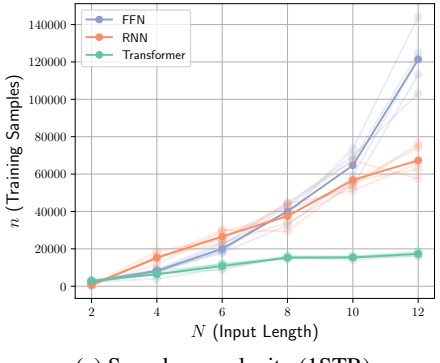

(a) Sample complexity (1STR)

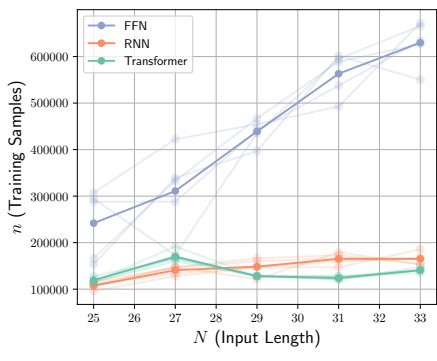

(b) Sample complexity (Simple-1STR)

Figure 1: Number of samples required to reach a certain test MSE loss threshold while training with online AdamW. We consider (a) the 1STR model with loss threshold 0.7 and (b) the simple-1STR model with loss threshold 0.02, averaged over 5 experiments. We use a linear link function, standard Gaussian input, $d = 10$ and $d_e = \lfloor 5 \log(N) \rfloor$. Positional encodings are sampled uniformly from the unit hypercube. Experimental details and additional results on the effect of $q$ are provided in Appendix E.

to the learnability of sparse Boolean functions. We refine these covering number bounds to better characterize generalization in sequence-to-sequence learning with dynamic sparsity [SHT23]. Our problems formalize long-context reasoning tasks, extending beyond simple retrieval to include challenges like *multi-round coreference resolution* [VOT⁺24].

**Expressivity of Transformers.** The expressive power of Transformers has been extensively studied in prior works. Universality results establish that Transformers can approximate the output of any continuous function or Turing machine [YBR⁺19, WCM21], as well as measure-to-measure maps [GRRB24], and their memorization capacity is well-understood [MLT24]. However, complexity limitations remain for bounded-size models. Transformers with fixed model sizes are unable to solve even regular languages, such as Dyck and Parity [BAG20, Hah20]. Further work [e.g. MS23] relates Transformers to boolean circuits to establish the hardness of solving tasks like graph connectivity with even polynomial-width Transformers. Additionally, work on self-attention complexity explores how the embedding dimension and number of heads affects the ability of attention layers to approximate sparse matrices [LCW21], recover nearest-neighbor associations [AYB24], and compute sparse averages [SHT23]. The final task closely resembles our $q$STR model and has been applied to relate the capabilities of deep Transformers to parallel algorithms [SHT24]. Several works [e.g. JBKM24, BHBK24, WDL24] introduce sequential tasks where Transformers outperform RNNs or other state space models in parameter-efficient expressivity. We establish similar architectural separations with an added focus on generalization capabilities.

**Statistical Separation.** Our work is conceptually related to studies on feature learning and adaptivity in feedforward networks, particularly in learning models with sparsity and low-dimensional structures. Prior work has analyzed how neural networks and gradient-based optimization introduce inductive biases that facilitates the learning of low-rank and low-dimensional functions [LMZ18, WLLM19, CB20, MHPG⁺23, OSSW24]. These studies often demonstrate favorable generalization properties based on certain structures of the solution such as large margin or low norm [BFT17, NLB⁺18, OWSS19, WLLM19]. Our goal is to extend efficient learning of low-dimensional concepts to sequential architectures, ensuring sample complexity remains efficient in both input dimension $d$ and context length $N$. Our approach, motivated by [SHT23, WWHL24], suggests that $q$STR is a sequential model whose sparsity serves as a low-dimensional structure, making it the primary determinant of generalization complexity for Transformers.

**Notation.** For a natural number $n$, define $[n] := \{1, \ldots, n\}$. We use $\|\cdot\|_p$ to denote the $\ell_p$ norm of vectors. For a matrix $\boldsymbol{A} \in \mathbb{R}^{m \times n}$, $\|\boldsymbol{A}\|_{p,q} := \left\| \left( \|\boldsymbol{A}_{:,1}\|_p, \ldots, \|\boldsymbol{A}_{:,n}\|_p \right) \right\|_q$, and $\|\boldsymbol{A}\|_{\text{op}}$ denotes the operator norm of $\boldsymbol{A}$. We use $a \lesssim b$ and $a \leq \mathcal{O}(b)$ interchangeably, which means $a \leq Cb$ for some absolute constant $C$. We similarly define $\gtrsim$ and $\Omega$. $\tilde{\mathcal{O}}$ and $\tilde{\Omega}$ hide multiplicative constants that depend polylogarithmically on problem parameters. $\sigma$ denotes the ReLU activation.

## 2 Problem Setup

**Statistical Model.** In this paper, we will focus on the ability of different architectures for learning the following data generating model.

**Definition 2** (q-Sparse Token Regression). *Suppose $\boldsymbol{p}, \boldsymbol{y} \sim \mathcal{P}$ where*

$$\boldsymbol{p} = \left( \begin{pmatrix} \boldsymbol{x}_1 \\ \boldsymbol{t}_1 \end{pmatrix}, \ldots, \begin{pmatrix} \boldsymbol{x}_N \\ \boldsymbol{t}_N \end{pmatrix} \right),$$

$\boldsymbol{t}_i \in [N]^q$ *and* $\boldsymbol{x}_i \in \mathbb{R}^d$ *for* $i \in [N]$. *In the q-sparse token regression (qSTR) data generating model, the output is given by* $\boldsymbol{y} = (y_1, \ldots, y_N)^\top \in \mathbb{R}^N$, *where*

$$y_i = g(\boldsymbol{x}_{t_{i1}}, \ldots, \boldsymbol{x}_{t_{iq}}),$$

*for some* $g : \mathbb{R}^{qd} \to \mathbb{R}$. *We call this model* simple-qSTR *if the data distribution is such that* $\boldsymbol{t}_i = \boldsymbol{t}$ *for all* $i \in [N]$ *and some* $\boldsymbol{t}$ *drawn from* $[N]^q$.

The above defines a class of sequence-to-sequence functions, where the label at position $i$ in the output sequence depends only on a subsequence of size $q$ of the input data, determined by the set of indices $\boldsymbol{t}_i$. $\boldsymbol{p}$ in the above definition denotes the prompt or context. Given the large context length of modern architectures, we are interested in a setting where $q \ll N$. In this setting, the answer at each position only depends on a few tokens, however the tokens it depends on change based on the context. Therefore, we seek architectures that are *adaptive* to this form of *dynamic sparsity* in the true data generating process, with computational and sample complexity independent of $N$. As a special case, choosing $g$ as the tokens' mean recovers the *sparse averaging* model proposed in [SHT23], where the authors separate the representational capacity of Transformers and other architectures.

While our main motivation for using the $q$STR model is the role of this model as a theoretical benchmark (cf. [SHT23, WWHL24]), we now present an example of how tasks similar to $q$STR can arise in natural language modeling. Consider the prompt "*For my vacation this summer, I'm considering either Paris or Tokyo. If I go to Paris, I want to visit their art museums, and if I end up in Tokyo, I want to try their cuisine. Can you tell me how much would my first and second option cost respectively?*" In this case, $\boldsymbol{t}_1$ is the token *first* and refers to the tokens *Paris* and *art museums*, while $\boldsymbol{t}_2$ is the token *second* and refers to the tokens *Tokyo* and *cuisine*. Note that for either $\boldsymbol{t}_1$ or $\boldsymbol{t}_2$, the answer to the prompt only depends on two tokens out of the entire context, thus this example demonstrates the case of $q = 2$. We refer the interested readers to the multi-round conference resolution task of [VOT+24] for more realistic examples in evaluating large models.

To obtain statistical guarantees, we will impose mild moment assumptions on the data.

**Assumption 1.** *Suppose* $\mathbb{E}[\|\boldsymbol{x}_i\|^r]^{1/r} \leq \sqrt{C_x d r}$ *and* $\mathbb{E}[|y_i|^r]^{1/r} \leq \sqrt{C_y r^s}$ *for all* $r \geq 1$, $i \in [N]$, *and some absolute constants* $s \geq 1$ *and* $C_x, C_y > 0$.

We only require the above assumption to establish standard concentration bounds, and it is satisfied as soon as $\|\boldsymbol{x}\|$ is subGaussian and $y$ is sub-Weibull (e.g. $g$ grows at most like a polynomial of degree $s$). Learning the $q$STR model requires two steps: $(i)$ extracting the relevant tokens at each position, $(ii)$ learning the link function $g$. We are interested in settings where the difficulty of learning is dominated by the first step, hence we assume $g$ can be approximated by a two-layer feedforward network.

**Assumption 2.** *There exist* $m_g \in \mathbb{N}$, $\boldsymbol{a}_g, \boldsymbol{b}_g \in \mathbb{R}^{m_g}$ *and* $\boldsymbol{W}_g \in \mathbb{R}^{m_g \times qd}$, *such that* $\|\boldsymbol{a}_g\|_2 \leq r_a / \sqrt{m_g}$, *and* $\|(\boldsymbol{W}_g, \boldsymbol{b}_g)\|_F \leq \sqrt{m_g} r_w$ *for some constants* $r_a, r_w > 0$, *and*

$$\sup_{\left\{ \|\boldsymbol{x}_i\|_2 \leq \sqrt{Cd \log(nN)}, \, \forall i \in [q] \right\}} \left| g(\boldsymbol{x}_1, \ldots, \boldsymbol{x}_q) - \boldsymbol{a}_g^\top \sigma(\boldsymbol{W}_g(\boldsymbol{x}_1^\top, \ldots, \boldsymbol{x}_q^\top)^\top + \boldsymbol{b}_g) \right|^2 \leq \varepsilon_{\texttt{2NN}},$$

*where* $C = 3 C_x e$ *and* $\varepsilon_{\texttt{2NN}}$ *is some absolute constant.*

Ideally, $\varepsilon_{\texttt{2NN}}$ above is a small constant denoting the approximation error. This assumption can be verified using various universal approximation results for ReLU networks. For example, when $g$ is an additive model of $P$ Lipschitz functions, where each function depends only on a $k$-dimensional projection of the input, the above holds for every $\varepsilon_{\texttt{2NN}} > 0$ and $m_g = \tilde{\mathcal{O}}((P/\sqrt{\varepsilon_{\texttt{2NN}}})^k)$, $r_a = \tilde{\mathcal{O}}((P/\sqrt{\varepsilon_{\texttt{2NN}}})^{(k+1)/2})$, and $r_w = 1$ (we can always have $r_w = 1$ by homogeneity) [Bac17].

**Empirical Risk Minimization.** While Empirical Risk Minimization (ERM) is a standard abstract learning algorithm to use for generalization analysis, its standard formalizations use risk functions for scalar-valued predictions. Before introducing the notions of ERM that we employ, we first state several sequential risk formulations to evaluate a predictor $\hat{\boldsymbol{y}}_{\texttt{arc}}(\cdot; \boldsymbol{\Theta}) \in \mathcal{F}_{\texttt{arc}}$ on i.i.d. training samples $\{\boldsymbol{p}^{(i)}, \boldsymbol{y}^{(i)}\}_{i=1}^n$, where $\texttt{arc}$ denotes a general architecture. We define the *population risk*, *averaged empirical risk*, and *point-wise empirical risk* respectively as

$$R^{\texttt{arc}}(\boldsymbol{\Theta}) \coloneqq \frac{1}{N} \mathbb{E}\left[ \sum_{j=1}^N (\hat{y}_{\texttt{arc}}(\boldsymbol{p}; \boldsymbol{\Theta})_j - y_j)^2 \right] = \frac{1}{N} \mathbb{E}\left[ \|\hat{\boldsymbol{y}}_{\texttt{arc}}(\boldsymbol{p}; \boldsymbol{\Theta}) - \boldsymbol{y}\|_2^2 \right], \qquad (2.1)$$

$$\hat{R}_{n,N}^{\texttt{arc}}(\boldsymbol{\Theta}) \coloneqq \frac{1}{nN} \sum_{i=1}^n \sum_{j=1}^N \left( \hat{y}_{\texttt{arc}}(\boldsymbol{p}^{(i)}; \boldsymbol{\Theta})_j - y_j^{(i)} \right)^2, \qquad (2.2)$$

$$\hat{R}_n^{\texttt{arc}}(\boldsymbol{\Theta}) \coloneqq \frac{1}{n} \sum_{i=1}^n \left( \hat{y}_{\texttt{arc}}(\boldsymbol{p}^{(i)}; \boldsymbol{\Theta})_{j^{(i)}} - y_{j^{(i)}}^{(i)} \right)^2, \qquad (2.3)$$

where $\{j^{(i)}\}_{i=1}^n$ are i.i.d. position indices drawn from $\text{Unif}([N])$. The goal is to minimize the population risk $R^{\texttt{arc}}(\boldsymbol{\Theta})$ by minimizing some empirical risk, potentially with weight regularization. We use three formalizations of learning algorithms to prove our results.

1. *Constrained ERM* minimizes an empirical risk $\hat{R}_n^{\texttt{arc}}$ subject to the model parameters belonging on some (e.g., norm-constrained) set $\Theta$. Concretely, let

$$\hat{\boldsymbol{\Theta}} \in \arg\min_{\boldsymbol{\Theta} \in \Theta} \hat{R}_n^{\texttt{arc}}(\boldsymbol{\Theta}).$$

   Theorem 3 considers constrained ERM algorithms for bounded-weight transformers with point-wise risk $\hat{R}_n^{\texttt{TR}}(\boldsymbol{\Theta})$, and Theorem 5 uses $\hat{R}_n^{\texttt{RNN}}(\boldsymbol{\Theta})$ for RNNs. Note that upper bounds for training with point-wise empirical risk $\hat{R}_n^{\texttt{arc}}$ readily transfer to training with averaged empirical risk $\hat{R}_{n,N}^{\texttt{arc}}$.

2. *Min-norm $\varepsilon$-ERM* minimizes the norm of the parameters, subject to sufficiently small loss:

$$\hat{\boldsymbol{\Theta}}_\varepsilon \in \underset{\{\boldsymbol{\Theta} : \hat{R}_n^{\texttt{arc}}(\boldsymbol{\Theta}) - \min \hat{R}_n^{\texttt{arc}} \leq \varepsilon\}}{\arg\min} \|\text{vec}(\boldsymbol{\Theta})\|_2. \qquad (2.4)$$

   Theorem 7 uses min-norm $\varepsilon$-ERM to place a sample complexity lower bound $\hat{R}_n^{\texttt{RNN}}(\boldsymbol{\Theta})$.

3. Beyond ERM, Theorem 9 also considers *stationary points* of the averaged or point-wise loss, with $\ell_2$ regularization. This learning algorithm is presented in greater detail in Definition 8.

If $\Theta$ is defined by a norm constraint, then min-norm $\varepsilon$-ERM with a proper $\varepsilon$ can be seen as an instance of constrained ERM. All three formulations are motivated by practical optimization algorithms that either minimize an explicitly regularized loss, or have an implicit bias towards min-norm solutions.

# 3 Transformers

A single-layer Transformer is composed of an attention and a parallel feedforward layer. Given a sequence $\{\boldsymbol{z}_i\}_{i=1}^N$ of input embeddings where $\boldsymbol{z}_i \in \mathbb{R}^{D_e}$ with embedding dimension $D_e$, a single head of attention outputs another sequence of length $N$ in $\mathbb{R}^{D_e}$, given by

$$f_{\texttt{Attn}}(\boldsymbol{p}; \boldsymbol{W}_Q, \boldsymbol{W}_K, \boldsymbol{W}_V) = \left[ \sum_{j=1}^N \boldsymbol{W}_V \boldsymbol{z}_j \frac{e^{\langle \boldsymbol{W}_Q \boldsymbol{z}_i, \boldsymbol{W}_K \boldsymbol{z}_j \rangle}}{\sum_{l=1}^N e^{\langle \boldsymbol{W}_Q \boldsymbol{z}_i, \boldsymbol{W}_K \boldsymbol{z}_l \rangle}} \right]_{i \in [N]}.$$

Where $\boldsymbol{W}_K, \boldsymbol{W}_Q, \boldsymbol{W}_V$ are the key, query, and value projection matrices respectively. The output of $H$ units of attention can be concatenated to form multi-head attention with output $\boldsymbol{h} \in \mathbb{R}^{HD_e}$. A two-layer neural network acts on $\boldsymbol{h}$ to generate the final output sequence via

$$f_{\texttt{2NN}}(\boldsymbol{h}; \boldsymbol{a}_{\texttt{2NN}}, \boldsymbol{W}_{\texttt{2NN}}, \boldsymbol{b}_{\texttt{2NN}}) = \boldsymbol{a}_{\texttt{2NN}}^\top \sigma(\boldsymbol{W}_{\texttt{2NN}} \boldsymbol{h} + \boldsymbol{b}_{\texttt{2NN}}), \quad \boldsymbol{W}_{\texttt{2NN}} \in \mathbb{R}^{m \times HD_e}, \boldsymbol{a}_{\texttt{2NN}}, \boldsymbol{b}_{\texttt{2NN}} \in \mathbb{R}^m$$

Our architectural choices are standard in theoretical studies of Transformers. We provide full details, including how to obtain input embeddings by positional encoding, in Appendix A.1.

### 3.1 Learning Guarantees for Multi-Head Transformers

We consider the following parameter class $\Theta_{\mathrm{TR}} = \{\|\mathrm{vec}(\boldsymbol{\Theta})\|_2 \leq R\}$ and provide a learning guarantee for empirical risk minimizers over $\Theta_{\mathrm{TR}}$, with its proof deferred to Appendix A.2.

**Theorem 3.** *Let $\hat{\boldsymbol{\Theta}} = \arg\min_{\boldsymbol{\Theta} \in \Theta_{\mathrm{TR}}} \hat{R}_n^{\mathrm{TR}}(\boldsymbol{\Theta})$ and $m = m_g$. Suppose we set $H = q$ and $R^2 = \tilde{\Theta}(r_a^2/m_g + m_g r_w^2 + q^2/d)$. Under Assumptions 1, 2 and 3, we have*

$$R^{\mathrm{TR}}(\hat{\boldsymbol{\Theta}}_n) \lesssim \varepsilon_{\mathrm{2NN}} + \tilde{\mathcal{O}}\left( C_1 \sqrt{\frac{m_g q(d+q) + q^3 + qd^2}{n}} \right)$$

*where $C_1 = R^2 qd$, with probability at least $1 - n^{-c}$ for some absolute constant $c > 0$.*

We make the following remarks.

- First, the sample complexity above depends on $N$ only up to log factors. Second, we can remove the $C_1$ factor by performing a clipping operation with a large constant on the Transformer output. Note that the first and second terms in the RHS above denote the approximation and estimation errors respectively. Extending the above guarantee to cover $m \geq m_g$ and $H \geq q$ is straightforward.

- This bound provides guidance on the relative merits of scaling the parameter complexity of the feedforward versus the attention layer (which is an active research area related to Transformer scaling laws [HSSL24, JMB$^+$24]), by highlighting the trade-off between the two to achieve minimal generalization error. Concretely, $m_g \gg d + q$ represents a regime where the complexity is dominated by the feedforward layer learning the downstream task $g$, while $m_g \ll d + q$ signifies dominance of the attention layer learning to retrieve the relevant tokens.

Finally, by incorporating additional structure in the ERM solution, it is possible to obtain improved sample complexities. A close study of the optimization dynamics may reveal such additional structure in the solution reached by gradient-based methods, pushing the sample complexity closer to the information-theoretic limit of $\Omega(qd)$. Figure 2 demonstrates that the attention weights achieved through standard optimization of a Transformer match our theoretical constructions – see Equation (A.2) – even while maintaining separate $\boldsymbol{W}_Q$ and $\boldsymbol{W}_K$ during training (we use the 1STR setup of Figure 1 with $N = 100$). We leave the study of optimization dynamics and the resulting sample complexity for future work.

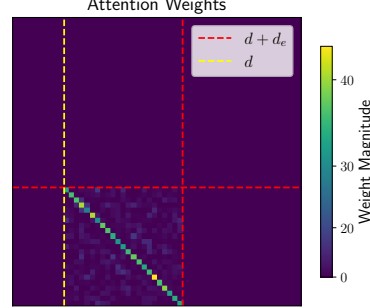

Figure 2: Trained attention weights match our theoretical construction (A.2).

### 3.2 Limitations of Transformers with Few Heads

We establish the necessity of the linear dependence of $H$ on $q$. In contrast to [AYB24], we do not put any assumptions on the rank of the key-query projections, i.e. our lower bound applies even when the key-query projection matrix is full-rank.

**Proposition 4.** *Consider a $q$STR model where $y_i = \frac{1}{\sqrt{qd}} \sum_{j=1}^q \left( \|\boldsymbol{x}_{t_{ij}}\|^2 - \mathbb{E}[\|\boldsymbol{x}_{t_{ij}}\|^2] \right)$, $\boldsymbol{x}_i \sim \mathcal{N}(0, \boldsymbol{\Sigma}_i)$ such that $\boldsymbol{\Sigma}_i = \mathbf{I}_d$ for $i < N/2$ and $\boldsymbol{\Sigma}_i = 0$ for $i \geq N/2$. Then, there exists a distribution over $(\boldsymbol{t}_i)_{i \in [N]}$ such that for any choice of $\boldsymbol{\Theta}_{\mathrm{TR}}$ (including arbitrary $\{\boldsymbol{W}_{\mathrm{QK}}^{(h)}\}_{h \in [H]}$), we have*

$$\frac{1}{N} \mathbb{E}\left[ \|\boldsymbol{y} - \hat{\boldsymbol{y}}_{\mathrm{TR}}(\boldsymbol{p}; \boldsymbol{\Theta}_{\mathrm{TR}})\|_2^2 \right] \geq 1 - \frac{(q+d)H}{qd}.$$

**Remark.** We highlight the importance of the nonlinear dependence of $y_i$ on $\boldsymbol{x}$ for the above lower bound. In particular, for the sparse token averaging task introduced in [SHT23], a single-head attention layer with a carefully constructed embedding suffices for approximation.

The above proposition implies that given sufficiently large dimensionality $d \gg q$, approximation alone necessitates at least $H = \Omega(q)$ heads. In Appendix A.3, we present the proof of Proposition 4, along with Proposition 21 which establishes an exact lower bound $H \geq q$ for all $d \geq 1$, at the expense of additional restrictions on the query-key projection matrix.

# 4 Recurrent Neural Networks

In this section, we first provide positive results for RNNs by proving that they can learn simple-$q$STR with a sample complexity only polylogarithmic in $N$, thus establishing a separation in their learning capability from feedforward networks. Next, we turn to general $q$STR, where we provide a negative result on RNNs, proving that to learn such models their sample complexity must scale with $N^{\Omega(1)}$ regardless of model size, making them less statistically efficient than Transformers. Throughout this section, we focus on bidirectional RNNs, since the $q$STR model is not necessarily causal and the output at position $i$ may depend on future tokens.

## 4.1 RNNs can learn simple-$q$STR

A bidirectional RNN maintains, for each position in the sequence, a forward and a reverse hidden state, denoted by $(\boldsymbol{h}_i^{\rightarrow})_{i=1}^N$ and $(\boldsymbol{h}_i^{\leftarrow})_{i=1}^N$, where $\boldsymbol{h}_i^{\rightarrow}, \boldsymbol{h}_i^{\leftarrow} \in \mathbb{R}^{d_h}$. These hidden states are obtained by initializing $\boldsymbol{h}_1^{\rightarrow} = \boldsymbol{h}_N^{\leftarrow} = \boldsymbol{0}_{d_h}$ and recursively applying

$$\boldsymbol{h}_i^{\rightarrow} = \Pi_{r_h}\big(\boldsymbol{h}_{i-1}^{\rightarrow} + f_h^{\rightarrow}(\boldsymbol{h}_{i-1}^{\rightarrow}, \boldsymbol{z}_{i-1}; \boldsymbol{\Theta}_h^{\rightarrow})\big), \quad \forall i \in \{2, \ldots, N\}$$

$$\boldsymbol{h}_i^{\leftarrow} = \Pi_{r_h}\big(\boldsymbol{h}_{i+1}^{\leftarrow} + f_h^{\leftarrow}(\boldsymbol{h}_{i+1}^{\leftarrow}, \boldsymbol{z}_{i+1}; \boldsymbol{\Theta}_h^{\leftarrow})\big), \quad \forall i \in \{1, \ldots, N-1\},$$

where $\Pi_{r_h} : \mathbb{R}^{d_h} \to \mathbb{R}^{d_h}$ is the projection $\Pi_{r_h}\boldsymbol{h} = (1 \wedge r_h/\|\boldsymbol{h}\|_2)\boldsymbol{h}$, and $f_h^{\rightarrow}$ and $f_h^{\leftarrow}$ are implemented by feedforward networks, parameterized by $\boldsymbol{\Theta}_h^{\rightarrow}$ and $\boldsymbol{\Theta}_h^{\leftarrow}$ respectively. Recall $\boldsymbol{z}_i = (\boldsymbol{x}_i^{\top}, \mathrm{enc}(i, \boldsymbol{t}_i)^{\top})^{\top}$ is the encoding of $\boldsymbol{x}_i$. We remark that while we add $\Pi_{r_h}$ for technical reasons, it resembles layer normalization which ensures stability of the state transitions on very long inputs; a more involved analysis can replace $\Pi_{r_h}$ with standard formulations of layer normalization. Additionally, directly adding $\boldsymbol{h}_{i-1}^{\rightarrow}$ and $\boldsymbol{h}_{i+1}^{\leftarrow}$ to the output of transition functions represents residual or skip connections. The output at position $i$ is generated by

$$y_i = f_y(\boldsymbol{h}_i^{\rightarrow}, \boldsymbol{h}_i^{\leftarrow}, \boldsymbol{z}_i; \boldsymbol{\Theta}_y),$$

which is an $L_y$-layer feedforward network. Specifically, we consider an RNN with deep transitions [PGCB13] and let $f_h^{\rightarrow}(\cdot; \boldsymbol{\Theta}_h^{\rightarrow})$ be an $L_h$-layer feedforward network (see Appendix B.1 for complete definitions). We denote the complete output of the RNN via

$$\hat{\boldsymbol{y}}_{\mathrm{RNN}}(\boldsymbol{p}; \boldsymbol{\Theta}_{\mathrm{RNN}}) = (f_y(\boldsymbol{h}_1^{\rightarrow}, \boldsymbol{h}_1^{\leftarrow}, \boldsymbol{z}_1; \boldsymbol{\Theta}_y), \ldots, f_y(\boldsymbol{h}_N^{\rightarrow}, \boldsymbol{h}_N^{\leftarrow}, \boldsymbol{z}_N; \boldsymbol{\Theta}_y)) \in \mathbb{R}^N.$$

We have the following guarantee for RNNs learning simple-$q$STR models.

**Theorem 5.** *Let $\hat{\boldsymbol{\Theta}} = \arg\min_{\boldsymbol{\Theta} \in \Theta_{\mathrm{RNN}}} \hat{R}_n^{\mathrm{RNN}}(\boldsymbol{\Theta})$ (with $\Theta_{\mathrm{RNN}}$ defined in Equation (B.2)). Suppose Assumptions 1, 2 and 3 hold with the simple-$q$STR model, i.e. $\boldsymbol{t}_i = \boldsymbol{t}$ for all $i \in [N]$ and some $\boldsymbol{t}$ drawn from $[N]^q$. Then, with $L_h, L_y = \mathcal{O}(1)$, $r_h = \tilde{\Theta}(\sqrt{qd})$, and proper hyperparameters in $\Theta_{\mathrm{RNN}}$ (see Appendix B.1), we obtain*

$$R^{\mathrm{RNN}}(\hat{\boldsymbol{\Theta}}) \lesssim \varepsilon_{\mathrm{2NN}} + \sqrt{\frac{\mathrm{poly}(d, q, m_g, r_a, r_w, \varepsilon_{\mathrm{2NN}}^{-1}, \log(nN))}{n}},$$

*with probability at least $1 - n^{-c}$ for some absolute constant $c > 0$.*

As desired, the above sample complexity depends on $N$ only up to polylogarithmic factors. The dimension and norm of RNN weights, implicit in the formulation above, must have a similar polynomial scaling as evident by the proof of the above theorem in Appendix B.

## 4.2 RNNs cannot learn general $q$STR

For our lower bound, we will consider a broad class of recurrent networks, without restricting to a specific form of parametrization. Specifically, we consider bidirectional RNNs chracterized by

$$\boldsymbol{h}_{i+1}^{\rightarrow} = \mathrm{proj}_{r_h}\big(f_h^{\rightarrow}(\boldsymbol{h}_i^{\rightarrow}, \boldsymbol{x}_i, \boldsymbol{t}_i, i)\big), \quad \forall i \in \{1, \ldots, N-1\}$$

$$\boldsymbol{h}_{i-1}^{\leftarrow} = \mathrm{proj}_{r_h}\big(f_h^{\leftarrow}(\boldsymbol{h}_i^{\leftarrow}, \boldsymbol{x}_i, \boldsymbol{t}_i, i)\big), \quad \forall i \in \{2, \ldots, N\}$$

$$y_i = f_y(\boldsymbol{U}^{\rightarrow}\boldsymbol{h}_i^{\rightarrow}, \boldsymbol{U}^{\leftarrow}\boldsymbol{h}_i^{\leftarrow}, \boldsymbol{x}_i, \boldsymbol{t}_i, i), \quad \forall i \in [N]$$

where $f_y : \mathbb{R}^{d_h} \times \mathbb{R}^{d_h} \times \mathbb{R}^d \times [N]^{q+1} \to \mathbb{R}$, $f_h^{\rightarrow}, f_h^{\leftarrow} : \mathbb{R}^{d_h} \times \mathbb{R}^d \times [N]^{q+1} \to \mathbb{R}^{d_h}$, $\boldsymbol{U}^{\rightarrow}, \boldsymbol{U}^{\leftarrow} \in \mathbb{R}^{d_h \times d_h}$, $d_h$ is the width of the model, and $r_h > 0$ is some constant. Moreover, $\mathrm{proj}_{r_h} : \mathbb{R}^{d_h} \to \mathbb{R}^{d_h}$

is any mapping that guarantees $\left\|\mathrm{proj}_{r_h}(\cdot)\right\|_2 \le r_h$. As mentioned before, this operation mirrors the layer normalization to ensure that $\boldsymbol{h}_i$ remains stable. Further, we assume $f_y(\cdot, \boldsymbol{x}, \boldsymbol{t})$ is $\mathfrak{L}/r_h$-Lipschitz for all $\boldsymbol{x} \in \mathbb{R}^d$ and $\boldsymbol{t} \in [N]^q$. This formulation covers different variants of (bidirectional) RNNs used in practice such as LSTM and GRU, and includes the RNN formulation of Section 4.1 as a special case. Define $\boldsymbol{U} := (\boldsymbol{U}^{\rightarrow}, \boldsymbol{U}^{\leftarrow}) \in \mathbb{R}^{d_h \times 2d_h}$ for conciseness. Note that in practice $f_y, f_h^{\rightarrow}, f_h^{\leftarrow}$ are determined by additional parameters. However, the only weight that we explicitly denote in this formulation is $\boldsymbol{U}$, since our lower bound will directly involve this projection, and we keep the rest of the parameters implicit for our representational lower bound.

Our technique for proving the RNN lower bound differs significantly from that of FFNs. In particular, we will control the representation cost of the $q$STR model, i.e., a lower bound on the norm of $\boldsymbol{\Theta}_{\mathrm{RNN}}$.

We will now present the RNN lower bound, with its proof deferred to Appendix B.5.

**Proposition 6.** *Consider the* 1STR *model where* $\boldsymbol{x} \sim \mathcal{N}(0, \mathbf{I}_{Nd})$ *with a linear link function, i.e.* $y_j = \langle \boldsymbol{u}, \boldsymbol{x}_{t_j} \rangle$ *for some* $\boldsymbol{u} \in \mathbb{S}^{d-1}$. *Further,* $t_i$ *is drawn independently from the rest of the prompt and uniformly from* $[N]$ *for all* $i \in [N]$. *Then, there exists an absolute constant* $c > 0$, *such that*

$$\frac{1}{N} \mathbb{E}\Big[\|\boldsymbol{y} - \hat{\boldsymbol{y}}_{\mathrm{RNN}}(\boldsymbol{p})\|^2\Big] \le c,$$

*implies*

$$d_h \ge \Omega\Big(\frac{N}{\log(1 + \mathfrak{L}^2\|\boldsymbol{U}\|_{op}^2)}\Big), \quad \text{and} \quad \|\boldsymbol{U}\|_{op}^2 \ge \Omega\Big(\frac{N}{\mathfrak{L}^2 \log(1 + d_h)}\Big).$$

**Remark.** Note that the unboundedness of Gaussian random variables is not an issue for approximation here, since $(g(\boldsymbol{x}_1), \ldots, g(\boldsymbol{x}_N))$ is highly concentrated around $\mathbb{S}^{N-1}(\sqrt{N})$. In fact, one can directly assume $(g(\boldsymbol{x}_1), \ldots, g(\boldsymbol{x}_N)) \sim \mathrm{Unif}(\mathbb{S}^{N-1}(\sqrt{N}))$ and derive a similar lower bound. The choice of Gaussian above is only made to simplify the presentation of the proof.

The above proposition has two implications. First, it has a *computational* consequence, implying that any RNN representing the $q$STR models requires a width that grows at least linearly with the context-length $N$. A similar lower bound in terms of bit complexity was derived in [SHT23] using different tools. More importantly, the norm lower bound $\|\boldsymbol{U}\|_{\mathrm{F}} \ge \tilde{\Omega}(\sqrt{N})$ has a *generalization* consequence, which we discuss below.

To translate the above representational cost result to a sample complexity lower bound, we now introduce the parametrization of the output function $f_y$. The exact parametrization of the transition functions will be unimportant, and we will use the notation $f_h^{\rightarrow}(\boldsymbol{h}, \boldsymbol{x}, \boldsymbol{t}; \boldsymbol{\Theta}_h^{\rightarrow})$ to denote a general parameterized function (similarly with $f^{\leftarrow}$). We will assume $f_y$ is given by a feedforward network,

$$f_y(\boldsymbol{U}^{\rightarrow}\boldsymbol{h}^{\rightarrow}, \boldsymbol{U}^{\leftarrow}\boldsymbol{h}^{\leftarrow}, \boldsymbol{x}, \boldsymbol{t}; \boldsymbol{\Theta}_y) = \boldsymbol{W}_{L_y}\sigma\big(\ldots\sigma(\boldsymbol{W}_2\sigma(\boldsymbol{U}\boldsymbol{h} + \boldsymbol{W}_y\boldsymbol{z} + \boldsymbol{b}_y) + \boldsymbol{b}_2)\ldots\big),$$

where $\boldsymbol{h} = (\boldsymbol{h}^{\rightarrow}, \boldsymbol{h}^{\leftarrow}) \in \mathbb{R}^{2d_h}$, $\boldsymbol{z} = (\boldsymbol{x}_i, f_E(\boldsymbol{t}_i, i)) \in \mathbb{R}^{d+d_E}$. Here, $f_E(\boldsymbol{t}_i, i)$ is an arbitrary encoding function with arbitrary dimension $d_E$. Then $\boldsymbol{\Theta}_y = (\boldsymbol{U}, \boldsymbol{W}_y, \boldsymbol{b}_y, \boldsymbol{W}_2, \boldsymbol{b}_2, \ldots, \boldsymbol{W}_{L_y})$, and $\boldsymbol{\Theta}_{\mathrm{RNN}} = (\boldsymbol{U}, \boldsymbol{\Theta}_y, \boldsymbol{\Theta}_h^{\rightarrow}, \boldsymbol{\Theta}_h^{\leftarrow})$. Note that thanks to the homogeneity of ReLU, we can always reparameterize the network by taking $\bar{\boldsymbol{h}} = \boldsymbol{h}/r_h$, $\bar{\boldsymbol{W}}_y = \boldsymbol{W}_y/r_h$, $\bar{\boldsymbol{b}}_y = \boldsymbol{b}_y/r_h$, and $\bar{\boldsymbol{W}}_2 = \boldsymbol{W}_2/r_h$ without changing the prediction function. Thus, in the following, we take $r_h = 1$ without losing the expressive power of the network. We then have the following sample complexity lower bound.

**Theorem 7.** *Consider the* 1STR *model of Proposition 6. Suppose the size of the hidden state, the depth of the prediction function, and the weight norm respectively satisfy* $d_h \le e^{N^c}$, $2 \le L_y \le C$, *and* $\|\mathrm{vec}(\boldsymbol{\Theta}_{\mathrm{RNN}})\|_2 \le e^{N^c/L_y}$ *for some absolute constants* $c < 1$ *and* $C \ge 2$, *and recall we set* $r_h = 1$ *due to homogeneity of the network. Let* $\hat{\boldsymbol{\Theta}}_\varepsilon$ *be the min-norm* $\varepsilon$*-ERM of* $\hat{R}_n^{\mathrm{RNN}}$, *defined in* (2.4). *Then, there exist absolute constants* $c_1, c_2, c_3 > 0$ *such that if* $n \le \mathcal{O}(N^{c_1})$, *for any* $\varepsilon \ge 0$, *with probability at least* $c_2$ *over the training set,*

$$\frac{1}{N}\mathbb{E}\bigg[\Big\|\hat{\boldsymbol{y}}_{\mathrm{RNN}}(\boldsymbol{p}; \hat{\boldsymbol{\Theta}}_{n,\varepsilon}) - \boldsymbol{y}\Big\|_2^2\bigg] \ge c_3.$$

**Remark.** It is possible to remove the subexponential bound on $\|\mathrm{vec}(\boldsymbol{\Theta}_{\mathrm{RNN}})\|$ by allowing the learner to search over families of RNNs with arbitrary $d_h \le e^{N^c}$ rather than fixing a single $d_h$. Additionally, one would avoid solutions that violate this norm constraint in practice due to numerical instability.

To prove the above theorem, we use the fact that an RNN that generalizes on the entire data distribution (hence approximates the 1STR model) requires a weight norm that scales with $\sqrt{N}$, while overfitting on the $n$ samples in the training set with zero empirical risk is possible with a $\text{poly}(n)$ weight norm. As a result, as long as $n \leq N^{c_1}$ for some small constant $c_1 > 0$, min-norm $\varepsilon$-ERM will choose models that overfit rather than generalize. A similar approach was taken in [POW$^+$24] to prove sample complexity separations between two and three-layer feedforward networks. The complete proof is presented in Appendix B.6.

# 5 Feedforward Neural Networks (FFNs)

In this section, we consider a general formulation of a feedforward network. Our only requirement will be that the first layer performs a fully-connected projection. The subsequent layers of the network can be arbitrarily implemented, e.g. using attention blocks or convolution filters. Specifically, the FFN implements the mapping $\boldsymbol{p} \mapsto f(\boldsymbol{T}, \boldsymbol{W}\boldsymbol{x})$ where $\boldsymbol{W} \in \mathbb{R}^{m_1 \times Nd}$ is the weight matrix in the first layer, $\boldsymbol{x} = (\boldsymbol{x}_1^\top, \ldots, \boldsymbol{x}_N^\top)^\top \in \mathbb{R}^{Nd}$, and $f : [N]^{qN} \times \mathbb{R}^{m_1} \to \mathbb{R}^N$ implements the rest of the network. Unlike the Transformer architecture, here we give the network full information of $\boldsymbol{T} = (\boldsymbol{t}_1, \ldots, \boldsymbol{t}_N)$, and in particular the network can implement arbitrary encodings of the position variables $\boldsymbol{t}_1, \ldots, \boldsymbol{t}_N$. This formulation covers usual approaches where encodings of $\boldsymbol{t}$ are added to or concatenated with $\boldsymbol{x}$.

For our negative result on feedforward networks, we can further restrict the class of $q$STR models, and only look at simple-$q$STR where $\hat{R}_n$ of (2.3) and $\hat{R}_{n,N}$ of (2.2) will be equivalent. Additionally, the lower bound of this section holds regardless of the loss function used for training; for some arbitrary loss $\ell : \mathbb{R} \times \mathbb{R} \to \mathbb{R}$, we define the empirical risk of the FFN as

$$\hat{\mathcal{L}}^{\text{FFN}}(f, \boldsymbol{W}) \coloneqq \frac{1}{nN} \sum_{i=1}^{n} \sum_{j=1}^{N} \ell(y_j^{(i)}, f(\boldsymbol{T}^{(i)}, \boldsymbol{W}\boldsymbol{x}^{(i)})_j),$$

where $\boldsymbol{T}^{(i)} = (\boldsymbol{t}_1^{(i)}, \ldots, \boldsymbol{t}_N^{(i)})$. We still use $R^{\text{FFN}}(f, \boldsymbol{W})$ for expected squared loss. Our lower bound covers a broad set of algorithms, characterized by the following definition.

**Definition 8.** *Let $\mathcal{A}_{\text{SP}}$ denote the set of algorithms that return a stationary point of the regularized empirical risk. Specifically, for every $A \in \mathcal{A}_{\text{SP}}$, $A(S_n)$ returns $f_{A(S_n)}$, $\boldsymbol{W}_{A(S_n)}$, such that*

$$\nabla_{\boldsymbol{W}} \hat{\mathcal{L}}^{\text{FFN}}(f_{A(S_n)}, \boldsymbol{W}_{A(S_n)}) + \lambda \boldsymbol{W}_{A(S_n)} = 0,$$

*for some $\lambda > 0$ depending on $A$. $S_n$ above denotes the training set. Let $\mathcal{A}_{\text{ERM}}$ denote the set of algorithms that return the min-norm approximate ERM. Specifically, every $A \in \mathcal{A}_{\text{ERM}}$ returns*

$$A(S_n) = \underset{\{f, \boldsymbol{W} : \hat{\mathcal{L}}^{\text{FFN}}(f, \boldsymbol{W}) \leq \varepsilon\}}{\arg\min} \|\boldsymbol{W}\|_F,$$

*for some $\varepsilon \geq 0$. Define $\mathcal{A} \coloneqq \mathcal{A}_{\text{SP}} \cup \mathcal{A}_{\text{ERM}}$.*

In particular, $\mathcal{A}$ goes beyond constrained ERM in that it also includes the (ideal) output of first-order optimization algorithms with weight decay, or ERM with additional $\ell_2$ penalty on the weights. The following minimax lower bound shows that all algorithms in class $\mathcal{A}$ fail to learn even the subset of simple-$q$STR models with a sample complexity sublinear in $N$.

**Theorem 9.** *Suppose $\boldsymbol{x} \sim \mathcal{N}(0, \mathbf{I}_{Nd})$, and consider the simple-1STR model with $t_{i1} = t_1$ for all $i \in [N]$, where $t_1$ is drawn independently and uniformly in $[N]$, and a linear link function, i.e. $y = \langle \boldsymbol{u}, \boldsymbol{x}_{t_1} \rangle$ for some $\boldsymbol{u} \in \mathbb{S}^{d-1}$. Let $\mathcal{A}$ be the class of algorithms in Definition 8. Then,*

$$\inf_{A \in \mathcal{A}} \sup_{\boldsymbol{u} \in \mathbb{S}^{d-1}} R^{\text{FFN}}(f_{A(S_n)}, \boldsymbol{W}_{A(S_n)}) \geq 1 - \frac{n}{Nd},$$

*with probability 1 over the training set $S_n$.*

**Remark.** The above lower bound implies that learning the simple 1STR model with FFNs requires at least $Nd$ samples. Note that here we do not have any assumption on $m_1$, i.e. the network can have infinite width. This is a crucial difference with the lower bounds in [SHT23, WWHL24] which are computational, i.e., a similar model cannot be learned unless $m_1 \geq Nd$.

The main intuition is that from the stationarity property of Definition 8, the rows of the trained $\boldsymbol{W}$ will always be in the span of the training data $\boldsymbol{x}^{(i)}$ for $i \in [n]$. This is an $n$-dimensional subspace, and the best predictor that only depends on this subspace still has a loss determined by the variance of $y$ conditioned on this subspace. By randomizing the target direction $\boldsymbol{u}$, the label $y$ can depend on all $Nd$ target directions. As a result, as long as $n < Nd$, this variance will be bounded away from zero, leading to the failure of FFNs, even with infinite compute/width. See Appendix D for detailed proof.

## 6 Conclusion

In this paper, we established a sample complexity separation between Transformers and baseline architectures, namely feedforward and recurrent networks, for learning sequence-to-sequence models where the output at each position depends on a sparse subset of input tokens described in the input itself, coined the $q$STR model. We proved that Transformers can learn such a model with sample complexity almost independent of the length of the input sequence $N$, while feedforward and recurrent networks have sample complexity lower bounds of $N$ and $N^{\Omega(1)}$, respectively. Further, we established a separation between FFNs and RNNs by proving that recurrent networks can learn the subset of simple-$q$STR models where the output at all positions is identical, whereas feedforward networks require at least $N$ samples. An important direction for future work is to develop an understanding of the optimization dynamics of Transformers to learn $q$STR models, and to study sample complexity separations that highlight the role of depth in Transformers.

## Acknowledgments and Disclosure of Funding

The authors thank Alberto Bietti and Song Mei for useful discussions. MAE was partially supported by the NSERC Grant [2019-06167], the CIFAR AI Chairs program, the CIFAR Catalyst grant, and the Ontario Early Researcher Award.

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

# A  Details of Section 3

Here we present the omitted details and proofs of Section 3. We begin by presenting the architectural details before proving sample complexity upper bounds for Transformers.

## A.1  Transformer Architectural Definition

We formally introduce the single-layer $H$-headed Transformer that appears in all Section 3 proofs.

**Positional encoding.**  To break the permutation equivaraince of Transformers, we append positional information to the input tokens. Given a prompt $\boldsymbol{p}$, we consider an encoding given by

$$\boldsymbol{Z}(\boldsymbol{p}) = \begin{pmatrix} \boldsymbol{x}_1 & \cdots & \boldsymbol{x}_N \\ \mathrm{enc}(1, \boldsymbol{t}_1) & \cdots & \mathrm{enc}(N, \boldsymbol{t}_N) \end{pmatrix} \in \mathbb{R}^{D_e \times N},$$

where $\mathrm{enc} : [N] \times [N]^q \to \mathbb{R}^{d_{\mathrm{enc}}}$ provides the encoding of the position and of $\boldsymbol{t}_i$, and $D_e := d + d_{\mathrm{enc}}$. We use $\boldsymbol{z}_i$ to refer to the $i$th column above. We remark that allowing $\mathrm{enc}$ to take $\boldsymbol{t}_i$ as input allows specific encodings of the indices $\boldsymbol{t}_i$ that take advantage of the $q$STR structure; examples of this have been considered in prior works [WWHL24]. In practice, we expect such useful encodings to be learned automatically by previous layers in the Transformer. We remark that for a fair comparison, in our lower bounds for other architectures we allow *arbitrary processing* of $\boldsymbol{t}_i$ in their encoding procedure. To specify $\mathrm{enc}$, we use a set of vectors $\{\boldsymbol{\omega}_i\}_{i=1}^N$ in $\mathbb{R}^{d_e}$ that satisfy the following property.

**Assumption 3.** *We have* $|\langle \boldsymbol{\omega}_i, \boldsymbol{\omega}_j \rangle| \le \frac{1}{2}$ *for all* $i \ne j$, *and* $\|\boldsymbol{\omega}_i\|^2 = 1$ *for all* $i$, *with* $d_e = \Theta(\log N)$.

Such a set of vectors can be obtained e.g., by sampling random Rademacher vectors from the unit cube $\{\pm 1/\sqrt{d_e}\}^{d_e}$ which will satisfy the assumption with high probability. We define

$$\mathrm{enc}(i, \boldsymbol{t}_i) = \sqrt{d/q}(\boldsymbol{\omega}_i, \boldsymbol{\omega}_{t_{i1}}, \ldots, \boldsymbol{\omega}_{t_{iq}})^\top \in \mathbb{R}^{(q+1)d_e},$$

hence $d_{\mathrm{enc}} = (q+1)d_e$ and $D_e = d + (q+1)d_e$. The $\sqrt{d/q}$ prefactor ensures that $\boldsymbol{x}_i$ and $\mathrm{enc}(i, \boldsymbol{t}_i)$ will roughly have the same $\ell_2$ norm, resulting in a balanced input to the attention layer.

**Multi-head attention.**  Given a sequence $\{\boldsymbol{z}_i\}_{i=1}^N$ where $\boldsymbol{z}_i \in \mathbb{R}^{D_e}$ with $D_e$ as the embedding dimension, a single head of attention outputs another sequence of length $N$ in $\mathbb{R}^{D_e}$, given by

$$f_{\mathtt{Attn}}(\boldsymbol{p}; \boldsymbol{W}_Q, \boldsymbol{W}_K, \boldsymbol{W}_V) = \left[ \sum_{j=1}^N \boldsymbol{W}_V \boldsymbol{z}_j \frac{e^{\langle \boldsymbol{W}_Q \boldsymbol{z}_i, \boldsymbol{W}_K \boldsymbol{z}_j \rangle}}{\sum_{l=1}^N e^{\langle \boldsymbol{W}_Q \boldsymbol{z}_i, \boldsymbol{W}_K \boldsymbol{z}_l \rangle}} \right]_{i \in [N]}.$$

Where $\boldsymbol{W}_K, \boldsymbol{W}_Q, \boldsymbol{W}_V$ are the key, query, and value projection matrices respectively. We can simplify the presentation by replacing $\boldsymbol{W}_Q^\top \boldsymbol{W}_K$ with a single parameterizing matrix for query-key projections denoted by $\boldsymbol{W}_{\mathrm{QK}} \in \mathbb{R}^{D_e \times D_e}$, and absorbing $\boldsymbol{W}_V$ into the weights of the feedforward layer. This provides us with a simplified parameterization of attention, which we denote by $f_{\mathtt{Attn}}(\boldsymbol{p}; \boldsymbol{W}_{\mathrm{QK}})$. This simplification is standard in theoretical works (see e.g. [LIPO23, ACDS23, ZFB24, WWHL24]). Our main separation results still apply when maintaining separate trainable projections.

We can concatenate the output of $H$ attention heads with separate key-query projection matrices to obtain a multi-head attention layer with $H$ heads. We denote the output of head $h \in [H]$ with $f_{\mathtt{Attn}}(\boldsymbol{p}; \boldsymbol{W}_{\mathrm{QK}}^{(h)})$. The output of the multi-head attention at position $i$ is then given by

$$f_{\mathtt{Attn}}^{(H)}(\boldsymbol{p}; \boldsymbol{W}_{\mathrm{QK}}^{(1)}, \ldots, \boldsymbol{W}_{\mathrm{QK}}^{(H)})_i = (f_{\mathtt{Attn}}(\boldsymbol{p}; \boldsymbol{W}_{\mathrm{QK}}^{(1)})_i, \ldots, f_{\mathtt{Attn}}(\boldsymbol{p}; \boldsymbol{W}_{\mathrm{QK}}^{(H)})_i)^\top \in \mathbb{R}^{HD_e}.$$

We will denote by $\boldsymbol{\Theta}_{\mathrm{QK}} = (\boldsymbol{W}_{\mathrm{QK}}^{(1)}, \ldots, \boldsymbol{W}_{\mathrm{QK}}^{(H)})$ the parameters of the multi-head attention.

Finally, a two-layer neural network acts on the output of the attention to generate labels. Given input $\boldsymbol{h} \in \mathbb{R}^{HD_e}$, the output of the network is given by

$$f_{\mathtt{2NN}}(\boldsymbol{h}; \boldsymbol{a}_{\mathtt{2NN}}, \boldsymbol{W}_{\mathtt{2NN}}, \boldsymbol{b}_{\mathtt{2NN}}) = \boldsymbol{a}_{\mathtt{2NN}}^\top \sigma(\boldsymbol{W}_{\mathtt{2NN}} \boldsymbol{h} + \boldsymbol{b}_{\mathtt{2NN}}),$$

where $\boldsymbol{W}_{\mathtt{2NN}} \in \mathbb{R}^{m \times HD_e}$ are the first layer weights, $\boldsymbol{b}_{\mathtt{2NN}}, \boldsymbol{a}_{\mathtt{2NN}} \in \mathbb{R}^m$ are the second layer weights and biases, and $m$ is the width. We also use the summarized notation $\boldsymbol{\Theta}_{\mathtt{2NN}} = (\boldsymbol{a}_{\mathtt{2NN}}, \boldsymbol{W}_{\mathtt{2NN}}, \boldsymbol{b}_{\mathtt{2NN}})$ to refer to the feedforward layer weights. The prediction of the transformer at position $i$ is given by

$$\hat{y}_{\mathrm{TR}}(\boldsymbol{p}; \boldsymbol{\Theta}_{\mathrm{TR}})_i = f_{\mathtt{2NN}}(f_{\mathtt{Attn}}^{(H)}(\boldsymbol{p}; \boldsymbol{\Theta}_{\mathrm{QK}})_i; \boldsymbol{\Theta}_{\mathtt{2NN}}),$$

where $\boldsymbol{\Theta}_{\mathrm{TR}} = (\boldsymbol{\Theta}_{\mathrm{QK}}, \boldsymbol{\Theta}_{\mathtt{2NN}})$ denotes the overall trainable parameters of the Transformer. We use the notation $\hat{\boldsymbol{y}}_{\mathrm{TR}}(\boldsymbol{p}; \boldsymbol{\Theta}_{\mathrm{TR}}) = (\hat{y}_{\mathrm{TR}}(\boldsymbol{p}; \boldsymbol{\Theta}_{\mathrm{TR}})_1, \ldots, \hat{y}_{\mathrm{TR}}(\boldsymbol{p}; \boldsymbol{\Theta}_{\mathrm{TR}})_N)^\top \in \mathbb{R}^N$ to denote the vectorized output.

## A.2 Proof of Theorem 3

To prove Theorem 3, we will prove the more general theorem below.

**Theorem 10.** *Let* $\hat{\boldsymbol{\Theta}} := \arg\min_{\boldsymbol{\Theta}\in\Theta_{\text{TR}}} \hat{R}_n^{\text{TR}}(\boldsymbol{\Theta})$, *where*

$$\Theta_{\text{TR}} := \left\{ \|\boldsymbol{a}_{\text{2NN}}\|_2 \leq r_a/\sqrt{m}, \|(\boldsymbol{W}_{\text{2NN}}, \boldsymbol{b}_{\text{2NN}})\|_F \leq r_w\sqrt{m}, \left\|\boldsymbol{W}_{\text{QK}}^{(h)}\right\|_{2,1} \leq \alpha \ \forall h \in [H] \right\}.$$

*Suppose* $H = q$, $m = m_g$, *and* $\alpha = \tilde{\Theta}(1)$ *(given in Lemma 11). Then, under Assumptions 1, 2 and 3, with probability at least* $1 - n^{-c}$ *for some absolute constant* $c > 0$, *we have*

$$R^{\text{TR}}(\hat{\boldsymbol{\Theta}}) \leq \mathcal{O}(\varepsilon_{\text{NN}}^2) + \tilde{\mathcal{O}}\left( C_1 \sqrt{\frac{(m_g q(d+q) + r_z^6 r_a^2 r_w^2 q^2 \wedge q(q^2+d^2))}{n}} \right), \tag{A.1}$$

*where* $C_1 = qr_a^2 r_w^2 r_z^2$.

We begin with a lemma establishing the capability of Transformers in approximating $q$STR models.

**Lemma 11.** *Suppose Assumption 2 holds. Let* $r_x = \sqrt{3C_x ed\log(nN)}$. *Assume* $H = q$ *and* $m_g = m$. *Then, there exists* $\boldsymbol{\Theta}_{\text{TR}}$ *such that*

$$\sup_{\{\|\boldsymbol{x}_j\|_2 \leq r_x, \forall j \in [N]\}} \left| g(\boldsymbol{x}_{t_{i1}}, \ldots, \boldsymbol{x}_{t_{iq}}) - \hat{y}_{\text{TR}}(\boldsymbol{p}; \boldsymbol{\Theta}_{\text{TR}})_i \right| \leq 2\sqrt{\varepsilon_{\text{2NN}}},$$

*and*

$$\|\boldsymbol{a}_{\text{2NN}}\|_2 \leq \frac{r_a}{\sqrt{m}}, \quad \|(\boldsymbol{W}_{\text{2NN}}, \boldsymbol{b}_{\text{2NN}})\|_F \leq \sqrt{m}r_w, \quad \left\|\boldsymbol{W}_{\text{QK}}^{(h)\top}\right\|_{2,1} \leq \frac{2d_e q}{d}\log\left(\frac{2r_a r_w r_x N\sqrt{q}}{\varepsilon_{\text{2NN}}}\right),$$

*for all* $h \in [H]$.

**Proof.** In our construction, the goal of attention head $h$ at position $i$ will be to output $\boldsymbol{z}_{t_{ih}}$. Namely, we want to achieve

$$f_{\text{Attn}}(\boldsymbol{p}; \boldsymbol{W}_{\text{QK}}^{(h)})_i \approx \boldsymbol{z}_{t_{ih}}.$$

Note that to do so, for each key token $\boldsymbol{z}_j$, we only need to compute $\langle\boldsymbol{\omega}_{t_{ih}}, \boldsymbol{\omega}_j\rangle$. Therefore, most entries in $\boldsymbol{W}_{\text{QK}}^{(h)}$ can be zero. We only require a block of $d_e \times d_e$, which corresponds to comparing $\boldsymbol{\omega}_j$ and $\boldsymbol{\omega}_{t_{ih}}$ when comparing query $\boldsymbol{z}_i$ and key $\boldsymbol{z}_j$. Thus, we let

$$\boldsymbol{W}_{\text{QK}}^{(h)} = \begin{pmatrix} \boldsymbol{0}_{(d+hd_e)\times d} & \boldsymbol{0}_{(d+hd_e)\times d_e} & \boldsymbol{0}_{(d+hd_e)\times qd_e} \\ \boldsymbol{0}_{d_e\times d} & \alpha\boldsymbol{I}_{d_e} & \boldsymbol{0}_{d_e\times qd_e} \\ \boldsymbol{0}_{(q-h)d_e\times d} & \boldsymbol{0}_{(q-h)d_e\times d_e} & \boldsymbol{0}_{(q-h)d_e\times qd_e} \end{pmatrix} \tag{A.2}$$

Then, we have $\left\langle \boldsymbol{z}_i, \boldsymbol{W}_{\text{QK}}^{(h)}\boldsymbol{z}_j \right\rangle = \alpha\langle\boldsymbol{\omega}_{t_{ih}}, \boldsymbol{\omega}_j\rangle d/q$. We can then verify that

$$\left\| \boldsymbol{A}f_{\text{Attn}}(\boldsymbol{p}; \boldsymbol{W}_{\text{QK}}^{(h)})_i - \boldsymbol{A}\boldsymbol{z}_{t_{ih}} \right\|_2 \leq \sum_{j\neq t_{ih}} e^{-\alpha d/(2q)}(\|\boldsymbol{A}\boldsymbol{z}_j\| + \|\boldsymbol{A}\boldsymbol{z}_{t_{ih}}\|_2)$$

for every matrix $\boldsymbol{A}$. We will specifically choose $\boldsymbol{A}$ to be the projection onto the first $d$ coordinates in the following. Hence, $\alpha$ will control the error in the softmax attention approximating a "hard-max" attention that would exactly choose $\boldsymbol{z}_{t_{ih}}$.

To construct the weights of the feedforward layer $\boldsymbol{a}_{\text{2NN}}, \boldsymbol{W}_{\text{2NN}}, \boldsymbol{b}_{\text{2NN}}$, we let $\boldsymbol{a}_{\text{2NN}} = \boldsymbol{a}_g$ and $\boldsymbol{b}_{\text{2NN}} = \boldsymbol{b}_g$ from Assumption 2, and define $\boldsymbol{W}_{\text{2NN}}$ by extending $\boldsymbol{W}_g$ with zero entries such that

$$\boldsymbol{W}_{\text{2NN}}\begin{pmatrix} \boldsymbol{z}_{t_{i1}} \\ \ldots \\ \boldsymbol{z}_{t_{iq}} \end{pmatrix} = \boldsymbol{W}_g\begin{pmatrix} \boldsymbol{x}_{t_{i1}} \\ \ldots \\ \boldsymbol{x}_{t_{iq}} \end{pmatrix}.$$

Then $\|\boldsymbol{W}_{\text{2NN}}\|_F = \|\boldsymbol{W}_g\|_F$. Notice that $\cdot \mapsto \boldsymbol{a}^\top\sigma(\boldsymbol{W}(\cdot) + \boldsymbol{b})$ is $r_a r_w$ Lipschitz. As a result, for any $\boldsymbol{x}$ with $\|\boldsymbol{x}\| \leq r_x$ we have

$$\left| g(\boldsymbol{x}_{t_{i1}}, \ldots, \boldsymbol{x}_{t_{iq}}) - \hat{y}_{\text{TR}}(\boldsymbol{p}; \boldsymbol{\Theta}_{\text{TR}})_i \right| \leq \sqrt{\varepsilon_{\text{2NN}}} + \varepsilon_{\text{Attn}},$$

where we recall

$$\left|g(\boldsymbol{x}_{t_{i1}},\ldots,\boldsymbol{x}_{t_{iq}}) - f_{\text{2NN}}((\boldsymbol{z}_{t_{i1}},\ldots,\boldsymbol{z}_{t_{iq}}); \boldsymbol{a}_{\text{2NN}}, \boldsymbol{W}_{\text{2NN}}, \boldsymbol{b}_{\text{2NN}})\right| \le \sqrt{\varepsilon_{\text{2NN}}},$$

and

$$\varepsilon_{\text{Attn}} = \left|f_{\text{2NN}}((\boldsymbol{z}_{t_{i1}},\ldots,\boldsymbol{z}_{t_{iq}}); \boldsymbol{\Theta}_{\text{2NN}}) - f_{\text{2NN}}(f_{\text{Attn}}^{(q)}(\boldsymbol{p}; \boldsymbol{\Theta}_{\text{QK}}); \boldsymbol{\Theta}_{\text{2NN}})\right|$$

$$\le r_a r_w \sqrt{\sum_{h=1}^{q} \left\|\boldsymbol{A}f_{\text{Attn}}(\boldsymbol{p}; \boldsymbol{W}_{\text{QK}}^{(h)})_i - \boldsymbol{A}\boldsymbol{z}_{t_{ih}}\right\|_2^2}$$

$$\le 2 r_a r_w r_x N \sqrt{q} e^{-\alpha d/(2q)},$$

where we recall $\boldsymbol{A}\boldsymbol{z}_j = \boldsymbol{x}_j$. Thus, with

$$\alpha = 2q \log(2 r_a r_w r_x N \sqrt{q}/\sqrt{\varepsilon_{\text{2NN}}})/d$$

we can guarantee the distance is at most $2\sqrt{\varepsilon_{\text{2NN}}}$. □

Before proceeding to obtain statistical guarantees, we will show that we can consider the encodings $\boldsymbol{z}_j^{(i)}$ to be bounded with high probability. This will be a useful event to consider throughout the proofs of various sections.

**Lemma 12.** *Suppose $\{\boldsymbol{p}^{(i)}\}_{i=1}^n$ are $n$ input prompts (not necessarily independent) drawn from the input distribution, with tokens denoted by $\{(\boldsymbol{x}_j^{(i)})_{j=1}^N\}_{i=1}^n$. Under Assumption 1, for any $r_x > 0$ we have*

$$\mathbb{P}\left(\max_{i \in [n], j \in [N]} \left\|\boldsymbol{x}_j^{(i)}\right\|_2 \ge r_x\right) \le nN e^{-r_x^2/(2 C_x e d)}.$$

*In particular, for $r_x = \sqrt{3 C_x e d \log(nN)}$ we have*

$$\mathbb{P}\left(\max_{i \in [n], j \in [N]} \left\|\boldsymbol{x}_j^{(i)}\right\|_2 \ge r_x\right) \le \sqrt{\frac{1}{nN}}.$$

**Proof.** Via Markov's inequality, for any $p > 0$ and $r_x > 0$, we have

$$\mathbb{P}\left(\max_{i,j} \left\|\boldsymbol{x}_j^{(i)}\right\|_2 \ge r_x\right) \le \frac{\mathbb{E}\left[\max_{i,j} \left\|\boldsymbol{x}_j^{(i)}\right\|_2^p\right]}{r_x^p} \le \frac{\mathbb{E}\left[\sum_{i,j} \left\|\boldsymbol{x}_j^{(i)}\right\|_2^p\right]}{r_x^p} \le \frac{Nn(C_x pd)^{p/2}}{r_x^p}.$$

Let $p = r_x^2/(C_x e d)$. Then,

$$\mathbb{P}\left(\max_{i,j} \left\|\boldsymbol{x}_j^{(i)}\right\|_2 \ge r_x\right) \le nN e^{-r_x^2/(2 C_x e d)},$$

which proves the first statement, and the second statement follows by plugging in the specific value of $r_x$. □

We are now ready to move to the generalization analysis of Transformers. First, we have to formally define the prediction function class of Transformers with a notation suitable for this section. We begin by defining the function class of attention. We have

$$\mathcal{F}_{\text{Attn}} = \{\boldsymbol{p}, j \mapsto f_{\text{Attn}}^{(H)}(\boldsymbol{p}; \boldsymbol{\Theta}_{\text{QK}})_j : \boldsymbol{\Theta}_{\text{QK}} \in \Theta_{\text{QK}}\},$$

where we will later specify $\Theta_{\text{QK}}$. Additionally, we define $\mathcal{F}_{\text{2NN}}$ by

$$\mathcal{F}_{\text{2NN}} = \{\boldsymbol{h} \mapsto f_{\text{2NN}}(\boldsymbol{h}; \boldsymbol{\Theta}_{\text{2NN}}) : \boldsymbol{\Theta}_{\text{2NN}} \in \Theta_{\text{2NN}}\},$$

where $\boldsymbol{\Theta}_{\text{2NN}} = (\boldsymbol{a}_{\text{2NN}}, \boldsymbol{W}_{\text{2NN}}, \boldsymbol{b}_{\text{2NN}})$, and we will later specify $\Theta_{\text{2NN}}$. Then the class $\mathcal{F}_{\text{TR}}$ can be defined as

$$\mathcal{F}_{\text{TR}} = \{\boldsymbol{p}, j \mapsto f_{\text{2NN}}(f_{\text{Attn}}(\boldsymbol{p})_j) : f_{\text{Attn}} \in \mathcal{F}_{\text{Attn}}, f_{\text{2NN}} \in \mathcal{F}_{\text{2NN}}\}.$$

Recall we use the $S_n$ to denote the training set. To avoid extra indices, we will use the notation $\boldsymbol{p}, j \in S_n$ to go over $\{\boldsymbol{p}^{(i)}, j^{(i)}\}_{i=1}^n$. We can then define the following distances on the introduced function classes

$$d_\infty^{\text{TR}}(f, f') := \sup_{\boldsymbol{p}, j} |f(\boldsymbol{p})_j - f'(\boldsymbol{p})_j|, \quad \forall f, f' \in \mathcal{F}_{\text{TR}}$$

$$d_\infty^{\text{Attn}}(f, f') := \sup_{\boldsymbol{p}, j} \|f(\boldsymbol{p})_j - f'(\boldsymbol{p})_j\|_2, \quad \forall f, f' \in \mathcal{F}_{\text{Attn}}$$

$$d_\infty^{\text{2NN}}(f, f') := \sup_{\|\cdot\|_2 \le \sqrt{H} r_z} |f(\cdot) - f'(\cdot)|, \quad \forall f, f' \in \mathcal{F}_{\text{2NN}}.$$

We choose the radius $\sqrt{H} r_z$ for defining $d_\infty^{\text{2NN}}$ since on the event of Lemma 12, this will be the norm bound on the output of the attention layer at every position.

Recall that for a distance $d_\infty$ and a set $\mathcal{F}$, an $\epsilon$-covering $\hat{\mathcal{F}}$ is a set such that for every $f \in \mathcal{F}$, there exists $\hat{f} \in \hat{\mathcal{F}}$ such that $d_\infty(f, \hat{f}) \le \epsilon$. The $\epsilon$-covering number of $\mathcal{F}$, denoted by $\mathcal{C}(\mathcal{F}, d_\infty, \epsilon)$, is the number of elements of the smallest such $\hat{\mathcal{F}}$. The following lemma relates the covering number of $\mathcal{F}_{\text{TR}}$ to those of $\mathcal{F}_{\text{Attn}}$ and $\mathcal{F}_{\text{2NN}}$.

**Lemma 13.** *Suppose $f_{\text{2NN}}$ is $L_f$ Lipschitz for every $f_{\text{2NN}} \in \mathcal{F}_{\text{2NN}}$. Then, for any $\epsilon_{\text{2NN}}, \epsilon_{\text{Attn}} > 0$, on the event of Lemma 12 we have*

$$\log \mathcal{C}(\mathcal{F}_{\text{TR}}, d_\infty^{\text{TR}}, \epsilon_{\text{2NN}} + L_f \epsilon_{\text{Attn}}) \le \log \mathcal{C}(\mathcal{F}_{\text{2NN}}, d_\infty^{\text{2NN}}, \epsilon_{\text{2NN}}) + \log \mathcal{C}(\mathcal{F}_{\text{Attn}}, d_\infty^{\text{Attn}}, \epsilon_{\text{Attn}}).$$

**Proof.** The proof simply follows from the triangle inequality, namely

$$\sup_{\boldsymbol{p}, j} \left| f_{\text{TR}}(\boldsymbol{p}; \boldsymbol{\Theta}_{\text{TR}})_j - f_{\text{TR}}(\boldsymbol{p}; \hat{\boldsymbol{\Theta}}_{\text{TR}})_j \right| \le \sup_{\|\boldsymbol{h}\|_2 \le \sqrt{H} r_z} \left\| f_{\text{2NN}}(\boldsymbol{h}; \boldsymbol{\Theta}_{\text{NN}}) - f_{\text{2NN}}(\boldsymbol{h}; \hat{\boldsymbol{\Theta}}_{\text{NN}}) \right\|_2$$
$$+ L_f \sup_{\boldsymbol{p}, j} \left\| f_{\text{Attn}}^{(H)}(\boldsymbol{p}; \boldsymbol{\Theta}_{\text{QK}})_j - f_{\text{Attn}}^{(H)}(\boldsymbol{p}; \hat{\boldsymbol{\Theta}}_{\text{QK}})_j \right\|_2.$$

$\square$

We have the following estimate for the covering number of $\mathcal{F}_{\text{2NN}}$.

**Lemma 14.** *Suppose $\|\text{vec}(\boldsymbol{\Theta}_{\text{RNN}})\|_2 \le R$ and $\left\| \boldsymbol{z}_j^{(i)} \right\|_2 \le R$ for all $i \in [n]$ and $j \in [N]$. Then,*

$$\log \mathcal{C}(\mathcal{F}_{\text{2NN}}, d_\infty^{\text{2NN}}, \epsilon) \lesssim m_g H D_e \log(1 + \text{poly}(R)/\epsilon).$$

This is a special case of Lemma 30, proved in Appendix B.

For the next step, define the distance

$$d_\infty^{\text{QK}}(\boldsymbol{\Theta}_{\text{QK}}, \boldsymbol{\Theta}_{\text{QK}}') := \sup_{\boldsymbol{p}, j} \left\| \boldsymbol{\Theta}_{\text{QK}}^\top \boldsymbol{z}_j - \boldsymbol{\Theta}_{\text{QK}}'^\top \boldsymbol{z}_j \right\|_2$$

on $\boldsymbol{\Theta}_{\text{QK}}$, where we recall $\boldsymbol{\Theta}_{\text{QK}} = (\boldsymbol{W}_{\text{QK}}^{(1)}, \ldots, \boldsymbol{W}_{\text{QK}}^{(H)}) \in \mathbb{R}^{D_e \times H D_e}$. The following lemma relates the covering number of the multi-head attention layer to the matrix covering number of the class of attention parameters.

**Lemma 15.** *Suppose $\left\| \boldsymbol{z}_j^{(i)} \right\|_2 \le r_z$ for all $i \in [n]$ and $j \in [N]$. Then,*

$$\log \mathcal{C}(\mathcal{F}_{\text{Attn}}, d_\infty^{\text{Attn}}, \epsilon) \le \log \mathcal{C}\left(\boldsymbol{\Theta}_{\text{QK}}, d_\infty^{\text{QK}}, \frac{\epsilon}{2r_z^2}\right).$$

**Proof.** We recall that $\boldsymbol{Z} \in \mathbb{R}^{N \times D_e}$ denotes the encoded prompt, and $\mathrm{softmax}$ is applied row-wise. For conciseness, Let $\Delta := \sup_{\boldsymbol{p},j} \left\| f_{\mathtt{Attn}}^{(H)}(\boldsymbol{p}; \Theta_{\mathrm{QK}})_j - f_{\mathtt{Attn}}^{(H)}(\boldsymbol{p}; \hat{\Theta}_{\mathrm{QK}})_j \right\|_2^2$. Then we have

$$
\begin{aligned}
\Delta &= \sup_{\boldsymbol{p},j \in S_n} \sum_{h \in [H]} \left\| f_{\mathtt{Attn}}(\boldsymbol{p}; \boldsymbol{W}_{\mathrm{QK}}^{(h)})_j - f_{\mathtt{Attn}}(\boldsymbol{p}; \hat{\boldsymbol{W}}_{\mathrm{QK}}^{(h)})_j \right\|_2^2 \\
&= \sup_{\boldsymbol{p},j \in S_n} \sum_{h \in [H]} \left\| \mathrm{softmax}\left( \boldsymbol{z}_j^\top \boldsymbol{W}_{\mathrm{QK}}^{(h)} \boldsymbol{Z}^\top \right) \boldsymbol{Z} - \mathrm{softmax}\left( \boldsymbol{z}_j^\top \hat{\boldsymbol{W}}_{\mathrm{QK}}^{(h)} \boldsymbol{Z}^\top \right) \boldsymbol{Z} \right\|_2^2 \\
&\leq \sup_{\boldsymbol{p},j \in S_n} \sum_{h \in [H]} \left\| \boldsymbol{Z}^\top \right\|_{2,\infty}^2 \left\| \mathrm{softmax}(\boldsymbol{z}_j^\top \boldsymbol{W}_{\mathrm{QK}}^{(h)} \boldsymbol{Z}^\top)^\top - \mathrm{softmax}(\boldsymbol{z}_j^\top \hat{\boldsymbol{W}}_{\mathrm{QK}}^{(h)} \boldsymbol{Z}^\top)^\top \right\|_1^2,
\end{aligned}
$$

where we used Lemma 39 for the last inequality. Moreover, by [EGKZ22, Corollary A.7],

$$
\begin{aligned}
\left\| \mathrm{softmax}\left( \boldsymbol{z}_j^\top \boldsymbol{W}_{\mathrm{QK}}^{(h)} \boldsymbol{Z}^\top \right)^\top - \mathrm{softmax}\left( \boldsymbol{z}_j^\top \hat{\boldsymbol{W}}_{\mathrm{QK}}^{(h)} \boldsymbol{Z}^\top \right) \right\|_1 &\leq 2 \left\| \boldsymbol{Z} \boldsymbol{W}^{(h)}{}_{\mathrm{QK}}^\top \boldsymbol{z}_j - \boldsymbol{Z} \hat{\boldsymbol{W}}^{(h)}{}_{\mathrm{QK}}^\top \boldsymbol{z}_j \right\|_\infty \\
&\leq 2 \left\| \boldsymbol{Z}^\top \right\|_{2,\infty} \left\| \boldsymbol{W}^{(h)}{}_{\mathrm{QK}}^\top \boldsymbol{z}_j - \hat{\boldsymbol{W}}^{(h)}{}_{\mathrm{QK}}^\top \boldsymbol{z}_j \right\|_2.
\end{aligned}
$$

Consequently,

$$
\begin{aligned}
\Delta &\leq 4 r_z^4 \sup_{\boldsymbol{p},j \in S_n} \sum_{h \in [H]} \left\| \boldsymbol{W}_{\mathrm{QK}}^{(h)}{}^\top \boldsymbol{z}_j - \hat{\boldsymbol{W}}^{(h)}{}_{\mathrm{QK}}^\top \boldsymbol{z}_j \right\|_2^2 \\
&= 4 r_z^4 \sup_{\boldsymbol{p},j \in S_n} \left\| \Theta_{\mathrm{QK}}^\top \boldsymbol{z}_j - \hat{\Theta}_{\mathrm{QK}}^\top \boldsymbol{z}_j \right\|_2^2,
\end{aligned}
$$

which completes the proof. $\qquad\square$

Further, we have the following covering number estimate for $\Theta_{\mathrm{QK}}$.

**Lemma 16.** *Suppose* $\Theta_{\mathrm{QK}} = \{ \|\Theta_{\mathrm{QK}}\|_{2,1} \leq R_{2,1}, \|\Theta_{\mathrm{QK}}\|_F \leq R_F \}$ *and* $\left\| \boldsymbol{z}_j^{(i)} \right\|_2 \leq r_z$ *for all* $i \in [n]$ *and* $j \in [N]$. *Then,*

$$
\log \mathcal{C}\left( \Theta_{\mathrm{QK}}, d_\infty^{\mathrm{QK}}, \epsilon \right) \lesssim \min\left( \frac{r_z^2 R_{2,1}^2 \log(2 H D_e^2)}{\epsilon^2}, H D_e^2 \log\left( 1 + \frac{2 R_F r_z}{\epsilon} \right) \right).
$$

**Proof.** The first estimate comes from Maurey's sparsification lemma [BFT17, Lemma 3.2], while the second estimate is based on the inequality

$$
\left\| \Theta_{\mathrm{QK}}^\top \boldsymbol{z}_j - \hat{\Theta}_{\mathrm{QK}}^\top \boldsymbol{z}_j \right\|_2 \leq r_z \left\| \Theta_{\mathrm{QK}} - \hat{\Theta}_{\mathrm{QK}} \right\|_{\mathrm{F}},
$$

and covering $\Theta_{\mathrm{QK}}$ with the Frobenius norm, see e.g. Lemma 41. $\qquad\square$

Finally, we obtain the following covering number for $\mathcal{F}_{\mathtt{TR}}$.

**Proposition 17.** *Suppose* $\|\boldsymbol{a}_{\mathtt{2NN}}\|_2 \leq r_{m,a}$, $\|(\boldsymbol{W}_{\mathtt{2NN}}, \boldsymbol{b}_{\mathtt{2NN}})\|_F \leq R_{m,w}$, *and* $\left\| \boldsymbol{W}_{\mathrm{QK}}^{(h)} \right\|_{2,1} \leq r_{\mathrm{QK}}$ *for all* $h \in [H]$. *Further assume* $\left\| \boldsymbol{z}_j^{(i)} \right\|_2 \leq r_z$ *for all* $i \in [n]$ *and* $j \in [N]$. *Let* $R := \max(r_{m,a}, R_{m,w}, r_z)$. *Then,*

$$
\begin{aligned}
\log \mathcal{C}\left( \mathcal{F}_{\mathtt{TR}}, d_{\mathcal{F}}, \epsilon \right) \lesssim\, & m_g H D_e \log(1 + R/\epsilon) \\
& + \min\left( \frac{r_z^6 r_{m,a}^2 R_{m,w}^2 H^2 r_{QK}^2 \log(H D_e^2)}{\epsilon^2}, H D_e^2 \log\left( 1 + \frac{\sqrt{H} r_{\mathrm{QK}} r_z^3 r_{m,a} R_{m,w}}{\epsilon} \right) \right).
\end{aligned}
$$

**Proof.** The proof follows from a number of observations. First, given the parameterization in the statement of the proposition, we have $L_f = r_{m,a} R_{m,w}$ in Lemma 13. Moreover, we have

$R_F \leq \sqrt{H} r_{\text{QK}}$ and $R_{2,1} \leq H r_{\text{QK}}$ in Lemma 16. The rest follows from combining the statements of the previous lemmas. □

Next, we will use the covering number bound to provide a bound for Rademacher complexity. Recall that for a class of loss functions $\mathcal{L}$, the empirical and population Rademacher complexities are defined as

$$\hat{\mathfrak{R}}_n(\mathcal{L}) := \mathbb{E}\left[\sup_{\ell \in \mathcal{L}} \frac{1}{n} \sum_{i=1}^{n} \xi_i \ell(\boldsymbol{p}^{(i)}, \boldsymbol{y}^{(i)}, j^{(i)})\right], \quad \mathfrak{R}_n(\mathcal{L}) := \mathbb{E}_{(\boldsymbol{p}, \boldsymbol{y}, j)}\left[\hat{\mathfrak{R}}_n(\mathcal{L})\right]$$

respectively, where $(\xi_i)$ are i.i.d. Rademacher random variables. Let the class of loss functions be defined by

$$\mathcal{L}_\tau := \{(\boldsymbol{p}, \boldsymbol{y}, j) \mapsto (f_{\text{TR}}(\boldsymbol{p})_j - y_j)^2 \wedge \tau : f_{\text{TR}} \in \mathcal{F}_{\text{TR}}\}, \tag{A.3}$$

for some constant $\tau > 0$ to be fixed later. We then have the following bound on Rademacher complexity.

**Lemma 18.** *Suppose* $\max_{i \in [n], j \in [N]} \left\|\boldsymbol{z}_j^{(i)}\right\|_2 \leq r_z$. *For the loss class* $\mathcal{L}_\tau$ *given by* (A.3), *we have*

$$\hat{\mathfrak{R}}_n(\mathcal{L}_\tau) \leq \tilde{\mathcal{O}}\left(\tau\sqrt{\frac{C_1 + (C_2 \wedge C_3)}{n}}\right),$$

*where* $C_1 = m_g H D_e$, $C_2 = r_z^6 r_{m,a}^2 R_{m,w}^2 H^2 r_{QK}^2$, *and* $C_3 = H D_e^2$.

**Proof.** Let $\mathcal{C}(\mathcal{L}, d_\infty^{\mathcal{L}}, \epsilon)$ denote the $\epsilon$-covering number of $\mathcal{L}$, where $\ell(\boldsymbol{p}, \boldsymbol{y}, j) = (f(\boldsymbol{p})_j - y_j)^2 \wedge \tau$ and $\ell'(\boldsymbol{p}, \boldsymbol{y}, j) = (f'(\boldsymbol{p})_j - y_j)^2 \wedge \tau$. Then, for any $\alpha \geq 0$, by a standard chaining argument,

$$\hat{\mathfrak{R}}_n(\mathcal{L}_\tau) \lesssim \alpha + \int_\alpha^\tau \sqrt{\frac{\log \mathcal{C}(\mathcal{L}, d_\infty^{\mathcal{L}}, \epsilon)}{n}} \, d\epsilon.$$

$$\lesssim \alpha + \int_\alpha^\tau \sqrt{\frac{\log \mathcal{C}(\mathcal{F}, d_\infty^{\text{TR}}, \epsilon/(2\sqrt{\tau}))}{n}}$$

$$\lesssim \alpha + \int_\alpha^\tau \sqrt{\frac{C_1 \log(R\sqrt{\tau}/\epsilon)}{n}} \, d\epsilon + \left\{\int_\alpha^\tau \sqrt{\frac{\tau C_2 \log(H D_e^2)}{n\epsilon^2}} \, d\epsilon\right\} \wedge \left\{\int_\alpha^\tau \sqrt{\frac{C_3 \log(1 + C_4\sqrt{\tau}/\epsilon)}{n}} \, d\epsilon\right\}$$

$$\lesssim \alpha + \sqrt{\frac{\tau^2 C_1 \log(R\sqrt{\tau}/\alpha)}{n}} + \left\{\sqrt{\frac{\tau C_2 \log(H D_e^2)}{n}} \log\left(\frac{\tau}{\alpha}\right)\right\} \wedge \left\{\sqrt{\frac{\tau^2 C_3 \log(1 + C_4\sqrt{\tau}/\alpha)}{n}}\right\},$$

where $(C_i)_{i=1}^3$ are given in the statement of the lemma and $C_4 = \sqrt{H} r_{\text{QK}} r_z^3 r_{m,a} R_{m,w}$. Choosing $\alpha = 1/\sqrt{n}$ completes the proof. □

Using standard symmetrization techniques, the above immediately yields a high probability upper bound for the expected truncated loss of any estimator in $\Theta_{\text{TR}}$.

**Corollary 19.** *Let* $\hat{\boldsymbol{\Theta}} = \arg\min_{\boldsymbol{\Theta} \in \Theta_{\text{TR}}} \hat{R}_n^{\text{TR}}(\boldsymbol{\Theta})$, *where* $\Theta_{\text{TR}}$ *is described in Proposition 17. Define* $r_z = \sqrt{r_x^2 + d(1 + 1/q)}$ *where* $r_x$ *is defined in Lemma 12. Let* $C_1$, $C_2$, *and* $C_3$ *be defined as in Lemma 18. Then, with probability at least* $1 - \delta - (nN)^{-1/2}$ *over* $S_n$, *we have*

$$R_\tau^{\text{TR}}(\hat{\boldsymbol{\Theta}}) - \hat{R}_n^{\text{TR}}(\hat{\boldsymbol{\Theta}}) \leq \tilde{\mathcal{O}}\left(\tau\sqrt{\frac{(C_1 + C_2 \wedge C_3)}{n}}\right) + \mathcal{O}\left(\tau\sqrt{\frac{\log(1/\delta)}{n}}\right),$$

*where* $R_\tau^{\text{RNN}}(\hat{\boldsymbol{\Theta}}) := \mathbb{E}_{\boldsymbol{p}, j, y}\left[(\hat{y}_{\text{TR}}(\boldsymbol{p}; \hat{\boldsymbol{\Theta}})_j - y_j)^2 \wedge \tau\right]$

**Proof.** The proof is a standard consequence of Rademacher-based generalization bounds, with the additional observation that

$$\frac{1}{n}\sum_{i=1}^{n} \left(\hat{y}_{\text{TR}}(\boldsymbol{p}^{(i)}; \hat{\boldsymbol{\Theta}})_{j^{(i)}} - y_{j^{(i)}}^{(i)}\right)^2 \wedge \tau \leq \hat{R}_n^{\text{TR}}(\hat{\boldsymbol{\Theta}}).$$

□

The last step in the proof of the generalization bound is to bound $R^{\mathrm{TR}}(\hat{\boldsymbol{\Theta}})$ with $R_\tau^{\mathrm{TR}}(\hat{\boldsymbol{\Theta}})$. This is achieved by the following lemma.

**Lemma 20.** *Define $\kappa^2 := H r_{m,a}^2 R_{m,w}^2 r_z^2$. Then, under Assumption 1, for $\tau \asymp \kappa^2 \log(\kappa^2 N \sqrt{n}) + \log(\kappa^2 \sqrt{n})^s$, we have*

$$R^{\mathrm{TR}}(\hat{\boldsymbol{\Theta}}) - R_\tau^{\mathrm{TR}}(\hat{\boldsymbol{\Theta}}) \leq \sqrt{\frac{1}{n}}.$$

**Proof.** For conciseness, define $\Delta_y := \left|\hat{y}_{\mathrm{TR}}(\boldsymbol{p};\hat{\boldsymbol{\Theta}})_j - y_j\right|$. By the Cuachy-Schwartz inequality, we have

$$R^{\mathrm{TR}}(\hat{\boldsymbol{\Theta}}) = \mathbb{E}\left[\Delta_y^2 \mathbb{1}\left[\Delta_y \leq \sqrt{\tau}\right]\right] + \mathbb{E}\left[\Delta_y^2 \mathbb{1}\left[\Delta_y > \sqrt{\tau}\right]\right]$$
$$\leq R_\tau^{\mathrm{TR}}(\hat{\boldsymbol{\Theta}}) + \mathbb{E}\left[\Delta_y^4\right]^{1/2} \mathbb{P}\left(\Delta_y \geq \sqrt{\tau}\right)^{1/2}.$$

Moreover,

$$\mathbb{E}\left[\Delta_y^4\right]^{1/2} \leq 2\,\mathbb{E}\left[y_j^4\right]^{1/2} + 2\,\mathbb{E}\left[\hat{y}(\boldsymbol{p};\hat{\boldsymbol{\Theta}})_j^4\right]^{1/2}.$$

By Assumption 1, we have $\mathbb{E}\left[y_j^4\right]^{1/2} \lesssim 1$. Additionally, note that

$$\left|\hat{y}(\boldsymbol{p};\hat{\boldsymbol{\Theta}})_j\right| \leq \|\boldsymbol{a}_{\mathrm{2NN}}\|_2 (\sqrt{H}\|\boldsymbol{W}_{\mathrm{2NN}}\|_{\mathrm{F}} \max_{l\in[N]}\|\boldsymbol{z}_l\|_2 + \|\boldsymbol{b}_{\mathrm{2NN}}\|_2)$$
$$\leq \sqrt{H} r_{m,a} R_{m,w} (1 + \max_{l\in[N]}\|\boldsymbol{z}_l\|_2).$$

To bound $\max_{l\in[N]}\|\boldsymbol{z}_l\|_2$, we use the subGaussianity of $\|\boldsymbol{x}_l\|_2$ characterized in Assumption 1. Specifically, for all $r \geq 1$

$$\mathbb{E}\left[\max_{l\in[N]}\|\boldsymbol{x}_l\|_2^4\right] \leq \mathbb{E}\left[\max_{l\in[N]}\|\boldsymbol{x}_l\|_2^{4r}\right]^{1/r} \leq \mathbb{E}\left[\sum_{l=1}^{N}\|\boldsymbol{x}_l\|_2^{4r}\right]^{1/r}$$
$$\leq N^{1/r}\,\mathbb{E}\left[\|\boldsymbol{x}_1\|_2^{4r}\right]^{1/r}$$
$$\lesssim N^{1/r} C_x^2 d^2 r^2$$
$$\lesssim (C_x d \log(N))^2,$$

where the last inequality follows from choosing $r = \log N$. As a result,

$$\mathbb{E}\left[\hat{y}(\boldsymbol{p};\hat{\boldsymbol{\Theta}})_j^4\right]^{1/2} \lesssim H r_{m,a}^2 R_{m,w}^2 r_z^2 \log(N)^2 =: \kappa^2 \log(N)^2.$$

We now turn to bounding the probability. We have

$$\mathbb{P}\left(\Delta_y \geq \sqrt{\tau}\right) \leq \mathbb{P}\left(|y_j| \geq \frac{\sqrt{\tau}}{2}\right) + \mathbb{P}\left(\left|\hat{y}(\boldsymbol{p};\hat{\boldsymbol{\Theta}})_j\right| \geq \frac{\sqrt{\tau}}{2}\right)$$
$$\leq \exp\left(-\Omega(\tau^{1/s})\right) + N\exp\left(-\Omega\left(\frac{\tau}{H r_{m,a}^2 R_{m,w}^2 r_z^2}\right)\right),$$

where the second inequality follows from sub-Weibull concentration bounds for $y$ and Lemma 12. Choosing $\tau = \Theta(\kappa^2 \log(\kappa^2 N \sqrt{n}) + \log(\kappa^2 \sqrt{n})^s)$ completes the proof. $\qquad\square$

**Proof of Theorem 10.** The theorem follows immediately from the approximation guarantee of Lemma 11, the generalization bound of Corollary 19, and the truncation control of Lemma 20. $\quad\square$

### A.3 Details on Limitations of Transformers with Few Heads

While Proposition 4 is only meaningful in the setting of $d = \Omega(q)$, the following proposition provides an exact lower bound $H \geq q$ on the number of heads for all $d$, at the expense of additional restrictions on the attention matrix.

**Proposition 21.** *Consider the qSTR data model. Suppose $d = 1$ and $y_i = \frac{1}{\sqrt{q}}\sum_{j=1}^{q}(x_{t_{ij}}^2 - \mathbb{E}[x_{t_{ij}}^2])$. Assume $x_i \sim \mathcal{N}(0, \sigma_i^2)$ independently, such that $\sigma_i = 1$ for $i < N/2$ and $\sigma_i = 0$ for $i \geq N/2$. Further, assume the attention weights between the data and positional encoding parts of the tokens are fixed at zero, i.e. $\boldsymbol{W}_{\mathrm{QK}}^{(h)} = \begin{pmatrix} \boldsymbol{W}_{\boldsymbol{x}}^{(h)} & \boldsymbol{0}_{d\times(q+1)d_e} \\ \boldsymbol{0}_{(q+1)d_e \times d} & \boldsymbol{W}_{\boldsymbol{\omega}}^{(h)} \end{pmatrix}$ where $\boldsymbol{W}_{\boldsymbol{x}}^{(h)} \in \mathbb{R}^{d\times d}$ and $\boldsymbol{W}_{\boldsymbol{\omega}}^{(h)} \in \mathbb{R}^{(q+1)d_e \times (q+1)d_e}$ are the attention parameters, for $i \in [H]$. Then, there exists a distribution over $(\boldsymbol{t}_i)_{i\in[N]}$ such that for any choice of $\boldsymbol{\Theta}_{\mathrm{TR}}$, we have*

$$\frac{1}{N}\mathbb{E}\Big[\|\boldsymbol{y} - \hat{\boldsymbol{y}}_{\mathrm{TR}}(\boldsymbol{p}; \boldsymbol{\Theta}_{\mathrm{TR}})\|_2^2\Big] \geq 1 - \frac{H}{q}.$$

Note that in our approximation constructions for learning $q$STR, we always fixed the attention weights between data and positional components to be zero, which is why we assume the same in Proposition 21.

**Proof of Proposition 21.** We will simply choose $\boldsymbol{t}_i = (1, \ldots, q)$ deterministically for $i \geq \frac{N}{2}$ and draw $\boldsymbol{t}_i$ from an arbitrary distribution for $i < N/2$. Note that we have

$$R^{\mathrm{TR}}(\boldsymbol{\Theta}_{\mathrm{TR}}) = \frac{1}{N}\sum_{i=1}^{N}\mathbb{E}\big[(y_i - \hat{y}_{\mathrm{TR}}(\boldsymbol{p}; \boldsymbol{\Theta}_{\mathrm{TR}})_i)^2\big] \geq \frac{1}{N}\sum_{i=N/2}^{N}\mathbb{E}\big[(y_i - \hat{y}_{\mathrm{TR}}(\boldsymbol{p}; \boldsymbol{\Theta}_{\mathrm{TR}})_i)^2\big].$$

Let $\phi : \mathbb{R}^{HD_e} \to \mathbb{R}$ denote the mapping by the feedforward layer. Fix some $i \geq N/2$. Note that

$$\begin{aligned}
\hat{y}_{\mathrm{TR}}(\boldsymbol{p}; \boldsymbol{\Theta}_{\mathrm{TR}})_i &= \phi(f_{\mathrm{Attn}}^{(H)}(\boldsymbol{p}; \boldsymbol{\Theta}_{\mathrm{QK}})_i) \\
&= \phi\big(\sum_{j=1}^{N}\alpha_{ij}^{(1)}\boldsymbol{z}_j, \ldots, \sum_{j=1}^{N}\alpha_{ij}^{(H)}\boldsymbol{z}_j\big) \\
&= \tilde{\phi}\Big(\sum_{j=1}^{q}\alpha_{ij}^{(1)}x_j, \ldots, \sum_{j=1}^{q}\alpha_{ij}^{(H)}x_j, (\boldsymbol{z}_l)_{l=q+1}^{N}\Big),
\end{aligned}$$

for some real-valued function $\tilde{\phi}$, where

$$\alpha_{ij}^{(h)} = \frac{e^{\langle \boldsymbol{z}_i, \boldsymbol{W}_{\mathrm{QK}}^{(h)}\boldsymbol{z}_j\rangle}}{\sum_{l=1}^{N}e^{\langle \boldsymbol{z}_i, \boldsymbol{W}_{\mathrm{QK}}^{(h)}\boldsymbol{z}_j\rangle}},$$

are the attention scores. Let $\boldsymbol{A}^{(i)} \in \mathbb{R}^{H\times q}$ be the matrix such that $A_{hj}^{(i)} = \alpha_{ij}^{(h)}$. Let $\boldsymbol{x}_{1:q} = (x_1, \ldots, x_q)^\top \in \mathbb{R}^q$. Then,

$$\begin{aligned}
R^{\mathrm{TR}}(\boldsymbol{\Theta}_{\mathrm{TR}}) &\geq \frac{1}{N}\sum_{i=N/2}^{N}\mathbb{E}\Big[\Big(y_i - \tilde{\phi}\Big(\boldsymbol{A}^{(i)}\boldsymbol{x}_{1:q}, (\boldsymbol{z}_l)_{l=q+1}^{N}\Big)\Big)^2\Big] \\
&\geq \frac{1}{Nq}\sum_{i=N/2}^{N}\mathbb{E}\Big[\mathrm{Var}\Big(\|\boldsymbol{x}_{1:q}\|^2 \mid \boldsymbol{V}^{(i)}\boldsymbol{x}_{1:q}\Big)\Big] \qquad\qquad (\mathrm{A.4})
\end{aligned}$$

where $\boldsymbol{V}^{(i)} \in \mathbb{R}^{H\times q}$ is a matrix whose rows form an orthonormal basis of $\mathrm{span}(\boldsymbol{\alpha}_i^{(1)}, \ldots, \boldsymbol{\alpha}_i^{(H)})$ where $\boldsymbol{\alpha}_i^{(h)} = (\alpha_{i1}^{(h)}, \ldots, \alpha_{iq}^{(h)})^\top \in \mathbb{R}^q$ (note that $\boldsymbol{V}^{(i)}$ may have fewer than $H$ rows, we consider the worst-case for the lower bound which is having $H$ rows). The second inequality follows from the fact that $\boldsymbol{z}_l$ is independent of $\boldsymbol{x}_{1:q}$ for $l \geq q+1$, and the fact that best predictor of $y_i$ (in $L_2$ error) given $\boldsymbol{A}^{(i)}\boldsymbol{x}_{1:q}$ is $\mathbb{E}\Big[y_i \mid \boldsymbol{V}^{(i)}\boldsymbol{x}_{1:q}\Big]$.

Next, thanks to the structural property of $\boldsymbol{W}_{\mathrm{QK}}^{(h)}$ in the assumption of the proposition and the fact that $x_i = 0$ for $i \geq N/2$, $\alpha_{ij}^{(h)}$ does not depend on $(x_l)_{l\in[q]}$ for all $h \in [H]$, $i \geq N/2$, and $j \in [q]$. As a result, $\boldsymbol{V}^{(i)}$ is independent of $\boldsymbol{x}_{1:q}$. Therefore,

$$\boldsymbol{x}_{1:q} \mid \boldsymbol{V}^{(i)}\boldsymbol{x}_{1:q} \sim \mathcal{N}(\boldsymbol{V}^{(i)\top}\boldsymbol{V}^{(i)}\boldsymbol{x}_{1:q}, \boldsymbol{I}_q - \boldsymbol{V}^{(i)\top}\boldsymbol{V}^{(i)}).$$

By Lemma 40, we have $\mathrm{Var}(\|\boldsymbol{x}_{1:q}\|^2 \mid \boldsymbol{V}^{(i)}\boldsymbol{x}_{1:q}) = 2(q - H)$, which combined with (A.4) completes the proof. $\qquad\square$

We now present the similarly structured proof of Proposition 4.

**Proof of Proposition 4.** The choice of distribution over $(\boldsymbol{t}_i)_{i \geq N/2}$ is similar to the one presented above, i.e. we let $\boldsymbol{t}_i = (1, \dots, q)$ deterministically for $i \geq \frac{N}{2}$. However, for $i < \frac{N}{2}$, we draw $\boldsymbol{t}_i$ such that they are independent from $\boldsymbol{x}$. Once again, we use the fact that

$$R^{\mathrm{TR}}(\boldsymbol{\Theta}_{\mathrm{TR}}) \geq \frac{1}{N} \sum_{i=N/2}^{N} \mathbb{E}\big[(y_i - \hat{y}_{\mathrm{TR}}(\boldsymbol{p}; \boldsymbol{\Theta}_{\mathrm{TR}})_i)^2\big].$$

Recall $\boldsymbol{z}_i = (\boldsymbol{x}_i^\top, \mathrm{enc}(i, \boldsymbol{t}_i)^\top)$. Fix some $i \geq N/2$, and define

$$\tilde{\alpha}_{ij}^{(h)} = e^{\left\langle \mathrm{enc}(i, \boldsymbol{t}_i), \boldsymbol{W}_{\mathrm{QK}}^{(h,e,x)} \boldsymbol{x}_j \right\rangle + \left\langle \mathrm{enc}(i, \boldsymbol{t}_i), \boldsymbol{W}_{\mathrm{QK}}^{(h,e,e)} \mathrm{enc}(j, \boldsymbol{t}_j) \right\rangle},$$

where we use the notation

$$\boldsymbol{W}_{\mathrm{QK}}^{(h)} = \begin{pmatrix} \boldsymbol{W}_{\mathrm{QK}}^{(h,x,x)} & \boldsymbol{W}_{\mathrm{QK}}^{(h,x,e)} \\ \boldsymbol{W}_{\mathrm{QK}}^{(h,e,x)} & \boldsymbol{W}_{\mathrm{QK}}^{(h,e,e)} \end{pmatrix},$$

for the query-key matrix of each head. Recall that $\boldsymbol{x}_i = 0$ for $i < N/2$, thus the attention weights are given by

$$\alpha_{ij}^{(h)} = \frac{\tilde{\alpha}_{ij}^{(h)}}{\sum_{l=1}^{N} \tilde{\alpha}_{il}^{(h)}}.$$

Recall from the proof of Proposition 21 that we denote the feedforward layer by $\phi : \mathbb{R}^{HD_e} \to \mathbb{R}$. With this notation, we have

$$\hat{y}_{\mathrm{TR}}(\boldsymbol{p}; \boldsymbol{\Theta}_{\mathrm{TR}})_i = \phi\big(\sum_{j=1}^{N} \alpha_{ij}^{(1)} \boldsymbol{z}_j, \dots, \sum_{j=1}^{N} \alpha_{ij}^{(H)} \boldsymbol{z}_j\big)$$

$$= \tilde{\phi}\Big(\sum_{j=1}^{q} \alpha_{ij}^{(1)} \boldsymbol{x}_j, \dots, \sum_{j=1}^{q} \alpha_{ij}^{(H)} \boldsymbol{x}_j, (\tilde{\alpha}_{ij}^{(h)})_{h=1,j=1}^{h=H,j=N}, (\boldsymbol{z}_j)_{j=l+1}^{N}\Big).$$

Therefore, using the fact that $\boldsymbol{z}_j$ and $\tilde{\alpha}_{ij}^{(h)}$ are independent of $\boldsymbol{x}_{1:q}$ for $j \geq l+1$, we have

$$R^{\mathrm{TR}}(\boldsymbol{\Theta}_{\mathrm{TR}}) = \frac{1}{N} \sum_{i=N/2}^{N} \mathbb{E}\left[\left(y_i - \tilde{\phi}\Big(\sum_{j=1}^{q} \alpha_{ij}^{(1)} \boldsymbol{x}_j, \dots, \sum_{j=1}^{q} \alpha_{ij}^{(H)} \boldsymbol{x}_j, (\tilde{\alpha}_{ij}^{(h)})_{h=1,j=1}^{h=H,j=N}, (\boldsymbol{z}_j)_{j=l+1}^{N}\Big)\right)^2\right]$$

$$\geq \frac{1}{Nqd} \sum_{i=N/2}^{N} \mathbb{E}\left[\mathrm{Var}\Big(\|\boldsymbol{x}_{1:q}\|^2 \mid \big(\big\langle \boldsymbol{\alpha}_i^{(h,r)}, \boldsymbol{x}_{1:q} \big\rangle\big)_{h=1,r=1}^{h=H,r=d}, (\tilde{\alpha}_{ij}^{(h)})_{h=1,j=1}^{h=H,j=q}\Big)\right]$$

$$\geq \frac{1}{Nqd} \sum_{i=N/2}^{N} \mathbb{E}\left[\mathrm{Var}\Big(\|\boldsymbol{x}_{1:q}\|^2 \mid \big(\big\langle \boldsymbol{\alpha}_i^{(h,r)}, \boldsymbol{x}_{1:q} \big\rangle\big)_{h=1,r=1}^{h=H,r=d}, \big(\big\langle \boldsymbol{w}_{i,j}^{(h)}, \boldsymbol{x}_{1:q} \big\rangle\big)_{h=1,j=1}^{H,q}\Big)\right]$$

$$= \frac{1}{Nqd} \sum_{i=N/2}^{N} \mathbb{E}\left[\mathrm{var}\Big(\|\boldsymbol{x}_{1:q}\|^2 \mid \boldsymbol{V}^{(i)}\boldsymbol{x}_{1:q}\Big)\right],$$

where $\boldsymbol{\alpha}_i^{(h,r)} \in \mathbb{R}^{qd}$ such that

$$(\alpha_i^{(h,r)})_{jl} = \begin{cases} \alpha_{ij}^{(h)}, & \text{if } l = r \\ 0, & \text{if } l \neq r, \end{cases}$$

which yields $\big\langle \boldsymbol{\alpha}_i^{(h,r)}, \boldsymbol{x}_{1:q} \big\rangle = \sum_{j=1}^{q} \alpha_{ij}^{(h)} x_{jr}$, and $\boldsymbol{w}_{i,j}^{(h)} \in \mathbb{R}^{qd}$ such that

$$(w_{i,j}^{(h)})_{sl} = \begin{cases} \big(\boldsymbol{W}_{\mathrm{QK}}^{(h,e,x)\top} \mathrm{enc}(i, \boldsymbol{t}_i)\big)_l, & \text{if } s = j \\ 0 & \text{if } s \neq j, \end{cases}$$

which yields $\left\langle \boldsymbol{w}_{i,j}^{(h)}, \boldsymbol{x}_{1:q} \right\rangle = \left\langle \boldsymbol{W}^{(h,e,x)\top} \mathrm{enc}(i, \boldsymbol{t}_i), \boldsymbol{x}_j \right\rangle$. Finally, $\boldsymbol{V}^{(i)}$ is a matrix whose rows form an orthonormal basis of $\mathrm{span}\left( \left( \boldsymbol{\alpha}_i^{(h,r)} \right)_{h=1,r=1}^{h=H,r=d}, \left( \boldsymbol{w}_{i,j}^{(h)} \right)_{h=1,j=1}^{h=H,j=q} \right)$. Namely, $\boldsymbol{V}^{(i)}$ has at most $H(d+q)$ rows. Recall that

$$\boldsymbol{x}_{1:q} \,|\, \boldsymbol{V}^{(i)} \boldsymbol{x}_{1:q} \sim \mathcal{N}(\boldsymbol{V}^{(i)\top} \boldsymbol{V}^{(i)} \boldsymbol{x}_{1:q}, \mathbf{I}_{qd} - \boldsymbol{V}^{(i)\top} \boldsymbol{V}^{(i)}).$$

Once again, by Lemma 40, we conclude that $\mathrm{var}(\|\boldsymbol{x}_{1:q}\|^2 \,|\, \boldsymbol{V}^{(i)} \boldsymbol{x}_{1:q}) \geq 2(qd - H(q+d))$, which completes the proof. $\qquad\square$

# B  Details and Proofs of Section 4

Before presenting the proofs, we state the omitted setup and parameterization of the network in the next section.

## B.1  Complete Setup of RNNs

When introducing RNNs in Section 4, we used $L_h$-layer deep feedforward networks to implement the transitions $f_h^\rightarrow(\cdot; \boldsymbol{\Theta}_h^\rightarrow)$ and $f_h^\leftarrow(\cdot; \boldsymbol{\Theta}_h^\leftarrow)$. These transitions are given by

$$f_h^\rightarrow(\cdot; \boldsymbol{\Theta}_h^\rightarrow) = \boldsymbol{W}_{L_h}^\rightarrow \sigma\big( \boldsymbol{W}_{L_h-1}^\rightarrow \ldots \sigma(\boldsymbol{W}_2^\rightarrow \sigma(\boldsymbol{W}_1^\rightarrow(\cdot) + \boldsymbol{b}_1^\rightarrow) + \boldsymbol{b}_2^\rightarrow) \ldots + \boldsymbol{b}_{L_h-1}^\rightarrow \big), \qquad \text{(B.1)}$$

with $\boldsymbol{\Theta}_h^\rightarrow = (\boldsymbol{W}_1^\rightarrow, \boldsymbol{b}_1^\rightarrow, \ldots, \boldsymbol{W}_{L_h-1}^\rightarrow, \boldsymbol{b}_{L_h-1}^\rightarrow, \boldsymbol{W}_{L_h}^\rightarrow)$ and a similar equation for $f^\leftarrow(\cdot; \boldsymbol{\Theta}_h^\leftarrow)$. Recall that the output of the RNN is denoted by

$$\hat{\boldsymbol{y}}_{\mathrm{RNN}}(\boldsymbol{p}; \boldsymbol{\Theta}_{\mathrm{RNN}}) = (f_y(\boldsymbol{h}_1^\rightarrow, \boldsymbol{h}_1^\leftarrow, \boldsymbol{z}_1; \boldsymbol{\Theta}_y), \ldots, f_y(\boldsymbol{h}_N^\rightarrow, \boldsymbol{h}_N^\leftarrow, \boldsymbol{z}_N; \boldsymbol{\Theta}_y)) \in \mathbb{R}^N.$$

We now define the constraint set of this architecture. Let

$$\boldsymbol{\Theta}_{\mathrm{RNN}} = \left\{ \boldsymbol{\Theta} \,:\, \|\mathrm{vec}(\boldsymbol{\Theta})\|_2 \leq R, \left\|\boldsymbol{W}_{L_h}^\rightarrow\right\|_{\mathrm{op}} \ldots \left\|\boldsymbol{W}_{1,h}^\rightarrow\right\|_{\mathrm{op}} \leq \alpha_N, \left\|\boldsymbol{W}_{L_h}^\leftarrow\right\|_{\mathrm{op}} \ldots \left\|\boldsymbol{W}_{1,h}^\leftarrow\right\|_{\mathrm{op}} \leq \alpha_N \right\},$$
$$\text{(B.2)}$$

where $\boldsymbol{W}_{1,h}^\rightarrow$ contains the first $d_h$ columns of $\boldsymbol{W}_1^\rightarrow$, and the conditions above are introduced to ensure $f_h^\rightarrow$ and $f_h^\leftarrow$ are at most $\alpha_N$-Lipschitz with respect to the hidden state input. One way to meet this requirement is to multiply $\boldsymbol{W}_{1,h}^\rightarrow$ by a factor of $\alpha_N / \prod_{l=2}^{L_h} \|\boldsymbol{W}_l^\rightarrow\|_{\mathrm{op}}$ in the forward pass. Without this Lipschitzness constraint, current techniques for proving uniform RNN generalization bounds will suffer from a sample complexity linear in $N$, see e.g. [CLZ20].

For Theorem 5 we only require $\alpha_N \leq N^{-1}$. In particular, we can choose $\alpha_N = 0$ and fix $\boldsymbol{W}_{1,h}^\rightarrow = \boldsymbol{W}_{1,h}^\leftarrow = \mathbf{0}$, which would simplify the parameterization of the network. Namely, in our construction $f^\rightarrow$ and $f^\leftarrow$ do not need to depend on $\boldsymbol{h}^\rightarrow$ and $\boldsymbol{h}^\leftarrow$ respectively.

## B.2  Overview of the Proof of Theorem 5

The following is the roadmap we will take for the proof of Section 4.1. The goal here is to implement a bi-directional RNN in such a way that

$$\boldsymbol{h}_i^\rightarrow \approx \big( \boldsymbol{x}_{t_1} \mathbb{1}[t_1 < i], \ldots, \boldsymbol{x}_{t_q} \mathbb{1}[t_q < i] \big),$$

and

$$\boldsymbol{h}_i^\leftarrow \approx \big( \boldsymbol{x}_{t_1} \mathbb{1}[t_1 > i], \ldots, \boldsymbol{x}_{t_q} \mathbb{1}[t_q > i] \big).$$

Throughout this section, we will use the notation

$$\Psi(\boldsymbol{x}, \boldsymbol{t}, i) = (\boldsymbol{x}^\top \mathbb{1}[t_1 = i], \ldots, \boldsymbol{x}^\top \mathbb{1}[t_q = i])^\top.$$

We can obtain the hidden states above through the following updates

$$\boldsymbol{h}_{i+1}^\rightarrow = \boldsymbol{h}_i^\rightarrow + \Psi(\boldsymbol{x}_i, \boldsymbol{\omega_t}, \boldsymbol{\omega}_i),$$

and

$$\boldsymbol{h}_{i-1}^\leftarrow = \boldsymbol{h}_i^\leftarrow + \Psi(\boldsymbol{x}_i, \boldsymbol{\omega_t}, \boldsymbol{\omega}_i).$$

where

$$\Psi(\boldsymbol{x}_i, \boldsymbol{\omega_t}, \boldsymbol{\omega}_i)_l = \frac{\boldsymbol{x}_i \sigma(\langle \boldsymbol{\omega}_i, \boldsymbol{\omega}_{t_l} \rangle - \delta)}{1 - \delta} = \boldsymbol{x}_i \mathbb{1}[t_l = i], \quad \forall l \in [q]$$

where we recall $\boldsymbol{\omega_t} = (\boldsymbol{\omega}_{t_1}, \ldots, \boldsymbol{\omega}_{t_q})$, and $\sigma$ is ReLU. As a result, our network must approximate

$$f_h^{\rightarrow}(\boldsymbol{h}_i^{\rightarrow}, \boldsymbol{x}_i, \boldsymbol{\omega_t}, \boldsymbol{\omega}_i; \boldsymbol{\Theta}_h^{\rightarrow}) = f_h^{\leftarrow}(\boldsymbol{h}_i^{\leftarrow}, \boldsymbol{x}_i, \boldsymbol{\omega_t}, \boldsymbol{\omega}_i; \boldsymbol{\Theta}_h^{\leftarrow}) \approx \Psi(\boldsymbol{x}_i, \boldsymbol{\omega_t}, \boldsymbol{\omega}_i).$$

A core challenge in this approximation is that if we simply control

$$\|f_h^{\rightarrow}(\boldsymbol{h}_i^{\rightarrow}, \boldsymbol{z}_i; \boldsymbol{\Theta}_h^{\rightarrow}) - \Psi(\boldsymbol{x}_i, \boldsymbol{\omega_t}, \boldsymbol{\omega}_i)\|_2 \le \varepsilon, \tag{B.3}$$

this error will propoagte through the forward pass, and we will have

$$\left\| \boldsymbol{h}_i^{\rightarrow} - \sum_{j=1}^{i-1} \Psi(\boldsymbol{x}_j, \boldsymbol{\omega_t}, \boldsymbol{\omega}_j) \right\|_2 \lesssim N\varepsilon.$$

As a result, we would like an implementation that satisfies the following

$$\|f_h^{\rightarrow}(\boldsymbol{h}_i^{\rightarrow}, \boldsymbol{z}_i; \boldsymbol{\Theta}_h^{\rightarrow})_l - \Psi(\boldsymbol{x}_i, \boldsymbol{\omega_t}, \boldsymbol{\omega}_i)_l\|_2 \le \begin{cases} 0 & t_l \ne i \\ \varepsilon & t_l = i. \end{cases} \tag{B.4}$$

Note that

$$\boldsymbol{h}_i^{\rightarrow} = \sum_{j=1}^{i-1} f_h^{\rightarrow}(\boldsymbol{h}_j^{\rightarrow}, \boldsymbol{z}_j; \boldsymbol{\Theta}_h^{\rightarrow}).$$

Since for each $l \in [q]$, $t_l = j$ is possible for at most one $j \in [N]$, (B.4) implies

$$\left\| \boldsymbol{h}_i^{\rightarrow} - \sum_{j=1}^{i-1} \Psi(\boldsymbol{x}_j, \boldsymbol{\omega_t}, \boldsymbol{\omega}_j) \right\|_2 \le \sqrt{q}\varepsilon,$$

for all $i \in [N]$, hence, we can avoid dependence on $N$.

We can implmenet $f_h^{\rightarrow}$ to satisfy (B.3) with a depth three network, where the first two layers implements $\langle \boldsymbol{\omega}_i, \boldsymbol{\omega}_{t_j} \rangle$ (as a sum of Lipschitz 2-dimensional functions, an example of their approximation is given by [Bac17, Proposition 6]), and the third performs coordinate-wise product between $\boldsymbol{x}_i$ and $\sigma(\langle \boldsymbol{\omega}_i, \boldsymbol{\omega}_{t_j} \rangle - 1/2)$ (which for each coordinate is a Lipschitz two-dimensional function). To ensure $f_h^{\rightarrow}$ satisfies (B.4), we can pass the outputs to a fourth layer which rectifies its input near zero to be exactly zero using ReLU activations.

To generate $y_i$ from $\boldsymbol{h}_i^{\rightarrow}$ and $\boldsymbol{h}_i^{\leftarrow}$, we first calculate

$$\begin{aligned} \boldsymbol{h}_i &= f_{hh}(\boldsymbol{h}_i^{\rightarrow}, \boldsymbol{h}_i^{\leftarrow}, \boldsymbol{x}_i, \boldsymbol{\omega}_i, \boldsymbol{\omega_t}) \\ &\approx \boldsymbol{h}_i^{\rightarrow} + \boldsymbol{h}_i^{\leftarrow} + \Psi(\boldsymbol{x}_i, \boldsymbol{\omega_t}, \boldsymbol{\omega}_i) \\ &\approx (\boldsymbol{x}_{t_1}, \ldots, \boldsymbol{x}_{t_q}). \end{aligned}$$

Finally, $y_i$ can be generated from $\boldsymbol{h}_i$ by applying the two-layer neural network from Assumption 2 that approximates $y_i = g(\boldsymbol{x_t})$.

Note that the construction above has a complexity $\mathrm{poly}(d, q, \log(nN))$ (both in terms of number and weight of parameters), only depending on $N$ up to log factors. As a result, by a simple parameter-counting approach, the sample complexity of regularized ERM would also be (almost) independent of $N$. We also simply use the encoding

$$\boldsymbol{z}_i = (\boldsymbol{x}_i, \boldsymbol{\omega}_i, \boldsymbol{\omega}_{t_{i1}}, \ldots, \boldsymbol{\omega}_{t_{iq}})^\top,$$

for the RNN positive result. The scaling difference with the encoding for Transofrmers is only made to simplify the exposition, as we no longer keep explicit dependence on $d$ and $q$.

## B.3  Approximations

As explained above, to implement $f_h^{\rightarrow}$ we first construct a depth three neural network (with two layers of non-linearity) which approximately performs the following mapping

$$\begin{pmatrix} \boldsymbol{h} \\ \boldsymbol{x} \\ \boldsymbol{\omega}_i \\ \boldsymbol{\omega}_{t_1} \\ \vdots \\ \boldsymbol{\omega}_{t_q} \end{pmatrix} \mapsto \begin{pmatrix} \boldsymbol{x} \\ \langle \boldsymbol{\omega}_i, \boldsymbol{\omega}_{t_1} \rangle \\ \vdots \\ \langle \boldsymbol{\omega}_i, \boldsymbol{\omega}_{t_q} \rangle \end{pmatrix} \mapsto \begin{pmatrix} 2\boldsymbol{x}\sigma(\langle \boldsymbol{\omega}_i, \boldsymbol{\omega}_{t_1} \rangle - 1/2) \\ \vdots \\ 2\boldsymbol{x}\sigma(\langle \boldsymbol{\omega}_i, \boldsymbol{\omega}_{t_q} \rangle - 1/2) \end{pmatrix}.$$

The first mapping will be provided by

$$\chi_1 = A_1 \sigma(W_1 \chi_0 + b_1),$$

where $\chi_0 = (h^\top, x^\top, \omega_i^\top, \omega_{t_1}^\top, \dots, \omega_{t_q}^\top)^\top \in \mathbb{R}^{d_h + d + (q+1)d_e}$, $W_1 \in \mathbb{R}^{m_1 \times (d_h + d + (q+1)d_e)}$, $b_1 \in \mathbb{R}^{m_1}$, and $A_1 \in \mathbb{R}^{(d+q) \times m_1}$, with $m_1$ as the width of the first layer. We will use the notation

$$\chi_1 = (\chi_1^x, \chi_1^\omega(1), \dots, \chi_1^\omega(q))$$

to refer for the first $d$ coordinates and the rest of the $q$ coordinates of $\chi_1$ respectively, thus ideally $\chi_1^x = x$ and $\chi_1^\omega(l) = \langle \omega_i, \omega_{t_l} \rangle$. The second mapping is provided by

$$\chi_2 = A_2 \sigma(W_2 \chi_1 + b_2),$$

where $W_2 \in \mathbb{R}^{m_2 \times (d+q)}$, $b_2 \in \mathbb{R}^{m_2}$, and $A_2 \in \mathbb{R}^{dq \times m_2}$. We will similarly use the notation $\chi_2 = (\chi_2(1), \dots, \chi_2(q))$, where our goal is to have $\chi_2(l) \approx 2x\sigma(\langle \omega_i, \omega_{t_l} \rangle - 1/2)$. To implement the first mapping, we rely on the following lemma.

**Lemma 22.** *Let $\sigma$ be the ReLU activation. For any $\varepsilon > 0$ and positive integer $d_e$, there exists $m = \mathcal{O}(d_e^3(\log(d_e/\varepsilon)/\varepsilon)^2)$, $a \in \mathbb{R}^m$, $W \in \mathbb{R}^{m \times 2d_e}$, and $b \in \mathbb{R}^m$, such that*

$$\sup_{\omega_1, \omega_2 \in \mathbb{S}^{d_e - 1}} \left| \langle \omega_1, \omega_2 \rangle - a^\top \sigma \left( W \begin{pmatrix} \omega_1 \\ \omega_2 \end{pmatrix} + b \right) \right| \leq \varepsilon,$$

*and*

$$\|a\|_2 \leq \mathcal{O}\left( d_e^{5/2} (\log(d_e/\varepsilon)/\varepsilon)^{3/2}/\sqrt{m} \right), \quad \left\| W^\top \right\|_{1,\infty} \leq 1, \quad \|b\|_\infty \leq 1.$$

**Proof.** Consider the mapping $e_{1j}, e_{2j} \mapsto e_{1j}e_{2j}$. Note that when $|e_{1j}| \leq 1$ and $|e_{2j}| \leq 1$, this mapping is $\sqrt{2}$-Lipschitz, and the output is bounded between $[-1, 1]$. Then, by Lemma 42, for every $\varepsilon_j > 0$, there exists $m_j \leq \mathcal{O}((1/\varepsilon_j \log(1/\varepsilon_j))^2)$, $a_j \in \mathbb{R}^{m_j}$, $W_j \in \mathbb{R}^{m_j \times 2d_e}$, and $b_j \in \mathbb{R}^{m_j}$, such that

$$\sup_{|e_{1j}| \leq 1, |e_{2j}| \leq 1} \left| e_{1j}e_{2j} - \sum_{l=1}^{m} a_{jl}\sigma\left( \langle w_{jl}, (\omega_1^\top, \omega_2^\top)^\top \rangle + b_{jl} \right) \right| \leq \varepsilon_j,$$

$\|a_j\|_2 \leq \mathcal{O}\left( (\log(1/\varepsilon_j)/\varepsilon_j)^{3/2}/\sqrt{m_j} \right)$, $\|b_j\|_\infty \leq 1$, and $\|w_{jl}\|_1 \leq 1$. Specifically, the only non-zero coordinates of $w_{jl}$ are the $j$th and $d_e + j$th coordinates.

Let $\varepsilon_j = \varepsilon/d_e$ and $m = \sum_{j=1}^{d_e} m_j = \mathcal{O}(d_e^3(\log(d_e/\varepsilon)/\varepsilon)^2)$. Construct $a, b \in \mathbb{R}^m$ and $W \in \mathbb{R}^{m \times 2d_e}$ by concatenating $(a_j)$, $(b_j)$, and $(W_j)$ respectively. The resulting network satisfies

$$\sup_{\omega_1, \omega_2 \in \mathbb{S}^{d_e - 1}} \left| \langle \omega_1, \omega_2 \rangle - a^\top \sigma \left( W \begin{pmatrix} \omega_1 \\ \omega_2 \end{pmatrix} + b \right) \right| \leq \varepsilon,$$

while $\|a\|_2 \leq \mathcal{O}\left( d_e^{5/2}(\log(d_e/\varepsilon)/\varepsilon)^{3/2}/\sqrt{m} \right)$, $\|b\|_\infty \leq 1$, and $\left\| W^\top \right\|_{1,\infty} \leq 1$, completing the proof. $\square$

We can now specify $A_1, W_1$, and $b_1$ in our construction.

**Lemma 23.** *For any $\varepsilon > 0$, let $\bar{m}_1 = \mathcal{O}(d_e^3(\log(d_e/\varepsilon)/\varepsilon)^2)$ and $m_1 = 2d + q\bar{m}_1$. Then, there exist $A_1 \in \mathbb{R}^{(d+q) \times m_1}$, $W_1 \in \mathbb{R}^{m_1 \times (d_h + d + (q+1)d_e)}$, and $b_1 \in \mathbb{R}^{m_1}$, given by Equations (B.5) to (B.9), such that*

$$\chi_1^x = x, \quad |\chi_1^\omega(l) - \langle \omega_i, \omega_{t_l} \rangle| \leq \varepsilon,$$

*for all $h \in \mathbb{R}^{d_h}$, $x \in \mathbb{R}^d$, $\omega_i, (\omega_{t_j})_{j \in [q]} \in \mathbb{S}^{d_e - 1}$, and $l \in [q]$. Furthermore, we have the following guarantees*

$$\left\| W_1^\top \right\|_{1,\infty} \leq \mathcal{O}(1), \quad \|b_1\|_\infty \leq \mathcal{O}(1), \quad \left\| A_1^\top \right\|_{1,\infty} \leq \mathcal{O}(d_e^{5/2}(\log(d_e/\varepsilon)/\varepsilon)^{3/2}).$$

**Proof.** We define the decompositions

$$W_1 = \begin{pmatrix} W_{11} \\ W_{12} \end{pmatrix}, \quad b_1 = \begin{pmatrix} b_{11} \\ b_{12} \end{pmatrix}, \quad A_1 = \begin{pmatrix} A_{11} \\ A_{12} \end{pmatrix}, \tag{B.5}$$

where $\boldsymbol{W}_{11} \in \mathbb{R}^{2d \times (d_h + d + d_e)}$, $\boldsymbol{W}_{12} \in \mathbb{R}^{q\bar{m}_1 \times (d_h + d + d_e)}$, $\boldsymbol{b}_{11} \in \mathbb{R}^{2d}$, $\boldsymbol{b}_{12} \in \mathbb{R}^{q\bar{m}_1}$, $\boldsymbol{A}_{11} \in \mathbb{R}^{d \times m_1}$, and $\boldsymbol{A}_{12} \in \mathbb{R}^{q \times m_1}$. Let $\boldsymbol{v}_1, \ldots, \boldsymbol{v}_d$ denote the standard basis of $\mathbb{R}^d$, and notice that $\sigma(z) - \sigma(-z) = z$. Therefore, we can implement the identity part of the mapping by letting

$$
\boldsymbol{W}_{11} = \begin{pmatrix} \boldsymbol{0}_{d_h} & \boldsymbol{v}_1^\top & \boldsymbol{0}_{(q+1)d_e}^\top \\ \boldsymbol{0}_{d_h} & -\boldsymbol{v}_1^\top & \boldsymbol{0}_{(q+1)d_e}^\top \\ \vdots & \vdots & \\ \boldsymbol{0}_{d_h} & \boldsymbol{v}_d^\top & \boldsymbol{0}_{(q+1)d_e}^\top \\ \boldsymbol{0}_{d_h} & -\boldsymbol{v}_d^\top & \boldsymbol{0}_{(q+1)d_e}^\top \end{pmatrix}, \tag{B.6}
$$

as well as

$$
\boldsymbol{b}_1 = \boldsymbol{0}_{2d}, \quad \text{and} \quad \boldsymbol{A}_{11} = \begin{pmatrix} 1 & -1 & 0 & 0 & \ldots & 0 & \boldsymbol{0}_{q\bar{m}_1}^\top \\ 0 & 0 & 1 & -1 & \ldots & 0 & \boldsymbol{0}_{q\bar{m}_1}^\top \\ \vdots & \vdots & \vdots & \vdots & \vdots & \vdots & \vdots \\ 0 & \ldots & 0 & 0 & 1 & -1 & \boldsymbol{0}_{q\bar{m}_1}^\top \end{pmatrix} \tag{B.7}
$$

Notice that $\left\| \boldsymbol{W}_{11}^\top \right\|_{1,\infty} = 1$ and $\left\| \boldsymbol{A}_{11}^\top \right\|_{1,\infty} = 2$. To implement the inner product part of the mapping, we take the construction of weights, biases, and second layer weights from Lemma 22, and rename them as $\tilde{\boldsymbol{W}}_1 \in \mathbb{R}^{\bar{m}_1 \times 2d_e}$, $\tilde{\boldsymbol{b}}_1 \in \mathbb{R}^{\bar{m}_1}$, and $\tilde{\boldsymbol{a}}_1 \in \mathbb{R}^{\bar{m}_1}$. Let us introduce the decomposition $\tilde{\boldsymbol{W}}_1 = \begin{pmatrix} \tilde{\boldsymbol{W}}_{11} & \tilde{\boldsymbol{W}}_{12} \end{pmatrix}$, where $\tilde{\boldsymbol{W}}_{11}, \tilde{\boldsymbol{W}}_{12} \in \mathbb{R}^{\bar{m}_1 \times d_e}$. With this decomposition, we can separate the projections applied to the first and second vectors in Lemma 22. We can then define

$$
\boldsymbol{W}_{12} = \begin{pmatrix} \boldsymbol{0}_{\bar{m}_1 \times (d_h+d)} & \tilde{\boldsymbol{W}}_{11} & \tilde{\boldsymbol{W}}_{12} & \boldsymbol{0}_{\bar{m}_1 \times d_e} & \cdots & \boldsymbol{0}_{\bar{m}_1 \times d_e} \\ \boldsymbol{0}_{\bar{m}_1 \times (d_h+d)} & \tilde{\boldsymbol{W}}_{11} & \boldsymbol{0}_{\bar{m}_1 \times d_e} & \tilde{\boldsymbol{W}}_{12} & \cdots & \boldsymbol{0}_{\bar{m}_1 \times d_e} \\ \vdots & \vdots & \vdots & \vdots & \vdots & \vdots \\ \boldsymbol{0}_{\bar{m}_1 \times (d_h+d)} & \tilde{\boldsymbol{W}}_{11} & \boldsymbol{0}_{\bar{m}_1 \times d_e} & \boldsymbol{0}_{\bar{m}_1 \times d_e} & \cdots & \tilde{\boldsymbol{W}}_{12} \end{pmatrix}, \tag{B.8}
$$

as well as

$$
\boldsymbol{b}_{12} = \begin{pmatrix} \tilde{\boldsymbol{b}}_1 \\ \vdots \\ \tilde{\boldsymbol{b}}_1 \end{pmatrix}, \quad \text{and} \quad \boldsymbol{A}_{12} = \begin{pmatrix} \boldsymbol{0}_{2d}^\top & \tilde{\boldsymbol{a}}_1^\top & \boldsymbol{0}_{\bar{m}_1}^\top & \cdots & \boldsymbol{0}_{\bar{m}_1}^\top \\ \boldsymbol{0}_{2d}^\top & \boldsymbol{0}_{\bar{m}_1}^\top & \tilde{\boldsymbol{a}}_1^\top & \cdots & \boldsymbol{0}_{\bar{m}_1}^\top \\ \vdots & \vdots & \vdots & \vdots & \vdots \\ \boldsymbol{0}_{2d}^\top & \boldsymbol{0}_{\bar{m}_1}^\top & \cdots & \boldsymbol{0}_{\bar{m}_1}^\top & \tilde{\boldsymbol{a}}_1^\top \end{pmatrix}. \tag{B.9}
$$

From Lemma 22, we have $\left\| \boldsymbol{W}_{12}^\top \right\|_{1,\infty} \leq 1$, $\|\boldsymbol{b}_{12}\|_\infty \leq 1$, and

$$
\left\| \boldsymbol{A}_{12}^\top \right\|_{1,\infty} = \|\tilde{\boldsymbol{a}}_1\|_1 \leq \mathcal{O}(d_e^{5/2}(\log(d_e/\varepsilon)/\varepsilon)^{3/2}),
$$

which completes the proof. $\qquad\square$

To introduce the construction of the next layer, we rely on the following lemma which establishes the desired approximation for a single coordinate, the proof of which is similar to that of Lemma 22.

**Lemma 24.** *Let $\sigma$ be the ReLU activation. Suppose $|h| \leq r_\infty^h$, $|x| \leq r_\infty^x$ and $|z| \leq 1$. Let $R := \sqrt{1 + r_\infty^x{}^2 + r_\infty^h{}^2}$. For any $\varepsilon > 0$, there exists $m = \mathcal{O}(R^6 (\log(R/\varepsilon)/\varepsilon)^3)$, $\boldsymbol{a} \in \mathbb{R}^m$, $\boldsymbol{W} \in \mathbb{R}^{m \times 2}$, and $\boldsymbol{b} \in \mathbb{R}^m$, such that*

$$
\sup_{|h| \leq r_\infty^h, |x| \leq r_\infty^x, |z| \leq 1} \left| h + 2x\sigma(z - 1/2) - \boldsymbol{a}^\top \sigma\big(\boldsymbol{W}(h, x, z)^\top + \boldsymbol{b}\big) \right| \leq \varepsilon
$$

*and*

$$
\|\boldsymbol{a}\|_2 \leq \mathcal{O}\big(R^6 (\log(R/\varepsilon)/\varepsilon)^2/\sqrt{m}\big), \quad \left\| \boldsymbol{W}^\top \right\|_{1,\infty} \leq R^{-1}, \quad \|\boldsymbol{b}\|_\infty \leq 1.
$$

*Additionally, if $r_\infty^h = 0$, we have the improved bounds*

$$
m = \mathcal{O}\big(R^4 (\log(R/\varepsilon)/\varepsilon)^2\big), \quad \|\boldsymbol{a}\|_2 \leq \mathcal{O}\big(R^5 (\log(R/\varepsilon)/\varepsilon)^{3/2}/\sqrt{m}\big)
$$

**Proof.** Note that $(h, x, z) \mapsto h + 2x\sigma(z - 1/2)$ is $2R$-Lipschitz, and $|h + 2x\sigma(z - 1/2)| \leq R$. The proof follows from Lemma 42 with dimension 3 when $r_\infty^h \neq 0$ and dimension 2 otherwise. $\qquad \square$

With that, we can now construct the weights for the second mapping in the network.

**Lemma 25.** *Suppose* $\|\boldsymbol{\chi}_1^{\boldsymbol{x}}\|_\infty \leq r_x$ *and* $\max_l |\chi^{\boldsymbol{\omega}}(l)| \leq 1$. *Let* $R := \sqrt{1 + r_x^2}$. *Then, for every* $\varepsilon > 0$ *and absolute constant* $\delta \in (0, 1)$, *there exists* $\bar{m}_2 \leq \mathcal{O}(R^4 (\log(R/\varepsilon)/\varepsilon)^{3/2})$, $m_2 := qd\bar{m}_2$, *and* $\boldsymbol{A}_2 \in \mathbb{R}^{d_h \times m_2}$, $\boldsymbol{W}_2 \in \mathbb{R}^{m_2 \times (d+q)}$, *and* $\boldsymbol{b}_2 \in \mathbb{R}^{m_2}$ *given by Equations* (B.10) *and* (B.11) *such that*

$$\|\boldsymbol{\chi}_2(l) - 2\boldsymbol{\chi}_1^{\boldsymbol{x}}\sigma(\chi_1^{\boldsymbol{\omega}}(l) - 1/2)\|_\infty \leq \varepsilon,$$

*for all such* $\boldsymbol{\chi}_1$ *and* $l \in [q]$, *where we recall* $\boldsymbol{\chi}_2 = \boldsymbol{A}_2\sigma(\boldsymbol{W}_2\boldsymbol{\chi}_1 + \boldsymbol{b}_2)$. *Moreover, we have*

$$\left\|\boldsymbol{A}_2^\top\right\|_{1,\infty} \leq \mathcal{O}(R^4 (\log(R/\varepsilon)/\varepsilon)^{3/2}), \quad \left\|\boldsymbol{W}_2^\top\right\|_{1,\infty} \leq R^{-1}, \quad \|\boldsymbol{b}_2\|_\infty \leq 1.$$

**Proof.** Let $\tilde{\boldsymbol{W}} = \begin{pmatrix} \tilde{\boldsymbol{w}}_{21} & \tilde{\boldsymbol{w}}_{22} \end{pmatrix}$, $\tilde{\boldsymbol{b}}$, and $\tilde{\boldsymbol{a}}$ be the weights obtained from Lemma 24, where $\tilde{\boldsymbol{w}}_{21}, \tilde{\boldsymbol{w}}_{22}, \tilde{\boldsymbol{b}}, \tilde{\boldsymbol{a}} \in \mathbb{R}^{\bar{m}_2}$. To construct $\boldsymbol{W}_2$ and $\boldsymbol{b}_2$, we let

$$\boldsymbol{W}_2 = \begin{pmatrix} \boldsymbol{W}_2(1, 1) \\ \vdots \\ \boldsymbol{W}_2(1, d) \\ \vdots \\ \boldsymbol{W}_2(q, 1) \\ \vdots \\ \boldsymbol{W}_2(q, d) \end{pmatrix}, \quad \boldsymbol{b}_{22} = \begin{pmatrix} \boldsymbol{b}_2(1, 1) \\ \vdots \\ \boldsymbol{b}_2(1, d) \\ \vdots \\ \boldsymbol{b}_2(q, 1) \\ \vdots \\ \boldsymbol{b}_2(q, d) \end{pmatrix}. \tag{B.10}$$

where $\boldsymbol{W}_2(l, j) \in \mathbb{R}^{\bar{m}_2 \times (d+q)}$ is given by

$$\boldsymbol{W}_2(l, j) = \begin{pmatrix} \boldsymbol{0}_{\bar{m}_2 \times (j-1)} & \tilde{\boldsymbol{w}}_{21} & \boldsymbol{0}_{\bar{m}_2 \times (d-j)} & \boldsymbol{0}_{\bar{m}_2 \times (l-1)} & \tilde{\boldsymbol{w}}_{22} & \boldsymbol{0}_{\bar{m}_2 \times (q-l)} \end{pmatrix},$$

and $\boldsymbol{b}_2(l, j) = \tilde{\boldsymbol{b}}$. Consequently, $\left\|\boldsymbol{W}_2^\top\right\|_{1,\infty} \leq 1$ and $\|\boldsymbol{b}_2\|_\infty \leq 1$. Finally, we have

$$\boldsymbol{A}_2 = \begin{pmatrix} \tilde{\boldsymbol{a}}_2^\top & \boldsymbol{0}_{\bar{m}_2}^\top & \cdots & \boldsymbol{0}_{\bar{m}_2}^\top \\ \boldsymbol{0}_{\bar{m}_2}^\top & \tilde{\boldsymbol{a}}_2^\top & \cdots & \boldsymbol{0}_{\bar{m}_2}^\top \\ \vdots & \vdots & \vdots & \vdots \\ \boldsymbol{0}_{\bar{m}_2}^\top & \cdots & \boldsymbol{0}_{\bar{m}_2}^\top & \tilde{\boldsymbol{a}}_2^\top \end{pmatrix}. \tag{B.11}$$

Consequently, we obtain $\left\|\boldsymbol{A}_2^\top\right\|_{1,\infty} \leq \mathcal{O}(R^4 (\log(R/\varepsilon)/\varepsilon)^{3/2})$, completing the proof. $\qquad \square$

We are now ready to provide the four-layer feedforward construction of $f^\rightarrow(\boldsymbol{h}, \boldsymbol{x}, \boldsymbol{t}; \boldsymbol{\Theta}_h^\rightarrow)$.

**Proposition 26.** *Let* $\boldsymbol{z} = (\boldsymbol{x}, \boldsymbol{\omega}_i, \boldsymbol{\omega}_{t_1}, \ldots, \boldsymbol{\omega}_{t_q})$. *Then, for every* $\varepsilon > 0$, *there exists a feedforward network with* $L_h = 4$ *layers given by*

$$f^\rightarrow(\boldsymbol{h}, \boldsymbol{z}; \boldsymbol{\Theta}_h^\rightarrow) = \boldsymbol{W}_{L_h}\sigma\Big(\ldots\sigma\big(\boldsymbol{W}_2\sigma\big(\boldsymbol{W}_1(\boldsymbol{h}^\top, \boldsymbol{z}^\top)^\top + \boldsymbol{b}_1\big) + \boldsymbol{b}_2\big)\ldots\Big)$$

*where* $\boldsymbol{W}_i \in \mathbb{R}^{m_i \times m_{i-1}}, \boldsymbol{b}_i \in \mathbb{R}_i^m$ *for* $i \in \{2, \ldots, L_h - 1\}$, $\boldsymbol{W}_1 \in \mathbb{R}^{m_1 \times d_h + d + (q+1)d_e}, \boldsymbol{b}_1 \in \mathbb{R}^{m_1}$, *and* $\boldsymbol{W}_{L_h} \in \mathbb{R}^{d_h \times m_{L_h - 1}}$ *that satisfies the following:*

1. *If* $t_l = i$, *then*

$$\left\|f^\rightarrow(\boldsymbol{h}, \boldsymbol{z}; \hat{\boldsymbol{\Theta}}_h^\rightarrow)_l - \boldsymbol{x}\right\|_2 \leq \varepsilon$$

2. *Else* $f^\rightarrow(\boldsymbol{h}, \boldsymbol{z}; \hat{\boldsymbol{\Theta}}^\rightarrow)_l = \boldsymbol{0}_d$,

*for all* $l \in [q]$, $\boldsymbol{h} \in \mathbb{R}^{d_h}$ *and* $\|\boldsymbol{x}\|_2 \leq r_x$. *Additionally* $\|\boldsymbol{W}_i\|_F \leq \text{poly}(r_x, D_e, \varepsilon^{-1})$ *for all* $i \in [L_h]$ *and* $m_i, \|\boldsymbol{b}_i\|_2 \leq \text{poly}(r_x, D_e, \varepsilon^{-1})$ *for all* $i \in [L_h - 1]$, *where we recall* $D_e = d + (q+1)d_e$.

**Proof.** Let $\tilde{A}_1 \in \mathbb{R}^{(d+q)\times m_1}, \tilde{W}_1 \in \mathbb{R}^{m_1 \times (d_h + d + (q+1)d_e)}, \tilde{b}_1 \in \mathbb{R}^{m_1}$ be given by Lemma 23 with error parameter $\varepsilon_1$ and $\tilde{A}_2 \in \mathbb{R}^{d_h \times m_2}, \tilde{W}_2 \in \mathbb{R}^{m_2 \times (d+q)}, \tilde{b}_2 \in \mathbb{R}^{m_2}$ be given by Lemma 25 with error parameter $\varepsilon_2$. Recall that

$$\boldsymbol{\chi}_1 = \tilde{A}_1 \sigma\big(\tilde{W}_1 \boldsymbol{\chi}_0 + \tilde{b}_1\big), \quad \boldsymbol{\chi}_2 = \tilde{A}_2 \sigma\big(\tilde{W}_2 \boldsymbol{\chi}_1 + \tilde{b}_2\big).$$

By the triangle inequality,

$$
\begin{aligned}
\left\| \Psi(\boldsymbol{x}, \boldsymbol{t}, i) - \tilde{A}_2 \sigma\big(\tilde{W}_2 \boldsymbol{\chi}_1 + \tilde{b}_2\big) \right\|_\infty \leq & \left\| \Psi(\boldsymbol{x}, \boldsymbol{t}, i) - \tilde{A}_2 \sigma\big(\tilde{W}_2 \bar{\boldsymbol{\chi}}_1 + \tilde{b}_2\big) \right\|_\infty \\
& + \left\| \tilde{A}_2 \sigma\big(\tilde{W}_2 \bar{\boldsymbol{\chi}}_1 + \tilde{b}_2\big) - \tilde{A}_2 \sigma\big(\tilde{W}_2 \boldsymbol{\chi}_1 + \tilde{b}_2\big) \right\|_\infty \\
\leq & \varepsilon_2 + \left\| \tilde{A}_2^\top \right\|_{1,\infty} \left\| \tilde{W}_2 \right\|_{1,\infty} \| \boldsymbol{\chi}_1 - \bar{\boldsymbol{\chi}}_1 \|_\infty \\
\leq & \varepsilon_2 + \left\| \tilde{A}_2 \right\|_{1,\infty} \left\| \tilde{W}_2 \right\|_{1,\infty} \varepsilon_1,
\end{aligned}
$$

where $\bar{\boldsymbol{\chi}}_1 = (\boldsymbol{x}^\top, \langle \boldsymbol{\omega}_i, \boldsymbol{\omega}_{t_1} \rangle, \ldots, \langle \boldsymbol{\omega}_i, \boldsymbol{\omega}_{t_q} \rangle)^\top$. By letting $\varepsilon_2 = \varepsilon/4$, we obtain

$$m_2, \left\| \tilde{A}_2 \right\|_{\mathrm{F}}, \left\| \tilde{W}_2 \right\|_{\mathrm{F}}, \left\| \tilde{b}_2 \right\|_2 \leq \mathrm{poly}(r_x, D_e, \varepsilon^{-1}).$$

Similarly, we can let $\varepsilon_1 = \varepsilon / \big(4 \left\| \tilde{A}_2 \right\|_{1,\infty} \left\| \tilde{W}_2 \right\|_{1,\infty}\big)$, which yields

$$m_1, \left\| \tilde{A}_2 \right\|_{\mathrm{F}}, \left\| \tilde{W}_2 \right\|_{\mathrm{F}}, \left\| \tilde{b}_2 \right\|_2 \leq \mathrm{poly}(r_x, D_e, \varepsilon^{-1}).$$

Let

$$\boldsymbol{W}_2 = \tilde{W}_2 \tilde{A}_1, \qquad \boldsymbol{W}_1 = \tilde{W}_1, \qquad \boldsymbol{b}_1 = \tilde{b}_1, \qquad \boldsymbol{b}_2 = \tilde{b}_2.$$

Then,

$$\boldsymbol{\chi}_2 = \tilde{A}_2 \sigma(\boldsymbol{W}_2 \sigma(\boldsymbol{W}_1(\boldsymbol{h}^\top \boldsymbol{z}^\top)^\top + \boldsymbol{b}_1) + \boldsymbol{b}_2),$$

satisfies $\|\boldsymbol{\chi}_2 - \Psi(\boldsymbol{x}, \boldsymbol{t}, i)\|_\infty \leq \varepsilon/2$ for all $\|\boldsymbol{x}\|_2 \leq r_x$.

Recall that when $t_l \neq i$ for some $l \in [q]$, we would like to guarantee the output of the network to be equal to $\Psi(\boldsymbol{x}, \boldsymbol{t}, i)_l = \boldsymbol{0}_d$. To do so, we rely on the fact that $z \mapsto \sigma(z - b) - \sigma(-z - b)$ is zero for $|z| \leq b$, and has an $L_\infty$ distance of $b$ from the identity, i.e. $|z - \sigma(z - b) + \sigma(-z - b)| \leq b$. This mapping needs to be applied element-wise to $\boldsymbol{\chi}_2$. Let $\tilde{W}_3 \in \mathbb{R}^{2d_h \times d_h}, \boldsymbol{b}_3 \in \mathbb{R}^{2d_h}$, and $\boldsymbol{W}_4 \in \mathbb{R}^{d_h \times 2d_h}$ via

$$
\tilde{W}_3 = \begin{pmatrix} \boldsymbol{v}_1^\top \\ -\boldsymbol{v}_1^\top \\ \vdots \\ \boldsymbol{v}_d^\top \\ -\boldsymbol{v}_d^\top \end{pmatrix}, \quad \boldsymbol{b}_3 = -\frac{\varepsilon}{2}\boldsymbol{1}_{2d_h}, \quad \boldsymbol{W}_4 = \begin{pmatrix} 1 & -1 & 0 & 0 & \ldots & 0 & 0 \\ 0 & 0 & 1 & -1 & \ldots & 0 & 0 \\ 0 & 0 & 0 & 0 & \ldots & 1 & -1 \end{pmatrix}.
$$

As a result, $\boldsymbol{\chi}_3 = \boldsymbol{W}_4 \sigma(\tilde{W}_3 \boldsymbol{\chi}_2 + \boldsymbol{b}_3)$ satisfies

$$|(\chi_3)_j - (\chi_2)_j| \leq \begin{cases} 0 & |(\chi_2)_j| \leq \varepsilon/2 \\ \varepsilon/2 & |(\chi_2)_j| > \varepsilon/2 \end{cases}, \quad \forall j \in [d_h]. \tag{B.12}$$

We thus make two observations. First, $\|\boldsymbol{\chi}_3 - \boldsymbol{\chi}_2\|_\infty \leq \varepsilon/2$, and consequently $\|\boldsymbol{\chi}_3(l) - \Psi(\boldsymbol{x}, \boldsymbol{t}, i)_l\|_\infty \leq \varepsilon$ for all $l \in [q]$. Second, when $t_l \neq i$, we have $\Psi(\boldsymbol{x}, \boldsymbol{t}, i)_l = \boldsymbol{0}_d$ and $|\chi_2(l)_j| \leq \varepsilon/2$ for all $j \in [d]$ since $\|\boldsymbol{\chi}_2(l) - \Psi(\boldsymbol{x}, \boldsymbol{t}, i)_l\|_\infty \leq \varepsilon/2$. Consequently, by the first case in (B.12), we have $\chi_3(l)_j = 0$ for all $j \in [d]$. We can summarize these two observations as follows

$$\|\boldsymbol{\chi}_3(l) - \Psi(\boldsymbol{x}, \boldsymbol{t}, i)_l\|_\infty \leq \begin{cases} 0 & t_l \neq i \\ \varepsilon & t_l = i \end{cases},$$

which completes the proof. $\qquad\square$

With the above implementation of $f^{\rightarrow}(\boldsymbol{h}, \boldsymbol{z}; \boldsymbol{\Theta}_h^{\rightarrow})$, we have the following guarantee on $\boldsymbol{h}_i^{\rightarrow}$ for all $i \in [N]$.

**Corollary 27.** *Let $f_h^{\rightarrow}$ be given by the construction in Proposition 26, and suppose $r_h \geq \sqrt{q}(r_x + \sqrt{d}\varepsilon)$. Then, $h_i^{\rightarrow}$ satisfies the following guarantees for all $i \in [N]$ and $l \in [q]$:*

1. *If $t_l \geq i$, then $h_i^{\rightarrow}(l) = \mathbf{0}_d$*

2. *If $t_l < i$, then $\|h_i^{\rightarrow}(l) - x_{t_l}\|_\infty \leq \varepsilon$.*

**Proof.** We can prove the statement by induction. Note that it holds for $i = 1$ since $h_1^{\rightarrow} = \mathbf{0}_d$. For the induction step, suppose it holds up to some $i$, and recall

$$h_{i+1}^{\rightarrow} = h_i^{\rightarrow} + f_h^{\rightarrow}(h_i^{\rightarrow}, z_i; \Theta_h^{\rightarrow}).$$

- If $t_l \geq i + 1$, then $h_i^{\rightarrow}(l) = \mathbf{0}_d$ and $f_h^{\rightarrow}(h_j^{\rightarrow}, z_i; \Theta_h^{\rightarrow}) = \mathbf{0}_d$ by Proposition 26.
- If $t_l < i < i + 1$, then $\|h_i^{\rightarrow}(l) - x_{t_l}\|_\infty \leq \varepsilon$ by induction hypothesis, and $f_h^{\rightarrow}(h_j^{\rightarrow}, z_j; \Theta_h^{\rightarrow}) = \mathbf{0}_d$.
- Finally, if $t_l = i < i + 1$, then $h_i^{\rightarrow}(l) = 0$ and $\|f_h^{\rightarrow}(h_i^{\rightarrow}, z_i; \Theta_h^{\rightarrow}) - x_{t_l}\|_\infty \leq \varepsilon$.

Note that since $\|h_j^{\rightarrow}\|_2 \leq r_h$ for all $j \in [N]$, the projection $\Pi_{r_h}$ will always be identity through the forward pass, concluding the proof. $\square$

By symmetry, the same construction for $f_h^{\leftarrow}$ would yield a similar guarantee on $h_j^{\leftarrow}$.

The last step is to design $f_y(h^{\rightarrow}, h^{\leftarrow}, z; \Theta_y)$ such that

$$f_y(h^{\rightarrow}, h^{\leftarrow}, z_i; \Theta_y) \approx g\big(h^{\rightarrow} + h^{\leftarrow} + (x_i^\top \mathbb{1}[t_1 = i], \ldots, x_i^\top \mathbb{1}[t_q = i])^\top\big).$$

The following proposition provides the end-to-end RNN guarantee for approximating simple $q$STR models.

**Proposition 28.** *Suppose $g$ satisfies Assumption 2. Then there exist RNN weights $\Theta_{\mathrm{RNN}}$ with $\mathrm{vec}(\Theta_{\mathrm{RNN}}) \in \mathbb{R}^p$ (i.e. with $p$ parameters) and $r_h \geq \sqrt{q}r_x + \sqrt{\varepsilon_{2\mathrm{NN}}}/(r_a r_w)$, such that*

$$\sup_{i \in [N]} \big|g(x_{t_1}, \ldots, x_{t_q}) - \hat{y}(p; \Theta_{\mathrm{RNN}})_i\big|^2 \leq 4\varepsilon_{2\mathrm{NN}} \tag{B.13}$$

*for all $t \in [N]^q$ and $\|x_j\|_2 \leq r_x$ for all $j \in [N]$. Additionally, we have*

$$\|\mathrm{vec}(\Theta_{\mathrm{RNN}})\|_2 \leq \mathrm{poly}(r_x, D_e, r_w, r_a, \varepsilon_{2\mathrm{NN}}^{-1}), \quad p \leq \mathrm{poly}(r_x, D_e, m_g, r_w, r_a, \varepsilon_{2\mathrm{NN}}^{-1}), \tag{B.14}$$

*and $f_h^{\rightarrow}, f_h^{\leftarrow}$ do not depend on $h^{\rightarrow}$ and $h^{\leftarrow}$, namely the first $d_h$ columns of $W_1^{\rightarrow}$ and $W_1^{\leftarrow}$ that are multiplied by $h^{\rightarrow}$ and $h^{\leftarrow}$ respectively are zero.*

**Proof.** As the proof of this proposition mostly follows from the previous proofs in this section, we only state the procedure for obtaining the desired weights.

Let $(v_j)_{j=1}^{d_h}$ denote the standard basis of $\mathbb{R}^{d_h}$. Since $\sigma(z) - \sigma(-z) = z$, we can implement the identity mapping in $\mathbb{R}^{d_h}$ via a two-layer feedforward network with the following weights

$$W_{\mathrm{id}} = \begin{pmatrix} v_1^\top \\ -v_1^\top \\ \vdots \\ v_{d_h}^\top \\ -v_{d_h}^\top \end{pmatrix}, \quad b_{\mathrm{id}} = \mathbf{0}_{2d_h}, \quad A_{\mathrm{id}} = \begin{pmatrix} 1 & -1 & 0 & 0 & \ldots & 0 & 0 \\ 0 & 0 & 1 & -1 & \ldots & 0 & 0 \\ 0 & 0 & 0 & 0 & \ldots & 1 & -1 \end{pmatrix},$$

where $W_{\mathrm{id}} \in \mathbb{R}^{2d_h \times d_h}$, $b_{\mathrm{id}} \in \mathbb{R}^{2d_h}$, and $A_{\mathrm{id}} \in \mathbb{R}^{d_h \times 2d_h}$. Let $W_1, b_1, \tilde{A}_1, \tilde{W}_2, b_2, \tilde{A}_2$ be given as in the proof of Proposition 26, for achieving an $L_\infty$ error of $\tilde{\varepsilon}$, to be fixed later. Recall $z_i = (x_i^\top, \omega_i^\top, \omega_{t_1}^\top, \ldots, \omega_{t_q}^\top)^\top$. In the following, we remove the zero columns of $W_1$ corresponding to the $h$ part of the input (see Lemma 23), which does not change the resulting function. Our construction can then be denoted by

$$
\begin{array}{llllllll}
h_i^{\rightarrow} & \xrightarrow{A_{\mathrm{id}}\sigma(W_{\mathrm{id}}\cdot)} & h_i^{\rightarrow} & \xrightarrow{A_{\mathrm{id}}\sigma(W_{\mathrm{id}}\cdot)} & h_i^{\rightarrow} & \searrow & & \\
h_i^{\leftarrow} & \xrightarrow{A_{\mathrm{id}}\sigma(W_{\mathrm{id}}\cdot)} & h_i^{\leftarrow} & \xrightarrow{A_{\mathrm{id}}\sigma(W_{\mathrm{id}}\cdot)} & h_i^{\leftarrow} & \rightarrow & h_i^{\rightarrow} + h_i^{\leftarrow} + \chi_2 & \xrightarrow{a_g^\top \sigma(W_g\cdot + b_g)} \hat{y}_{\mathrm{RNN}}(p; \Theta_{\mathrm{RNN}})_i \\
z_i & \xrightarrow{\tilde{A}_1\sigma(W_1\cdot + b_1)} & \chi_1 & \xrightarrow{\tilde{A}_2\sigma(\tilde{W}_2\cdot + b_2)} & \chi_2 & \nearrow & &
\end{array}
$$

Note that the addition above can be implemented exactly by using the fact that $\sigma(z_1 + z_2 + z_3) - \sigma(-z_1 - z_2 - z_3) = z_1 + z_2 + z_3$. Specifically, the weights of this layer are given by

$$\boldsymbol{W}_{\text{add}} = \begin{pmatrix} \boldsymbol{v}_1^\top & \boldsymbol{v}_1^\top & \boldsymbol{v}_1^\top \\ -\boldsymbol{v}_1^\top & -\boldsymbol{v}_1^\top & -\boldsymbol{v}_1^\top \\ \vdots & \vdots & \vdots \\ \boldsymbol{v}_{d_h}^\top & \boldsymbol{v}_{d_h}^\top & \boldsymbol{v}_{d_h}^\top \\ -\boldsymbol{v}_{d_h}^\top & -\boldsymbol{v}_{d_h}^\top & -\boldsymbol{v}_{d_h}^\top \end{pmatrix}, \quad \boldsymbol{b}_{\text{add}} = \boldsymbol{0}_{2d_h}, \quad \boldsymbol{A}_{\text{add}} = \boldsymbol{A}_{\text{id}},$$

where $\boldsymbol{W}_{\text{add}} \in \mathbb{R}^{2d_h \times 3d_h}$, $\boldsymbol{b}_{\text{add}} \in \mathbb{R}^{2d_h}$, $\boldsymbol{A}_{\text{add}} \in \mathbb{R}^{d_h \times 2d_h}$.

Let $\boldsymbol{\Theta}_h^{\rightarrow}$ (and similarly $\boldsymbol{\Theta}_h^{\leftarrow}$) be given by Proposition 26 with corresponding error $\varepsilon_h$. Using the shorthand notation $\boldsymbol{x_t} = (\boldsymbol{x}_{t_1}, \dots, \boldsymbol{x}_{t_q}) \in \mathbb{R}^{dq}$ and $\hat{\boldsymbol{x}}_t = \boldsymbol{h}_i^{\rightarrow} + \boldsymbol{h}_i^{\leftarrow} + \boldsymbol{\chi}_2$, we have

$$\|\boldsymbol{h}_i^{\rightarrow} + \boldsymbol{h}_i^{\leftarrow} + \boldsymbol{\chi}_2 - \hat{\boldsymbol{x}}_t\|_2 \le \left\| \boldsymbol{h}_i^{\rightarrow} - \sum_{j=1}^{i-1} \Psi(\boldsymbol{x}_j, \boldsymbol{t}, j) \right\|_2 + \left\| \boldsymbol{h}_i^{\leftarrow} - \sum_{j=N}^{i+1} \Psi(\boldsymbol{x}_j, \boldsymbol{t}, j) \right\|_2 + \|\boldsymbol{\chi}_2 - \Psi(\boldsymbol{x}_i, \boldsymbol{t}, i)\|_2$$

$$\le \sqrt{qd}(2\varepsilon_h + \tilde{\varepsilon}),$$

which holds for all input prompts $\boldsymbol{p}$ with $\|\boldsymbol{x}_j\|_2 \le r_x$ for all $j \in [N]$. Finally, we have

$$\sup_{\|\boldsymbol{x}_j\|_2 \le r_x, \forall j \in [N]} \left| g(\boldsymbol{x_t}) - \boldsymbol{a}_g^\top \sigma(\boldsymbol{W}_g \hat{\boldsymbol{x}}_t + \boldsymbol{b}_g) \right| \le \sup_{\|\boldsymbol{x}_j\|_2 \le r_x, \forall j \in [N]} \left| g(\boldsymbol{x_t}) - \boldsymbol{a}_g^\top \sigma(\boldsymbol{W}_g \boldsymbol{x_t} + \boldsymbol{b}_g) \right|$$

$$+ \sup_{\|\boldsymbol{x}_j\|_2 \le r_x, \forall j \in [N]} \left| \boldsymbol{a}_g^\top \sigma(\boldsymbol{W}_g \boldsymbol{x_t} + \boldsymbol{b}_g) - \boldsymbol{a}_g^\top \sigma(\boldsymbol{W}_g \hat{\boldsymbol{x}}_t + \boldsymbol{b}_g) \right|$$

$$\le \sqrt{\varepsilon_{\text{2NN}}} + r_a r_w \sqrt{qd}(2\varepsilon_h + \tilde{\varepsilon}).$$

Choosing $\varepsilon_h = \sqrt{\varepsilon_{\text{2NN}}}/(4\sqrt{qd}r_a r_w)$ and $\tilde{\varepsilon} = \sqrt{\varepsilon_{\text{2NN}}}/(2\sqrt{qd}r_a r_w)$, we obtain RNN weights that saitsfy $\|\text{vec}(\boldsymbol{\Theta}_{\text{RNN}})\|_2 \le \text{poly}(r_x, D_e, r_a, r_w, \varepsilon_{\text{2NN}}^{-1})$, completing the proof. $\square$

## B.4 Generalization Upper Bounds for RNNs

Recall the state transitions

$$\boldsymbol{h}_{j+1}^{\rightarrow} = \Pi_{r_h}\left( \boldsymbol{h}_j^{\rightarrow} + f_h^{\rightarrow}(\boldsymbol{h}_j^{\rightarrow}, \boldsymbol{z}_j; \boldsymbol{\Theta}_h^{\rightarrow}) \right)$$
$$\boldsymbol{h}_{j-1}^{\leftarrow} = \Pi_{r_h}\left( \boldsymbol{h}_j^{\leftarrow} + f^{\leftarrow}(\boldsymbol{h}^{\leftarrow}, \boldsymbol{z}_j; \boldsymbol{\Theta}^{\leftarrow}) \right).$$

We will use the notation $\boldsymbol{h}_j^{\rightarrow}(\boldsymbol{p}; \boldsymbol{\Theta}_h^{\rightarrow})$ and $\boldsymbol{h}_j^{\leftarrow}(\boldsymbol{p}; \boldsymbol{\Theta}_j^{\leftarrow})$ to highlight the dependence of the hidden states on the prompt $\boldsymbol{p}$ and parameters $\boldsymbol{\Theta}_h^{\rightarrow}$ and $\boldsymbol{\Theta}_h^{\leftarrow}$. We then define the prediction function as $F(\boldsymbol{p}; \boldsymbol{\Theta}_h^{\rightarrow}, \boldsymbol{\Theta}_h^{\leftarrow}, \boldsymbol{\Theta}_y)$ where

$$F(\boldsymbol{p}; \boldsymbol{\Theta}_h^{\rightarrow}, \boldsymbol{\Theta}_h^{\leftarrow}, \boldsymbol{\Theta}_y)_j = f_y(\boldsymbol{h}_j^{\rightarrow}(\boldsymbol{p}; \boldsymbol{\Theta}_h^{\rightarrow}), \boldsymbol{h}_j^{\leftarrow}(\boldsymbol{p}; \boldsymbol{\Theta}_h^{\leftarrow}), \boldsymbol{z}_j; \boldsymbol{\Theta}_y).$$

We can now define the function class

$$\mathcal{F}_{\text{RNN}} = \{\boldsymbol{p}, j \mapsto F(\boldsymbol{p}; \boldsymbol{\Theta}_h^{\rightarrow}, \boldsymbol{\Theta}_h^{\leftarrow}, \boldsymbol{\Theta}_y)_j : \boldsymbol{\Theta}_h^{\rightarrow}, \boldsymbol{\Theta}_h^{\leftarrow}, \boldsymbol{\Theta}_y \in \Theta_{\text{RNN}}\}.$$

We can then define our distance function by going over $\{\boldsymbol{p}, j \in S_n\}$,

$$d_\infty(F, \hat{F}) = \sup_{\boldsymbol{p}, j \in S_n} \left| F(\boldsymbol{p}; \boldsymbol{\Theta}_h^{\rightarrow}, \boldsymbol{\Theta}_h^{\leftarrow}, \boldsymbol{\Theta}_y)_j - F(\boldsymbol{p}; \hat{\boldsymbol{\Theta}}_h^{\rightarrow}, \hat{\boldsymbol{\Theta}}_h^{\leftarrow}, \boldsymbol{\Theta}_y)_j \right|.$$

We will further use the notation

$$f_y(\cdot; \boldsymbol{\Theta}_y) = \boldsymbol{W}_{L_y}^y \sigma\left( \boldsymbol{W}_{L_y-1}^y \dots \sigma(\boldsymbol{W}_1^1(\cdot) + \boldsymbol{b}_1^y) \dots + \boldsymbol{b}_{L_y-1}^y \right) \in \mathcal{F}_{\text{NN}, L_y}^y,$$

and

$$f_h^{\rightarrow}(\cdot; \boldsymbol{\Theta}_h^{\rightarrow}) = \boldsymbol{W}_{L_h}^{\rightarrow} \sigma\left( \boldsymbol{W}_{L_h-1}^{\rightarrow} \dots \sigma(\boldsymbol{W}_1^{\rightarrow}(\cdot) + \boldsymbol{b}_1^{\rightarrow}) \dots + \boldsymbol{b}_{L_h-1}^{\rightarrow} \right) \in \mathcal{F}_{\text{NN}, L_h}^{\rightarrow}.$$

We similarly define $\mathcal{F}_{\text{NN}, L_h}^{\leftarrow}$. The covering number of $\mathcal{F}_{\text{RNN}}$ can be related to that of $\mathcal{F}_{\text{NN}, L_y}^y, \mathcal{F}_{\text{NN}, L_h}^{\rightarrow}$, and $\mathcal{F}_{\text{NN}, L_y}^{\rightarrow}$, through the following lemma.

**Lemma 29.** *Suppose for every* $\Theta_h^{\rightarrow}, \Theta_h^{\leftarrow}, \Theta_y \in \Theta_{\mathrm{RNN}}$ *we have*

$$\left\|\boldsymbol{W}_{L_y}^y \ldots \boldsymbol{W}_1^y\right\|_{op} \le C_W^y, \quad \left\|\boldsymbol{W}_{L_h}^{\rightarrow}\right\|_{op} \ldots \left\|\boldsymbol{W}_{1,h}^{\rightarrow}\right\|_{op} \le \alpha_N, \quad \left\|\boldsymbol{W}_{L_h}^{\leftarrow}\right\|_{op} \ldots \left\|\boldsymbol{W}_{1,h}^{\leftarrow}\right\|_{op} \le \alpha_N,$$

*where* $\alpha_N \le N^{-1}$. *Then,*

$$\log \mathcal{C}(\mathcal{F}_{\mathrm{RNN}}, d_\infty, \epsilon) \le \log \mathcal{C}(\mathcal{F}_{\mathrm{NN}, L_y}^y, d_\infty, \epsilon/2) + \log \mathcal{C}\left(\mathcal{F}_{\mathrm{NN}, L_h}^{\rightarrow}, d_\infty, \frac{\epsilon}{4eC_w^y N}\right)$$
$$+ \log \mathcal{C}\left(\mathcal{F}_{\mathrm{NN}, L_h}^{\leftarrow}, d_\infty, \frac{\epsilon}{4eC_w^y N}\right)$$

**Proof.** Throughout the proof, we will use the shorthand notation $\boldsymbol{h}_j^{\rightarrow} = \boldsymbol{h}_j^{\rightarrow}(\boldsymbol{p}; \Theta_h^{\rightarrow})$ and $\hat{\boldsymbol{h}}_j^{\rightarrow} = \boldsymbol{h}_j^{\rightarrow}(\boldsymbol{p}; \hat{\Theta}_h^{\rightarrow})$, with similarly define $\boldsymbol{h}_j^{\leftarrow}$ and $\hat{\boldsymbol{h}}_j^{\leftarrow}$. We begin by observing

$$\sup_{\boldsymbol{p}, j \in S_n} \left| f_y(\boldsymbol{h}_j^{\rightarrow}, \boldsymbol{h}_j^{\leftarrow}, \boldsymbol{z}_j; \Theta_y) - f_y(\hat{\boldsymbol{h}}_j^{\rightarrow}, \hat{\boldsymbol{h}}_j^{\leftarrow}, \boldsymbol{z}_j; \hat{\Theta}_y) \right| \le \mathcal{E}_1 + \mathcal{E}_2$$

where

$$\mathcal{E}_1 := \sup_{\boldsymbol{p}, j \in S_n} \left| f_y(\boldsymbol{h}_j^{\rightarrow}, \boldsymbol{h}_j^{\leftarrow}, \boldsymbol{z}_j; \Theta_y) - f_y(\boldsymbol{h}_j^{\rightarrow}, \boldsymbol{h}_j^{\leftarrow}, \boldsymbol{z}_j; \hat{\Theta}_y) \right|$$
$$\mathcal{E}_2 := \sup_{\boldsymbol{p}, j \in S_n} \left| f_y(\boldsymbol{h}_j^{\rightarrow}, \boldsymbol{h}_j^{\leftarrow}, \boldsymbol{z}_j; \hat{\Theta}_y) - f_y(\hat{\boldsymbol{h}}_j^{\rightarrow}, \hat{\boldsymbol{h}}_j^{\leftarrow}, \boldsymbol{z}_j; \hat{\Theta}_y) \right|.$$

Then, we observe that $\mathcal{E}_1 = d_\infty(f_y(\cdot; \Theta_y), f_y(\cdot; \hat{\Theta}_y))$. Thus, we can ensure $\mathcal{E}_1 \le \epsilon/2$ with a covering $\{\hat{\Theta}_y\}$ of size $\mathcal{C}(\mathcal{F}_{\mathrm{NN}, L_y}^y, d_\infty, \epsilon/2)$. Hence, we move to $\mathcal{E}_2$.

Using the Lipschitzness of $f_y$, we obtain

$$\mathcal{E}_2 \le \left\|\boldsymbol{W}_{L_y}^y \ldots \boldsymbol{W}_1^y\right\|_{op} \left(\sup_{\boldsymbol{p}, j} \left\|\boldsymbol{h}_j^{\rightarrow} - \hat{\boldsymbol{h}}_j^{\rightarrow}\right\|_2 + \sup_{\boldsymbol{p}, j} \left\|\boldsymbol{h}_j^{\leftarrow} - \hat{\boldsymbol{h}}_j^{\leftarrow}\right\|_2\right)$$
$$\le C_W^y \left(\sup_{\boldsymbol{p}, j} \left\|\boldsymbol{h}_j^{\rightarrow} - \hat{\boldsymbol{h}}_j^{\rightarrow}\right\|_2 + \sup_{\boldsymbol{p}, j} \left\|\boldsymbol{h}_j^{\leftarrow} - \hat{\boldsymbol{h}}_j^{\leftarrow}\right\|_2\right).$$

Further, by Lipschitzness of $\Pi_{r_h}$, we have

$$\sup_{\boldsymbol{p}, j} \left\|\boldsymbol{h}_j^{\rightarrow} - \hat{\boldsymbol{h}}_j^{\rightarrow}\right\|_2 \le \sup_{\boldsymbol{p}, j} \left\|\boldsymbol{h}_{j-1}^{\rightarrow} - \hat{\boldsymbol{h}}_{j-1}^{\rightarrow}\right\|_2 + \underbrace{\sup_{\boldsymbol{p}, j} \left\|f_h^{\rightarrow}(\boldsymbol{h}_{j-1}^{\rightarrow}, \boldsymbol{z}_{j-1}; \hat{\Theta}_h^{\rightarrow}) - f_h^{\rightarrow}(\hat{\boldsymbol{h}}_{j-1}^{\rightarrow}, \boldsymbol{z}_{j-1}; \hat{\Theta}_h^{\rightarrow})\right\|_2}_{=:\mathcal{E}_1^h}$$
$$+ \underbrace{\sup_{\boldsymbol{p}, j} \left\|f_h^{\rightarrow}(\boldsymbol{h}_{j-1}^{\rightarrow}, \boldsymbol{z}_{j-1}; \Theta_h^{\rightarrow}) - f_h^{\rightarrow}(\boldsymbol{h}_{j-1}^{\rightarrow}, \boldsymbol{z}_{j-1}; \hat{\Theta}_h^{\rightarrow})\right\|_2}_{=:\mathcal{E}_2^h}.$$

By the Lipschitzness of $f_h^{\rightarrow}$, for the second term we have

$$\mathcal{E}_1^h \le \left\|\hat{\boldsymbol{W}}_{L_h}^{\rightarrow} \ldots \hat{\boldsymbol{W}}_{1,h}^{\rightarrow}\right\|_{op} \left\|\boldsymbol{h}_{j-1}^{\rightarrow} - \hat{\boldsymbol{h}}_{j-1}^{\rightarrow}\right\|_2 \le \alpha_N \left\|\boldsymbol{h}_{j-1}^{\rightarrow} - \hat{\boldsymbol{h}}_{j-1}^{\rightarrow}\right\|_2.$$

Moreover, we have $\mathcal{E}_2^h \le d_\infty(f_h^{\rightarrow}(\cdot; \Theta_h^{\rightarrow}), f_h^{\rightarrow}(\cdot; \hat{\Theta}_h^{\rightarrow}))$. Consequently, we obtain

$$\sup_{\boldsymbol{p}, j} \left\|\boldsymbol{h}_j^{\rightarrow} - \hat{\boldsymbol{h}}_j^{\rightarrow}\right\|_2 \le (1 + \alpha_N) \sup_{\boldsymbol{p}, j} \left\|\boldsymbol{h}_{j-1}^{\rightarrow} - \hat{\boldsymbol{h}}_{j-1}^{\rightarrow}\right\|_2 + d_\infty(f_h^{\rightarrow}(\cdot; \Theta_h^{\rightarrow}), f_h^{\rightarrow}(\cdot; \hat{\Theta}_h^{\rightarrow}))$$
$$\le \sum_{l=0}^{j-2} (1 + \alpha_N)^l d_\infty(f_h^{\rightarrow}(\cdot; \Theta_h^{\rightarrow}), f^{\rightarrow}(\cdot; \hat{\Theta}_h^{\rightarrow}))$$
$$\le \frac{(1 + \alpha_N)^{j-1} - 1}{\alpha_N} d_\infty(f_h^{\rightarrow}(\cdot; \Theta_h^{\rightarrow}), f_h^{\rightarrow}(\cdot; \hat{\Theta}_h^{\rightarrow}))$$
$$\le eN d_\infty(f_h^{\rightarrow}(\cdot; \Theta_h^{\rightarrow}), f_h^{\rightarrow}(\cdot; \hat{\Theta}_h^{\rightarrow})).$$

We can similarly obtain an upper bound on $\sup_{\boldsymbol{p},j}\left\|\boldsymbol{h}_j^{\leftarrow} - \hat{\boldsymbol{h}}_j^{\leftarrow}\right\|_2$. Hence, we have

$$\mathcal{E}_2 \leq eC_w^y N\Big\{d_\infty(f_h^{\rightarrow}(\cdot;\boldsymbol{\Theta}_h^{\rightarrow}), f_h^{\rightarrow}(\cdot;\hat{\boldsymbol{\Theta}}_h^{\rightarrow})) + d_\infty(f_h^{\leftarrow}(\cdot;\boldsymbol{\Theta}_h^{\leftarrow}), f_h^{\leftarrow}(\cdot;\hat{\boldsymbol{\Theta}}_h^{\leftarrow}))\Big\}.$$

Therefore, by constructing $\epsilon/(2eC_w^y N)$ coverings $\{\hat{\boldsymbol{\Theta}}_h^{\rightarrow}\}$ and $\{\hat{\boldsymbol{\Theta}}_h^{\leftarrow}\}$ which have sizes

$$\mathcal{C}(\mathcal{F}_{\text{NN},L_h}^{\rightarrow}, \epsilon/(4eC_w^y N)), \quad \text{and,} \quad \mathcal{C}(\mathcal{F}_{\text{NN},L_h}^{\leftarrow}, \epsilon/(4eC_w^y N))$$

respectively, we complete the covering of $\mathcal{F}_{\text{RNN}}$. $\qquad\square$

The next step is to bound the covering number of the class of feedforward networks, as performed by the following lemma.

**Lemma 30.** *Let*

$$\mathcal{F}_{\text{NN},L} = \{\boldsymbol{x} \mapsto \boldsymbol{W}_L\sigma(\boldsymbol{W}_{L-1}\sigma(\dots \boldsymbol{W}_2(\sigma(\boldsymbol{W}_1\boldsymbol{x}+\boldsymbol{b}_1)\dots+\boldsymbol{b}_{L-1}) : \boldsymbol{\Theta}_{\text{NN}} \in \Theta_{\text{NN}}\},$$

*where $\boldsymbol{\Theta}_{\text{NN}} = (\boldsymbol{W}_1, \boldsymbol{b}_1, \dots, \boldsymbol{W}_{L-1}, \boldsymbol{b}_{L-1}, \boldsymbol{W}_L)$ and $\text{vec}(\boldsymbol{\Theta}_{\text{NN}}) \in \mathbb{R}^p$. Further, define the distance function*

$$d_\infty(f, f') = \sup_{\|\boldsymbol{x}\|\leq R} |f(\boldsymbol{x}) - f'(\boldsymbol{x})|, \quad \forall f, f' \in \mathcal{F}_{\text{NN},L}.$$

*Suppose $\|\boldsymbol{W}_l\|_F, \|\boldsymbol{b}_l\|_2 \leq R$ for all $l$. Then, for any absolute constant depth $L = \mathcal{O}(1)$, we have*

$$\log \mathcal{C}(\mathcal{F}_{\text{NN},L}, d_\infty, \epsilon) \leq p\log(1 + \text{poly}(R)/\epsilon).$$

**Proof.** Let $\boldsymbol{x}_0 = \boldsymbol{x}$, $\boldsymbol{x}_l = \sigma(\boldsymbol{W}_l\boldsymbol{x}_{l-1} + \boldsymbol{b}_l)$ for $l \in [L-1]$, and $\boldsymbol{x}_L = \boldsymbol{W}_L\boldsymbol{x}_{L-1}$. Also let $(\hat{\boldsymbol{x}}_l)$ be the corresponding definitions under weights and biases $(\hat{\boldsymbol{W}}_l)$ and $(\hat{\boldsymbol{b}}_l)$. First, we remark that for $l \in [L-1]$,

$$\|\boldsymbol{x}_l\|_2 \leq \|\boldsymbol{W}_l\|_{\text{op}}\|\boldsymbol{x}_{l-1}\|_2 + \|\boldsymbol{b}_l\|_2 \tag{B.15}$$
$$\leq \prod_{i=1}^l \|\boldsymbol{W}_i\|_{\text{op}}\|\boldsymbol{x}_0\|_2 + \sum_{i=0}^{l-1}\|\boldsymbol{b}_{l-i-1}\|_2\prod_{j=0}^i\|\boldsymbol{W}_{l-j}\|_{\text{op}} + \|\boldsymbol{b}_l\|_2$$
$$\leq \text{poly}(R), \tag{B.16}$$

where we used the fact that $L$ is an absolute constant. Next, for $l \in [L-1]$, we have

$$\|\boldsymbol{x}_l - \hat{\boldsymbol{x}}_l\|_2 \leq \left\|\boldsymbol{W}_l\boldsymbol{x}_{l-1} - \hat{\boldsymbol{W}}_l\hat{\boldsymbol{x}}_{l-1}\right\|_2 + \left\|\boldsymbol{b}_l - \hat{\boldsymbol{b}}_l\right\|_2$$
$$\leq \|\boldsymbol{W}_l\|_{\text{op}}\|\boldsymbol{x}_{l-1} - \hat{\boldsymbol{x}}_{l-1}\|_2 + \|\hat{\boldsymbol{x}}_{l-1}\|_2\left\|\boldsymbol{W}_l - \hat{\boldsymbol{W}}_l\right\|_{\text{op}} + \left\|\boldsymbol{b}_l - \hat{\boldsymbol{b}}_l\right\|_2$$
$$\leq \text{poly}(R)\left\{\|\boldsymbol{x}_{l-1} - \hat{\boldsymbol{x}}_{l-1}\|_2 + \left\|\boldsymbol{W}_l - \hat{\boldsymbol{W}}_l\right\|_F + \left\|\boldsymbol{b}_l - \hat{\boldsymbol{b}}_l\right\|_2\right\}.$$

Once again, using the fact that $L$ is an absolute constant and by expnaind the above inequality, we obtain

$$\|\boldsymbol{x}_l - \hat{\boldsymbol{x}}_l\|_2 \leq \text{poly}(R)\left\{\sum_{i=1}^l\left\|\boldsymbol{W}_i - \hat{\boldsymbol{W}}_i\right\|_F + \left\|\boldsymbol{b}_i - \hat{\boldsymbol{b}}_i\right\|_2\right\}.$$

Finally, we have the bound

$$\|\boldsymbol{x}_L - \hat{\boldsymbol{x}}_L\|_2 \leq \|\boldsymbol{W}_L\|_{\text{op}}\|\boldsymbol{x}_{L-1} - \hat{\boldsymbol{x}}_{L-1}\|_2 + \|\hat{\boldsymbol{x}}_{L-1}\|_2\left\|\boldsymbol{W}_L - \hat{\boldsymbol{W}}_L\right\|_{\text{op}}$$
$$\leq \text{poly}(R)\left\|\text{vec}(\boldsymbol{\Theta}_{\text{NN}}) - \text{vec}(\hat{\boldsymbol{\Theta}}_{\text{NN}})\right\|_2.$$

Consequently, we have

$$\log \mathcal{C}(\mathcal{F}_{\text{NN},L}, d_\infty, \epsilon) \leq \log \mathcal{C}(\{\boldsymbol{\Theta} \in \mathbb{R}^p : \|\boldsymbol{\Theta}\|_2 \leq \text{poly}(d,q)\}, \|\cdot\|_2, \epsilon/\text{poly}(R))$$
$$\leq p\log(1 + \text{poly}(R)/\epsilon),$$

where the last inequality follows from Lemma 41. $\qquad\square$

Therefore, we immediately obtain the following bound on the covering number of $\mathcal{F}_{\text{RNN}}$.

**Corollary 31.** *Suppose $\Theta_{\text{RNN}} \subseteq \{\Theta \in \mathbb{R}^p : \|\text{vec}(\Theta)\|_2 \le R\}$ and $\left\|z_j^{(i)}\right\|_2 \le R$ for all $i \in [n]$ and $j \in [N]$. Then,*

$$\log \mathcal{C}(\mathcal{F}_{\text{RNN}}, d_\infty, \epsilon) \le p \log(1 + \text{poly}(R)N/\epsilon).$$

We can now proceed with standard Rademacher complexity based arguments. Similar to the argument in Appendix A.2, we define a truncated version of the loss by considering the loss class

$$\mathcal{L}_\tau^{\text{RNN}} = \{(\boldsymbol{p}, \boldsymbol{y}, j) \mapsto (f_{\text{RNN}}(\boldsymbol{p})_j - y_j)^2 \wedge \tau \ : \ f_{\text{RNN}} \in \mathcal{F}_{\text{RNN}}\},$$

where the constant $\tau > 0$ will be chosen later. We then have the following bound on the empirical Rademacher complexity of $\mathcal{L}_\tau^{\text{RNN}}$.

**Lemma 32.** *In the same setting as Corollary 31 and with $\tau \ge 1$, we have*

$$\hat{\mathfrak{R}}_n(\mathcal{L}_\tau^{\text{RNN}}) \le \mathcal{O}\left(\tau\sqrt{\frac{p \log(RNn\tau)}{n}}\right).$$

**Proof.** By a standard discretization bound for Rademacher complexity, for all $\epsilon > 0$ we have

$$\hat{\mathfrak{R}}_n(\mathcal{L}_\tau^{\text{RNN}}) \le \epsilon + \tau\sqrt{\frac{2 \log \mathcal{C}(\mathcal{L}_\tau^{\text{RNN}}, d_\infty, \epsilon)}{n}}$$

$$\le \epsilon + \tau\sqrt{\frac{2 \log \mathcal{C}(\mathcal{F}_{\text{RNN}}, d_\infty, \epsilon/(2\sqrt{\tau}))}{n}}$$

$$\le \epsilon + \tau\sqrt{\frac{2p \log(1 + \text{poly}(R)N\sqrt{\tau}/\epsilon)}{n}},$$

where the second inequality follows from Lipschitzness of $(\cdot)^2 \wedge \tau$. We conclude the proof by choosing $\epsilon = 1/\sqrt{n}$. $\square$

We can directly turn the above bound on the empirical Rademacher complexity into a bound on generalization gap.

**Corollary 33.** *Let $\hat{\Theta} = \arg\min_{\Theta \in \Theta_{\text{RNN}}} \hat{R}_n^{\text{RNN}}(\Theta)$. Suppose $\Theta_{\text{RNN}} \subseteq \{\Theta \in \mathbb{R}^p : \|\text{vec}(\Theta)\|_2 \le R\}$, and additionally $\sqrt{3C_x ed \log(nN)} + q + 1 \le R$. Then, for every $\delta > 0$, with probability at least $1 - \delta - (nN)^{-1/2}$ over the training set, we have*

$$R_\tau^{\text{RNN}}(\hat{\Theta}) - \hat{R}_\tau^{\text{RNN}}(\hat{\Theta}) \le \mathcal{O}\left(\tau\sqrt{\frac{p \log(RNn\tau)}{n}} + \tau\sqrt{\frac{\log(1/\delta)}{n}}\right).$$

**Proof.** We highlight that for the specified $R$, Lemma 12 guarantees $\left\|z_j^{(i)}\right\|_2 \le R$ for all $i \in [n]$ and $j \in [N]$ with probability at least $1 - (nN)^{-1/2}$. Standard Rademacher complexity generalization arguments applied to Lemma 32 complete the proof. $\square$

Note that $\hat{R}_\tau^{\text{RNN}}(\hat{\Theta}) \le \hat{R}_n^{\text{RNN}}(\hat{\Theta})$ which is further controlled in the approximation section by Proposition 28. Therefore, the last step is to demonstrate that choosing $\tau = \text{poly}(d, q, \log n)$ suffices to achieve a desirable bound on $R^{\text{RNN}}(\hat{\Theta})$ through $R_\tau^{\text{RNN}}(\hat{\Theta})$.

**Lemma 34.** *Consider the setting of Corollary 33, and additionally assume $R \ge r_h$. Then, for some $\tau = \text{poly}(R, \log n)$, we have*

$$R^{\text{RNN}}(\hat{\Theta}) - R_\tau^{\text{RNN}}(\hat{\Theta}) \le \sqrt{\frac{1}{n}}.$$

.

**Proof.** The proof of this lemma proceeds similarly to the proof of Lemma 20. By defining

$$\Delta_y := \left|\hat{y}_{\text{RNN}}(\boldsymbol{p}; \hat{\Theta})_j - y_j\right|$$

and following the same steps (where we recall $j \sim \mathrm{Unif}([N])$), we obtain

$$R^{\mathrm{RNN}}(\hat{\boldsymbol{\Theta}}) = \mathbb{E}\big[\Delta_y^2 \mathbb{1}[\Delta_y \leq \sqrt{\tau}]\big] + \mathbb{E}\big[\Delta_y^2 \mathbb{1}[\Delta_y > \sqrt{\tau}]\big]$$
$$\leq R_\tau^{\mathrm{RNN}}(\hat{\boldsymbol{\Theta}}) + \mathbb{E}\big[\Delta_y^4\big]^{1/2}\mathbb{P}\big(\Delta_y \geq \sqrt{\tau}\big)^{1/2},$$

where

$$\mathbb{E}\big[\Delta_y^4\big]^{1/2} \leq 2\,\mathbb{E}\big[y_j^4\big]^{1/2} + 2\,\mathbb{E}\Big[\hat{y}_{\mathrm{RNN}}(\boldsymbol{p};\hat{\boldsymbol{\Theta}})_j^4\Big]^{1/2}$$

and

$$\mathbb{P}\big(\Delta_y > \sqrt{\tau}\big) \leq \mathbb{P}\bigg(|y_j| \geq \frac{\sqrt{\tau}}{2}\bigg) + \mathbb{P}\bigg(\big|\hat{y}_{\mathrm{RNN}}(\boldsymbol{p};\hat{\boldsymbol{\Theta}})_j\big| \geq \frac{\sqrt{\tau}}{2}\bigg)$$

From Assumption 1, we have $\mathbb{E}\big[y_j^4\big]^{1/2} \lesssim 1$ and $\mathbb{P}(|y_j| \geq \sqrt{\tau}/2) \leq e^{-\Omega(\tau^{1/s})}$. For the prediction of the RNN, we have the following bound (see (B.16) for the derivation)

$$\Big|\hat{y}_{\mathrm{RNN}}(\boldsymbol{p};\hat{\boldsymbol{\Theta}})_j\Big| \leq \prod_{l=1}^{L_y}\|\boldsymbol{W}_l^y\|_{\mathrm{op}}\big\|(\boldsymbol{h}_j^\rightarrow, \boldsymbol{h}_j^\leftarrow, \boldsymbol{z}_j)\big\|_2 + \sum_{i=0}^{L_y-1}\Big\|\boldsymbol{b}_{L_y-i-1}^y\Big\|_2 \prod_{l=0}^{i}\Big\|\boldsymbol{W}_{L_y-l}^y\Big\|_{\mathrm{op}}.$$

As a result,

$$\Big|\hat{y}_{\mathrm{RNN}}(\boldsymbol{p};\hat{\boldsymbol{\Theta}})_j\Big| \leq \mathrm{poly}(R)(1 + r_h + \|\boldsymbol{z}_j\|).$$

As a result, by the fact that $r_h \leq R$ and Assumption 1, after taking an expectation, we immediately have

$$\mathbb{E}\Big[\hat{y}_{\mathrm{RNN}}(\boldsymbol{p};\hat{\boldsymbol{\Theta}})_j^4\Big]^{1/2} \leq \mathrm{poly}(R).$$

On the other hand, from Lemma 12 (with $n = N = 1$), we obtain

$$\mathbb{P}\bigg(\big|\hat{y}_{\mathrm{RNN}}(\boldsymbol{p};\hat{\boldsymbol{\Theta}})\big| \geq \frac{\sqrt{\tau}}{2}\bigg) \leq e^{-\Omega(\tau/\mathrm{poly}(R))}$$

Therefore, for some $\tau = \mathrm{poly}(R, \log n)$ we can obtain the bound stated in the lemma. $\qquad\square$

We can summarize the above facts into the proof of Theorem 5.

**Proof of Theorem 5.** From the approximation bound of Proposition 28, we know that for some $R = \mathrm{poly}(d, q, r_a, r_w, \varepsilon_{\mathrm{2NN}}^{-1}, \log(nN))$ and the constraint set

$$\Theta_{\mathrm{RNN}} = \Big\{\boldsymbol{\Theta} \,:\, \|\mathrm{vec}(\boldsymbol{\Theta})\|_2 \leq R, \big\|\boldsymbol{W}_{\overrightarrow{L_h}}\big\|_{\mathrm{op}} \cdots \big\|\boldsymbol{W}_{\overrightarrow{1,h}}\big\|_{\mathrm{op}} \leq \alpha_N, \big\|\boldsymbol{W}_{\overleftarrow{L_h}}\big\|_{\mathrm{op}} \cdots \big\|\boldsymbol{W}_{\overleftarrow{1,h}}\big\|_{\mathrm{op}} \leq \alpha_N\Big\}$$

with any $\alpha_N \leq N^{-1}$, we have $\hat{R}^{\mathrm{RNN}}(\hat{\boldsymbol{\Theta}}) \lesssim \varepsilon_{\mathrm{2NN}}$. The proof is then completed by letting $r_h = \sqrt{q}r_x + \sqrt{\varepsilon_{\mathrm{2NN}}}/(r_a r_w)$, invoking the generalization bound of Corollary 33, and the bound on truncation error given in Lemma 34, with $R = \mathrm{poly}(d, q, r_a, r_w, \varepsilon_{\mathrm{2NN}}^{-1}, \log(nN))$.

$\qquad\square$

## B.5 Proof of Proposition 6

The crux of the proof of Proposition 6 is to show the following position, which provides a lower bound on the prediction error at any fixed position in the prompt.

**Proposition 35.** *Consider the same setting as in Proposition 6. There exists an absolute constant $c > 0$, such that for any fixed $j \in [N]$, if*

$$\mathbb{E}\big[(\hat{y}_{\mathrm{RNN}}(\boldsymbol{p})_j - y_j)^2\big] \leq c,$$

*then*

$$d_h \geq \Omega\bigg(\frac{N}{\log(1 + \mathfrak{L}^2\|\boldsymbol{U}\|_{op}^2)}\bigg), \quad and \quad \|\boldsymbol{U}\|_{op}^2 \geq \Omega\bigg(\frac{N}{\mathfrak{L}^2\log(1 + d_h)}\bigg).$$

We shortly remark that the statement of Proposition 6 directly follows from that of Proposition 35.

**Proof of Proposition 6.** Let $c$ be the constant given by Proposition 35. Suppose that

$$\frac{1}{N}\mathbb{E}\Big[\|\hat{\boldsymbol{y}}_{\text{RNN}}(\boldsymbol{p}) - \boldsymbol{y}\|_2^2\Big] \leq c.$$

Then,

$$\min_{j \in [N]}\mathbb{E}\big[(\hat{y}_{\text{RNN}}(\boldsymbol{p})_j - y_j)^2\big] \leq \frac{1}{N}\sum_{j=1}^{N}\mathbb{E}\big[(\hat{y}_{\text{RNN}}(\boldsymbol{p})_j - y_j)^2\big] \leq c.$$

As a result, there exists some $j \in [N]$ such that $\mathbb{E}\big[(\hat{y}_{\text{RNN}}(\boldsymbol{p})_j - y_j)^2\big] \leq c$. We can then invoke Proposition 35 to obtain lower bounds on $d_h$ and $\|\boldsymbol{U}\|_{\text{op}}$, completing the proof of Proposition 6. $\quad\square$

We now present the proof of Proposition 35.

**Proof of Proposition 35.** Let $\boldsymbol{h}_j = (\boldsymbol{U}^{\rightarrow}\boldsymbol{h}_j^{\rightarrow}, \boldsymbol{U}^{\leftarrow}\boldsymbol{h}_j^{\leftarrow}) \in \mathbb{R}^{2d_h}$, and define

$$\Phi(\boldsymbol{h}_j) := \Big(f_y(\boldsymbol{h}_j, \boldsymbol{x}_j, (1), j), \ldots, f_y(\boldsymbol{h}_j, \boldsymbol{x}_j, (j-1), j), f_y(\boldsymbol{h}_j, \boldsymbol{x}_j, (j+1), j), \ldots, f_y(\boldsymbol{h}_j, \boldsymbol{x}_j, (N), j)\Big)^{\top} \in \mathbb{R}^{N-1}.$$

In other words, $\Phi : \mathbb{R}^{2d_h} \to \mathbb{R}^{N-1}$ captures all possible outcomes of $\hat{y}_{\text{RNN}}(\boldsymbol{p})_j$ depending on the value of $t_j$ (excluding the case where $t_j = j$). Ideally, we must have $f_y(\boldsymbol{h}_j, \boldsymbol{x}_j, (k), j) \approx g(\boldsymbol{x}_k)$.

Let $\boldsymbol{p}^{(1)}, \ldots, \boldsymbol{p}^{(P)}$ be an i.i.d. sequence of prompts, then modify them to share the $j$th input token, i.e. $\boldsymbol{x}_j^{(i)} = \boldsymbol{x}_j^{(1)}$ for all $i \in [P]$, with $P$ to be determined later. Note that by our assumption on prompt distribution, this operation does not change the marginal distribution of each $\boldsymbol{p}^{(i)}$. Similarly, define

$$\boldsymbol{g}^{(i)} := (\boldsymbol{g}(\boldsymbol{x}_1^{(i)}), \ldots, \boldsymbol{g}(\boldsymbol{x}_{j-1}^{(i)}), \boldsymbol{g}(\boldsymbol{x}_{j+1}^{(i)}), \ldots, \boldsymbol{g}(\boldsymbol{x}_N^{(i)}))^{\top} \in \mathbb{R}^{N-1}$$

for each prompt. We also let $\boldsymbol{h}^{(i)\rightarrow}_j, \boldsymbol{h}^{(i)\leftarrow}_j$ be the corresponding hidden states obtained from passing these prompts through the RNN, and define $\boldsymbol{h}_j^{(i)}$ using them. Note that $\boldsymbol{g}^{(1)}, \ldots, \boldsymbol{g}^{(P)}$ is an i.i.d. sequence of vectors drawn from $\mathcal{N}(0, \mathbf{I}_{N-1})$.

We now define two events $E_1$ and $E_2$, where

$$E_1 = \Big\{\forall i \neq k, \quad \big\|\boldsymbol{g}^{(i)} - \boldsymbol{g}^{(k)}\big\|_2 \geq \varepsilon_g\sqrt{N-1}\Big\},$$

and

$$E_2 = \Bigg\{\sum_{i=1}^{P}\mathbb{1}\bigg[\big\|\Phi(\boldsymbol{h}_j^{(i)}) - \boldsymbol{g}^{(i)}\big\|_2 \geq \frac{\varepsilon\sqrt{N}}{\delta}\bigg] \leq 2\delta^2 P\Bigg\},$$

where $\delta \in (0, 1)$ will be chosen later. In other words, $E_1$ is the event in which $\boldsymbol{g}^{(i)}$ are "packed" in the space, while $E_2$ is the event where the RNN will be "wrong" at position $j$ on at most $2\delta^2$ fraction of the prompts. We will now attempt to lower bound $\mathbb{P}(E_1 \cap E_2)$.

Note that $\boldsymbol{g}^{(i)} - \boldsymbol{g}^{(k)} \overset{(d)}{=} \sqrt{2}\boldsymbol{g}$ where $\boldsymbol{g} \sim \mathcal{N}(0, \mathbf{I}_{N-1})$. By a union bound we have

$$\begin{aligned}
\mathbb{P}\big(E_1^C\big) &\leq \sum_{i \neq k}\mathbb{P}\Big(\big\|\boldsymbol{g}^{(i)} - \boldsymbol{g}^{(k)}\big\|_2 \leq \varepsilon_g\sqrt{N-1}\Big)\\
&\leq P^2\mathbb{P}\Big(\sqrt{2}\|\boldsymbol{g}\|_2 \leq \varepsilon_g\sqrt{N-1}\Big)\\
&\leq P^2\mathbb{P}\bigg(\|\boldsymbol{g}\|_2 - \mathbb{E}[\|\boldsymbol{g}\|_2] \leq \big(\frac{\varepsilon_g}{\sqrt{2}} - c\big)\sqrt{N-1}\bigg)\\
&\leq P^2 e^{-(c-\varepsilon_g/\sqrt{2})^2(N-1)/2},
\end{aligned}$$

for all $\varepsilon_g \leq c\sqrt{2}$, where $c > 0$ is an absolute constant such that $c\sqrt{N-1} \leq \mathbb{E}[\|\boldsymbol{g}\|]$, and the last inequality holds by subGaussianity of the norm of a standard Gaussian random vector. From here on, we will choose $\varepsilon_g = c/\sqrt{2}$ (and simply denote $\varepsilon_g \asymp 1$), which implies $\mathbb{P}(E_1^C) \leq P^2 e^{-c^2(N-1)/8}$.

To lower bound $\mathbb{P}(E_2)$, consider a random prompt-label pair $\boldsymbol{p}, \boldsymbol{y}$ and the corresponding $\boldsymbol{g}$. Note that in the prompt $\boldsymbol{p}$, the index $t_j$ is drawn independently of the rest of $\boldsymbol{p}$, and has a uniform distribution

in $[N]$. Let $\boldsymbol{p}[t_j \mapsto k]$ denote a modification of $\boldsymbol{p}$ where we set $t_j$ equal to $k$, and let $\boldsymbol{y}[t_j \mapsto k]$ be the labels corresponding to this modified prompt. We then have

$$\frac{1}{N}\|\Phi(\boldsymbol{h}_j) - \boldsymbol{g}\|_2^2 = \frac{1}{N}\sum_{k \neq j}\left(\hat{y}_{\text{RNN}}(\boldsymbol{p}[t_j \mapsto k])_j - g(\boldsymbol{x}_k)\right)^2$$

$$\leq \frac{1}{N}\sum_{k=1}^{N}\left(\hat{y}_{\text{RNN}}(\boldsymbol{p}[t_j \mapsto k])_j - y(\boldsymbol{p}[t_j \mapsto k])_j\right)^2$$

$$= \mathbb{E}_{t_j}\left[(\hat{y}_{\text{RNN}}(\boldsymbol{p})_j - y_j)^2\right]$$

As a result, via a Markov inequality, we obtain

$$\mathbb{P}\left(\frac{1}{N}\|\Phi(\boldsymbol{h}_j) - \boldsymbol{g}\|_2^2 \geq \frac{\varepsilon^2}{\delta^2}\right) = \mathbb{P}\left(\mathbb{E}_{t_j}\left[(\hat{y}_{\text{RNN}}(\boldsymbol{p})_j - y_j)^2\right] \geq \frac{\varepsilon^2}{\delta^2}\right)$$

$$\leq \frac{\delta^2\, \mathbb{E}\left[(\hat{y}_{\text{RNN}}(\boldsymbol{p})_j - y_j)^2\right]}{\varepsilon^2}$$

$$\leq \delta^2.$$

Going back to our lower bound on $\mathbb{P}(E_2)$, define the Bernoulli random variable

$$z^{(i)} = \mathbb{1}\left[\left\|\Phi(\boldsymbol{h}_j^{(i)}) - \boldsymbol{g}^{(i)}\right\|_2 \geq \frac{\varepsilon\sqrt{N}}{\delta}\right].$$

Note that $(z^{(i)})$ are i.i.d. since $\boldsymbol{h}_j^{(i)}$ and $\boldsymbol{g}^{(i)}$ do not depend on $\boldsymbol{x}_j$. Then, by Hoeffding's inequality,

$$\mathbb{P}\left(E_2^C\right) = \mathbb{P}\left(\sum_{j=1}^{P} z^{(i)} \geq 2\delta^2 P\right) \leq e^{-2P\delta^4}.$$

We now have our desired lower bound on $\mathbb{P}(E_1 \cap E_2)$, given by

$$\mathbb{P}(E_1 \cap E_2) \geq 1 - \mathbb{P}\left(E_1^C\right) - \mathbb{P}\left(E_2^C\right) \geq 1 - e^{-2P\delta^4} - P^2 e^{-c^2(N-1)/8}.$$

Suppose $\delta \geq e^{-c'N}$ for some absolute constant $c' > 0$. Then, choosing $P = \lfloor e^{c''N}\rfloor$ for some absolute constant $c'' > 0$ would ensure $\mathbb{P}(E_1 \cap E_2) > 0$, and allows us to look at this intersection.

Let $\mathcal{I} = \{i : z^{(i)} = 0\}$. On $E_1$, and for $i, k \in \mathcal{I}$ with $i \neq k$ we have

$$\left\|\Phi(\boldsymbol{h}_j^{(i)}) - \Phi(\boldsymbol{h}_j^{(k)})\right\|_2 \geq \left\|\boldsymbol{g}^{(i)} - \boldsymbol{g}^{(k)}\right\|_2 - \left\|\Phi(\boldsymbol{h}_j^{(i)}) - \boldsymbol{g}^{(i)}\right\|_2 - \left\|\Phi(\boldsymbol{h}_j^{(k)}) - \boldsymbol{g}^{(k)}\right\|_2$$

$$\geq \varepsilon_g\sqrt{N-1} - \frac{2\varepsilon\sqrt{N}}{\delta} =: \mathfrak{L}\sqrt{N}\varepsilon_h.$$

Note that from the Lipschitzness of $f_y$, we have $\left\|\Phi(\boldsymbol{h}_j^{(i)}) - \Phi(\boldsymbol{h}_j^{(k)})\right\|_2 \leq \frac{\mathfrak{L}\sqrt{N}}{r_h}\left\|\boldsymbol{h}_j^{(i)} - \boldsymbol{h}_j^{(k)}\right\|_2$. As a result, the set $\left\{\boldsymbol{h}_j^{(i)} : i \in \mathcal{I}\right\}$ is an $r_h\varepsilon_h$-packing for $\{\boldsymbol{h} : \|\boldsymbol{h}\|_2 \leq \sqrt{2}\|\boldsymbol{U}\|_{\text{op}}r_h\}$. Using Lemma 41, the log packing number can be bounded by

$$\log\mathcal{I} \leq \left\{d_h\log\left(1 + \frac{2\sqrt{2}\|\boldsymbol{U}\|_{\text{op}}}{\varepsilon_h}\right)\right\} \wedge \left\{\frac{2\|\boldsymbol{U}\|_{\text{op}}^2}{\varepsilon_h^2}\left(1 + \log\left(1 + \frac{M\varepsilon_h^2}{2\|\boldsymbol{U}\|_{\text{op}}^2}\right)\right)\right\}.$$

On $E_1 \cap E_2$, we have $\mathcal{I} \geq (1 - 2\delta^2)P \geq (1 - 2\delta^2)e^{cN}$ for some absolute constant $c > 0$. Therefore,

$$\frac{\log(1 - 2\delta^2) + cN}{\log(1 + 2\sqrt{2}\|\boldsymbol{U}\|_{\text{op}}/\varepsilon_h)} \leq d_h,$$

and

$$\frac{\varepsilon_h^2\left(\log(1 - 2\delta^2) + cN\right)}{2 + 2\log(1 + d_h\varepsilon_h^2/(2\|\boldsymbol{U}\|_{\text{op}}^2))} \leq \|\boldsymbol{U}\|_{\text{op}}^2.$$

Choosing $\delta = 1/2$ and recalling $\varepsilon_g \asymp 1$, we obtain $\varepsilon_h \gtrsim (1 - C\varepsilon)/\mathfrak{L}$ for some absolute constant $C > 0$, which concludes the proof. $\qquad\square$

## B.6 Proof of Theorem 7

We first provide an estimate for the capacity of two-layer feedforward networks to interpolate $n$ samples.

**Lemma 36.** *Suppose $\{\boldsymbol{x}^{(i)}\}_{i=1}^n \overset{\text{i.i.d.}}{\sim} \mathcal{N}(0, \mathbf{I}_d)$ and let $y^{(i)} = \langle \boldsymbol{u}, \boldsymbol{x}_{t_i} \rangle$ for arbitrary $t_i \in [N]$ and $\boldsymbol{u} \in \mathbb{S}^{d-1}$. Then, there exists an absolute constant $c > 0$ such that for all $m \geq n$ and with probability at least $c$, there exist data dependent weights $\boldsymbol{a}, \boldsymbol{b} \in \mathbb{R}^m$ and $\boldsymbol{W} \in \mathbb{R}^{m \times d}$, such that*

$$\boldsymbol{a}^\top \sigma(\boldsymbol{W} \boldsymbol{x}^{(i)} + \boldsymbol{b}) = y^{(i)}, \quad \forall i \in [n]$$

*and*

$$\|\boldsymbol{a}\|_2^2 + \|\boldsymbol{W}\|_F^2 + \|\boldsymbol{b}\|_2^2 \leq \mathcal{O}(n^3).$$

**Proof.** The proof of Lemma 36 is an immediate consequence of two lemmas.

1. Lemma 37 shows that the inputs $\boldsymbol{x}^{(1)}, \ldots, \boldsymbol{x}^{(n)}$ can be projected to sufficiently separated scalar values with a unit vector $\boldsymbol{v}$.

2. Lemma 38 perfectly fits $n$ univariate samples using a two-layer ReLU neural network. When invoking this lemma, we use $\|\boldsymbol{z}\|_2 = \mathcal{O}(\sqrt{n})$ and $\epsilon = \Omega(1/n^2)$ as given by Lemma 37.

The only missing piece is to upper bound $\|\boldsymbol{y}\|_2$ appearing in the final bound of Lemma 38. To that end, we apply the following Markov inequality,

$$\mathbb{P}\Big(\|\boldsymbol{y}\|_2^2 \geq 6n\Big) \leq \frac{\mathbb{E}\Big[\|\boldsymbol{y}\|_2^2\Big]}{6n} \leq \frac{1}{6}.$$

As the statement of Lemma 37 holds with probability at least $\frac{1}{3}$, this suggests that the statement of Lemma 36 holds with probability at least $\frac{1}{6}$, concluding the proof. $\qquad\square$

**Lemma 37.** *Suppose $\{\boldsymbol{x}^{(i)}\}_{i=1}^n \overset{\text{i.i.d.}}{\sim} \mathcal{N}(0, \mathbf{I}_d)$. Then, with probability at least $1/3$, there exists some $\boldsymbol{v} \in \mathbb{S}^{d-1}$ (dependent on $\{\boldsymbol{x}^{(i)}\}$) such that for all $i \neq j$,*

$$\left| \boldsymbol{v}^\top \boldsymbol{x}^{(i)} - \boldsymbol{v}^\top \boldsymbol{x}^{(j)} \right| = \Omega\left(\frac{1}{n^2}\right). \tag{B.17}$$

*and $\sum_{i=1}^n (\boldsymbol{v}^\top \boldsymbol{x}^{(i)})^2 = \mathcal{O}(n)$.*

**Proof.** The proof follows the probabilistic method. Sample $\boldsymbol{v} \sim \text{Unif}(\mathbb{S}^{d-1})$ independent of $\{\boldsymbol{x}^{(i)}\}$. For each $i \neq j$, let

$$a_{i,j} = \boldsymbol{u}^\top (\boldsymbol{x}^{(i)} - \boldsymbol{x}^{(j)})$$

and note that $a_{i,j} \mid \boldsymbol{v} \sim \mathcal{N}(0, 2)$. We apply basic Gaussian anti-concentration to place a lower bound on the probability of any $a_{i,j}$ being close to zero,

$$\mathbb{P}(\exists i, j \text{ s.t. } |a_{i,j}| \leq \epsilon) \leq \sum_{i \neq j} \mathbb{P}(|a_{i,j}| \leq \epsilon) = \sum_{i \neq j} \mathbb{E}[\mathbb{P}(|a_{i,j}| \leq \epsilon \mid \boldsymbol{v})] \leq \frac{n^2 \epsilon}{\sqrt{\pi}} \leq \frac{1}{3},$$

where the last inequality follows by taking $\epsilon = \sqrt{\pi}/(3n^2)$. Furthermore,

$$\mathbb{P}\left(\sum_{i=1}^n (\boldsymbol{v}^\top \boldsymbol{x}^{(i)})^2 \geq 3n\right) \leq \frac{\sum_{i=1}^n \mathbb{E}[(\boldsymbol{v}^\top \boldsymbol{x}^{(i)})^2]}{3n} = \frac{1}{3},$$

by Markov's inequality. Combining the two events completes the proof. $\qquad\square$

**Lemma 38.** *Consider some $\boldsymbol{z} = (z^{(1)}, \ldots, z^{(n)})^\top \in \mathbb{R}^n$ and $\boldsymbol{y} = (y^{(1)}, \ldots, y^{(n)})^\top \in \mathbb{R}^n$, such that $\left| z^{(i)} - z^{(j)} \right| \geq \epsilon$ for all $i \neq j$. For simplicity, assume $\epsilon \leq 1$. Then, there exists a two-layer ReLU neural network*

$$g(t) = \sum_{j=1}^m a_j \sigma(w_j t + b_j)$$

*that satisfies $g(z^{(i)}) = y^{(i)}$ for all $i \in [n]$, $m = n$, and*

$$\|\boldsymbol{a}\|_2^2 + \|\boldsymbol{w}\|_2^2 + \|\boldsymbol{b}\|_2^2 = \mathcal{O}\left(\frac{\|\boldsymbol{y}\|_2\sqrt{n + \|\boldsymbol{z}\|_2^2}}{\epsilon}\right). \tag{B.18}$$

**Proof.** Without loss of generality, we assume that $z^{(1)} \leq \cdots \leq z^{(n)}$. Then, we define the neural network $g$ as follows:

$$g(t) = \sum_{i=1}^{n} a_i'\sigma(w_i't - b_i') = y^{(1)}\sigma(t - z^{(1)} + 1) + \left(\frac{y^{(2)} - y^{(1)}}{z^{(2)} - z^{(1)}} - y^{(1)}\right)\sigma(t - z^{(1)})$$

$$+ \sum_{i=3}^{n}\left(\frac{y^{(i)} - y^{(i-1)}}{z^{(i)} - z^{(i-1)}} - \frac{y^{(i-1)} - y^{(i-2)}}{z^{(i-1)} - z^{(i-2)}}\right)\sigma(t - z^{(i-1)}).$$

One can verify by induction that $g(z^{(i)}) = y^{(i)}$ for every $i$ by noting that the slope of $g$ is

$$(y^{(i)} - y^{(i-1)})/(z^{(i)} - z^{(i-1)})$$

between $(z^{(i-1)}, y^{(i-1)})$ and $(z^{(i)}, y^{(i)})$. From the above, we have $w_i' = 1$, $\left\|\boldsymbol{b}'\right\|_2^2 \lesssim \|\boldsymbol{z}\|_2^2 + 1$, and $\|\boldsymbol{a}'\|_2^2 \lesssim \|\boldsymbol{y}\|_2^2/\epsilon^2$. For $\alpha = \left((\|\boldsymbol{z}\|_2^2 + n)\epsilon^2/\|\boldsymbol{y}\|_2^2\right)^{1/4}$, let $\boldsymbol{u} = \alpha\boldsymbol{u}'$, $\boldsymbol{w} = \boldsymbol{w}'/\alpha$, and $\boldsymbol{b} = \boldsymbol{b}'/\alpha$. By homogeneity, the neural network with weights $(\boldsymbol{u}, \boldsymbol{w}, \boldsymbol{b})$ has identical outputs to that of $(\boldsymbol{u}', \boldsymbol{w}', \boldsymbol{b}')$ and satisfies (B.18), completing the proof. $\square$

We are now ready to present the proof of the sample complexity lower bound for RNNs.

**Proof of Theorem 7.** First, consider the case where $d_h < n$. Note that as a function of $\boldsymbol{U}\boldsymbol{h} = (\boldsymbol{U}^{\rightarrow}\boldsymbol{h}^{\rightarrow}, \boldsymbol{U}^{\leftarrow}\boldsymbol{h}^{\leftarrow})$, $f_y$ is $\mathfrak{L}$-Lipschitz with

$$\mathfrak{L} = \left\|\boldsymbol{W}_{L_y}\right\|_{\mathrm{op}}\left\|\boldsymbol{W}_{L_y-1}\right\|_{\mathrm{op}}\cdots\left\|\boldsymbol{W}_2\right\|_{\mathrm{op}}.$$

Using the AM-GM inequality,

$$\left(\mathfrak{L}^2\|\boldsymbol{U}\|_{\mathrm{op}}^2\right)^{1/L_y} \leq \frac{1}{L_y}\|\mathrm{vec}(\boldsymbol{\Theta})\|_2^2 \leq e^{N^c/L_y}.$$

As a result, we have $\mathfrak{L}\|\boldsymbol{U}\|_{\mathrm{op}} \leq e^{N^c/2}$. By invoking Proposition 26, to obtain population risk less than some absolute constant $c_3 > 0$, we need

$$d_h \geq \Omega\left(\frac{N}{\log(1 + \mathfrak{L}^2\|\boldsymbol{U}\|_{\mathrm{op}}^2)}\right) \geq \Omega(N^{1-c}).$$

This implies $n \geq d_h \geq \Omega(N^{1-c})$. By taking $c_1$ in the theorem statement to be less than $1 - c$, we obtain a contradiction. Therefore, we must have either a population risk at least $c_3$ or $d_h \geq n$.

Suppose now that $d_h \geq n$. We show that with constant probability, we can construct an RNN that interpolates the $n$ training samples with norm independent of $n$. We simply let $\boldsymbol{\Theta}_h^{\rightarrow} = \boldsymbol{0}$, $\boldsymbol{\Theta}_h^{\leftarrow} = \boldsymbol{0}$, $\boldsymbol{U} = \boldsymbol{0}$, and describe the construction of $\boldsymbol{W}_{L_y}, \ldots, \boldsymbol{W}_2, \boldsymbol{W}_y$, and $(\boldsymbol{b}_l)$ in the following. Using the construction of Lemma 36, we can let

$$\boldsymbol{W}_y = \begin{pmatrix} \boldsymbol{W} & \boldsymbol{0}_{n\times d_E} \\ \boldsymbol{0}_{(m-n)\times d} & \boldsymbol{0}_{(m-n)\times d_E} \end{pmatrix}, \quad \boldsymbol{b}_1 = \begin{pmatrix} \boldsymbol{b} \\ \boldsymbol{0}_{m-n} \end{pmatrix}, \quad \boldsymbol{W}_2 = \begin{pmatrix} \boldsymbol{a}^\top & \boldsymbol{0}_{m-n}^\top \\ -\boldsymbol{a}^\top & \boldsymbol{0}_{m-n}^\top \\ \boldsymbol{0}_{(m-2)\times n} & \boldsymbol{0}_{(m-2)\times(m-n)} \end{pmatrix},$$

where $\boldsymbol{W} \in \mathbb{R}^{n\times d}$, and $\boldsymbol{a}, \boldsymbol{b} \in \mathbb{R}^n$ are given by Lemma 36. Then,

$$\boldsymbol{W}_2^\top\sigma(\boldsymbol{W}_y\boldsymbol{x}_{j^{(i)}}^{(i)} + \boldsymbol{b}_y) = (y_{j^{(i)}}^{(i)}, -y_{j^{(i)}}^{(i)}, 0, \ldots, 0)^\top.$$

For $(\boldsymbol{W}_l)_{l=3}^{L_y-1}$, we let $(W_l)_{11} = (W_l)_{22} = 1$, and choose the rest of the coordinates of $\boldsymbol{W}_l$ to be zero. Therefore, the output of the $l$th layer is given by

$$(\sigma(y_{j^{(i)}}^{(i)}), \sigma(-y_{j^{(i)}}^{(i)}), 0, \ldots, 0)^\top.$$

For the final layer, we let $\boldsymbol{W}_{L_y} = (1, -1, 0, \ldots, 0)$. Using the fact that $\sigma(z) - \sigma(-z) = z$, we obtain

$$f_y(\boldsymbol{U}^{\rightarrow}\boldsymbol{h}_j^{\rightarrow}, \boldsymbol{U}^{\leftarrow}\boldsymbol{h}_j^{\leftarrow}, \boldsymbol{z}_{j^{(i)}}^{(i)}; \boldsymbol{\Theta}_y) = y_{j^{(i)}}^{(i)}$$

We have found $\boldsymbol{\Theta}$ such that $\hat{R}_n^{\mathtt{RNN}}(\boldsymbol{\Theta}) = 0$ and $\|\mathrm{vec}(\boldsymbol{\Theta})\|_2^2 \leq \mathcal{O}(n^3)$ (recall that $L_y \leq \mathcal{O}(1)$). As a result, $\hat{\boldsymbol{\Theta}}_\varepsilon$ must also satisfy $\left\|\mathrm{vec}(\hat{\boldsymbol{\Theta}}_\varepsilon)\right\|_2^2 \leq \mathcal{O}(n^3)$.

On the other hand, notice that as a function of $\boldsymbol{U}\boldsymbol{h} = (\boldsymbol{U}^{\rightarrow}\boldsymbol{h}^{\rightarrow}, \boldsymbol{U}^{\leftarrow}\boldsymbol{h}^{\leftarrow})$, $f_y$ is $\mathfrak{L}$-Lipschitz with

$$\mathfrak{L} = \left\|\boldsymbol{W}_{L_y}\right\|_{\mathrm{op}}\left\|\boldsymbol{W}_{L_y-1}\right\|_{\mathrm{op}} \cdots \left\|\boldsymbol{W}_2\right\|_{\mathrm{op}}.$$

From Proposition 6, using the fact that $\|\cdot\|_{\mathrm{op}} \leq \|\cdot\|_{\mathrm{F}}$ and the AM-GM inequality, we obtain

$$\frac{1}{L_y}\|\mathrm{vec}(\boldsymbol{\Theta})\|_2^2 \geq \left(\mathfrak{L}^2\|\boldsymbol{U}\|_{\mathrm{op}}^2\right)^{1/L_y} \geq \Omega\left(\left(\frac{N}{\log d_h}\right)^{1/L_y}\right)$$

to achieve population risk less than some absolute constant $c_3 > 0$. Recall that $\log d_h \leq N^c$ for some $c < 1$. The proof is completed by noticing that unless $n \geq \Omega(N^{c_1})$ for some absolute constant $c_1 > 0$, $\left\|\mathrm{vec}(\hat{\boldsymbol{\Theta}}_\varepsilon)\right\|_2$ will always be less than the lower bound above, with some absolute constant probability $c_2 > 0$ over the training set. $\qquad\square$

## C   Auxiliary Lemmas

**Lemma 39.** *Suppose $\boldsymbol{A} \in \mathbb{R}^{d_1 \times d_2}$ and $\boldsymbol{B} \in \mathbb{R}^{d_2 \times d_3}$. Then, for all $r, s \geq 1$ and $p, q \geq 1$ such that $1/p + 1/q = 1$, we have*

$$\|\boldsymbol{A}\boldsymbol{B}\|_{r,s} \leq \|\boldsymbol{A}\|_{r,p}\|\boldsymbol{B}\|_{q,s}.$$

**Proof.** First, we note that for any vector $\boldsymbol{b} \in \mathbb{R}^{d_2}$ we have

$$\|\boldsymbol{A}\boldsymbol{b}\|_r = \left\|\sum_{j=1}^{d_2} b_j \boldsymbol{A}_{:,j}\right\|_r \leq \sum_{j=1}^{d_2} |b_j| \|\boldsymbol{A}_{:,j}\|_r \leq \|\boldsymbol{A}\|_{r,p}\|\boldsymbol{b}\|_q,$$

where the last inequality holds for all conjugate indices $p, q$ and follows from Hölder's inequality. We now have

$$\|\boldsymbol{A}\boldsymbol{B}\|_{r,s}^s = \sum_{j=1}^{d_3} \|\boldsymbol{A}\boldsymbol{B}_{:,j}\|_r^s \leq \sum_{j=1}^{d_3} \|\boldsymbol{A}\|_{r,p}^s \|\boldsymbol{B}_{:,j}\|_q^s = \|\boldsymbol{A}\|_{r,p}\|\boldsymbol{B}\|_{q,s}.$$

$\qquad\square$

The next lemma follows from standard Gaussian integration.

**Lemma 40.** *Suppose $\boldsymbol{x} \sim \mathcal{N}(\boldsymbol{\mu}, \boldsymbol{\Sigma})$. Then $\mathrm{Var}(\|\boldsymbol{x}\|^2) = 2\,\mathrm{tr}(\boldsymbol{\Sigma}^\top\boldsymbol{\Sigma}) + 4\boldsymbol{\mu}^\top\boldsymbol{\Sigma}\boldsymbol{\mu}$.*

The following lemma combines two different techniques for establishing a packing number over the unit ball, the first construction uses volume comparison, whereas the second construction uses Maurey's sparsification lemma, both of which are well-established in the literature.

**Lemma 41.** *Let $\mathcal{P}$ denote the $\epsilon$-packing number of the unit ball in $\mathbb{R}^d$. We have*

$$\log \mathcal{P} \leq \left\{d \log\left(1 + \frac{2}{\epsilon}\right)\right\} \wedge \left\{\frac{1}{\epsilon^2}(1 + \log(1 + 2d\epsilon^2))\right\}.$$

Finally, the lemma below allows us to approximate arbitrary Lipschitz functions with two-layer feedforward networks.

**Lemma 42** ([Bac17], Propositions 1 and 6)**.** *Suppose $f : \mathbb{R}^d \to \mathbb{R}$ satisfies $|f(\boldsymbol{x})| \leq LR$ and $|f(\boldsymbol{x}) - f(\boldsymbol{x}')| \leq L\|\boldsymbol{x} - \boldsymbol{x}'\|_2$ for all $\boldsymbol{x}, \boldsymbol{x}' \in \mathbb{R}^d$ with $\|\boldsymbol{x}\|_2 \leq R$ and $\|\boldsymbol{x}'\|_2 \leq R$ and some*

*constants $L, R > 0$. Then, for every $\varepsilon > 0$, there exists a positive integer $m$ and $\boldsymbol{W} \in \mathbb{R}^{m \times d}$, $\boldsymbol{b} \in \mathbb{R}^m$, and $\boldsymbol{a} \in \mathbb{R}^m$, such that*

$$\sup_{\|\boldsymbol{x}\|_2 \leq R} \left| f(\boldsymbol{x}) - \boldsymbol{a}^\top \sigma(\boldsymbol{W}\boldsymbol{x} + \boldsymbol{b}) \right| \leq \varepsilon.$$

*Additionally, we have*

$$m \leq C_d \Big( \frac{LR(1 + \log(LR/\varepsilon))}{\varepsilon} \Big)^d, \quad \Big\| \boldsymbol{W}^\top \Big\|_{2,\infty} \leq \frac{1}{R}, \quad \|\boldsymbol{b}\|_\infty \leq 1, \quad \|\boldsymbol{a}\|_2 \leq \frac{C_d LR}{\sqrt{m}} \cdot \Big( \frac{LR(1 + \log(LR/\varepsilon))}{\varepsilon} \Big)^{\frac{d+1}{2}}.$$

# D  Proof of Theorem 9

Let $\boldsymbol{u}$ be sampled uniformly from $\mathbb{S}^{d-1}$ independently from $\boldsymbol{p} = (t_1, \boldsymbol{x})$, and note that we have

$$\sup_{\boldsymbol{u} \in \mathbb{S}^{d-1}} \mathbb{E}\big[(y_j - f_{A(S_n)}(t_1, \boldsymbol{W}_{A(S_n)}\boldsymbol{x})_j)^2\big] \geq \mathbb{E}_{\boldsymbol{u} \sim \text{Unif}(\mathbb{S}^{d-1}), j, y, \boldsymbol{p} \sim \mathcal{P}}\big[(y_j - f_{A(S_n)}(t_1, \boldsymbol{W}_{A(S_n)}\boldsymbol{x})_j)^2\big],$$

for all $A \in \mathcal{A}$. From this point, we will simply use $f$ for $f_{A(S_n)}$ and $\boldsymbol{W}$ for $\boldsymbol{W}_{A(S_n)}$. Next, we argue that the output weights of any algorithm in $\mathcal{A}$ satisfy

$$\boldsymbol{w}_k = \sum_{i=1}^n \alpha_k^{(i)} \boldsymbol{x}^{(i)}, \quad \forall k \in [m_1],$$

for some coefficients $(\alpha_k^{(i)})_{i \in [n], k \in [m_1]}$. This is straightforward to verify for $A \in \mathcal{A}_{\text{SP}}$, as

$$\nabla_{\boldsymbol{w}_k} \hat{\mathcal{L}}^{\text{FFN}}(f, \boldsymbol{W}) \in \text{span}(\boldsymbol{x}^{(1)}, \ldots, \boldsymbol{x}^{(n)}).$$

For $A \in \mathcal{A}_{\text{ERM}}$, note that $\hat{\mathcal{L}}^{\text{FFN}}$ only depends on $\boldsymbol{w}_k$ through its projection on $\text{span}(\boldsymbol{x}^{(1)}, \ldots, \boldsymbol{x}^{(n)})$. As a result, any minimum-norm $\varepsilon$-ERM would satisfy $\boldsymbol{w}_k \in \text{span}(\boldsymbol{x}^{(1)}, \ldots, \boldsymbol{x}^{(n)})$.

Note that for $n \leq Nd$, the span of $\boldsymbol{x}^{(1)}, \ldots, \boldsymbol{x}^{(n)}$ is $n$-dimensional with probability 1 over $S_n$. Let $\boldsymbol{v}^{(1)}, \ldots, \boldsymbol{v}^{(n)}$ denote an orthonormal basis of $\text{span}(\boldsymbol{x}^{(1)}, \ldots, \boldsymbol{x}^{(n)})$, and let $\boldsymbol{V} = (\boldsymbol{v}^{(1)}, \ldots, \boldsymbol{v}^{(n)})^\top \in \mathbb{R}^{n \times Nd}$. Recall that for the simple-1STR model considered here, $y_j = y = \langle \boldsymbol{u}, \boldsymbol{x}_{t_q} \rangle$ for $j \in [N]$. Then,

$$\mathbb{E}_{\boldsymbol{u}, y, j, \boldsymbol{p}}\big[(y_j - f(t_1, \boldsymbol{W}\boldsymbol{x})_j)^2\big] \geq \mathbb{E}_{\boldsymbol{u}, t_1, \boldsymbol{V}\boldsymbol{x}}[\text{Var}(y \mid \boldsymbol{u}, t_1, \boldsymbol{V}\boldsymbol{x})] = \mathbb{E}_{\boldsymbol{u}, t_1, \boldsymbol{V}\boldsymbol{x}}[\text{Var}(\langle \boldsymbol{P}_{t_1}\boldsymbol{u}, \boldsymbol{x}\rangle \mid \boldsymbol{u}, t_1, \boldsymbol{V}\boldsymbol{x})],$$

where $\boldsymbol{P}_{t_1} \in \mathbb{R}^{Nd \times d}$ has the form $\big( \underbrace{\boldsymbol{0}_d, \ldots, \boldsymbol{I}_d}_{t_1}, \ldots, \boldsymbol{0}_d \big)^\top$. The conditioning above comes from the fact that via training, $f$ and $\boldsymbol{W}$ can depend on $\boldsymbol{u}$, but the prediction depends on $\boldsymbol{x}$ only through $\boldsymbol{V}\boldsymbol{x}$. Consequently, we replace the predicition of the FFN by the best predictor having access to $\boldsymbol{u}, t_1$, and $\boldsymbol{V}\boldsymbol{x}$. Note that $t_1$, $\boldsymbol{u}$, and $\boldsymbol{V}\boldsymbol{x}$ are jointly independent, and the joint distribution $(\langle \boldsymbol{P}_{t_1}\boldsymbol{u}, \boldsymbol{x}\rangle, \boldsymbol{V}\boldsymbol{x})$ is given by $\mathcal{N}\Big( 0, \begin{pmatrix} 1 & \boldsymbol{V}\boldsymbol{P}_{t_1}\boldsymbol{u} \\ \boldsymbol{u}^\top \boldsymbol{P}_{t_1}^\top \boldsymbol{V}^\top & \boldsymbol{I}_n \end{pmatrix} \Big)$, thus we have

$$\text{Var}(\langle \boldsymbol{P}_{t_1}\boldsymbol{u}, \boldsymbol{x}\rangle \mid \boldsymbol{u}, t_1, \boldsymbol{V}\boldsymbol{x}) = 1 - \|\boldsymbol{V}\boldsymbol{P}_{t_1}\boldsymbol{u}\|^2.$$

In particular,

$$\mathbb{E}_{\boldsymbol{u}}[\text{Var}(\langle \boldsymbol{P}_{t_1}\boldsymbol{u}, \boldsymbol{x}\rangle \mid \boldsymbol{u}, t_1, \boldsymbol{V}\boldsymbol{x})] = 1 - \frac{1}{d}\sum_{i=1}^n \Big\| \boldsymbol{P}_{t_1}^\top \boldsymbol{v}^{(i)} \Big\|^2,$$

and

$$\mathbb{E}_{\boldsymbol{u}, t_1}[\text{Var}(\langle \boldsymbol{P}_{t_1}\boldsymbol{u}, \boldsymbol{x}\rangle \mid \boldsymbol{u}, t_1, \boldsymbol{V}\boldsymbol{x})] = 1 - \frac{1}{Nd}\sum_{t_1=1}^N \sum_{i=1}^n \Big\| \boldsymbol{P}_{t_1}^\top \boldsymbol{v}^{(i)} \Big\|^2$$

$$= 1 - \frac{1}{Nd}\sum_{i=1}^n \Big\| \boldsymbol{v}^{(i)} \Big\|^2 = 1 - \frac{n}{Nd}.$$

$\square$

# E   Experimental Details and Additional Results

In this section, we provide the details of our experimental setup, as well as additional results on the effect of $q$ in Figure 3.

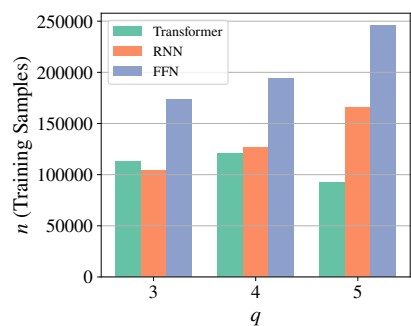

Figure 3: Number of samples required to get to test MSE loss 0.88 while training with online AdamW for the quadratic $q$STR model explained in Appendix E with $N = 7$. The gap increases with larger $q$. A closer theoretical analysis capturing the effect of large $q$ can be an interesting direction for future work.

**Architectures.** We use a Transformer composed of a multihead attention layer with $q$ heads, where each heads observes the entire $d + (q+1)d_e$-dimensional input token, followed by a fully connected ReLU layer with width 100. For the RNN, we use a simple bidirectional RNN with a hidden state size $500 \times q$, and a linear readout layer. For the FFN, we use a depth-3 fully connected ReLU network, where the first layer has width $Ndq$ and the second layer has width 1000. The output layer of the FFN has width $N$ to match the input sequence.

**Optimization.** For Figures 1 and 3 we use online AdamW with weight decay 0.1, where in Figure 1 we use a learning rate of $10^{-3}$ and in Figure 3 we use a learning rate of $10^{-4}$. Each optimization step uses an independent batch size of 64 samples, and we track the test MSE loss using an independent set of 10,000 samples. For Figure 2 we use AdamW with weight decay 0.2 and learning rate $10^{-3}$ on a fixed training set of 50,000 samples.

**Data Generating Model.** In all experiments, we sample $\boldsymbol{x} \sim \mathcal{N}(\boldsymbol{0}, \mathbf{I}_{Nd})$. For Figures 1 and 2 we have $q = 1$ and define $g(\boldsymbol{x}_1) = \langle \boldsymbol{u}, \boldsymbol{x}_1 \rangle$ for a unit-norm $\boldsymbol{u}$ uniformly sampled from the unit sphere. For Figure 3 we let $g(\boldsymbol{x}_1, \ldots, \boldsymbol{x}_q) = \frac{1}{\sqrt{q}} \sum_{i=1}^{q} \mathrm{He}_2(\langle \boldsymbol{u}_i, \boldsymbol{x}_i \rangle)$ where $\mathrm{He}_2(z) = (z^2 - 1)/\sqrt{2}$ is the normalized second Hermite polynomial. We use a non-linear $g$ as this is a more challenging setting where e.g. Transformers require $q$ heads by Theorem 4.

The code to reproduce all our experiments is provided at: https://github.com/mousavih/transformers-separation.

