# OpenReview forum: "When Do Transformers Outperform Feedforward and Recurrent Networks? A Statistical Perspective"
_NeurIPS.cc/2025/Conference — NeurIPS 2025 poster_

### Official Review · Reviewer_Dqgk · 2025-06-21

**Clarity:** 3
**Significance:** 2
**Originality:** 3
**Rating:** 5
**Confidence:** 3

**Summary:**

The paper tries to answer which function classes can Transformers learn with fewer samples compared to feedforward networks and recurrent neural networks. The dependence of sample complexity and input sequence length is shown to be critical, as sub linear dependence would imply lower error in longer sequences, and hence length generalisable solutions. The main task used to study the same is $q$-Sparse Token Regression where the output at every position depends on $q$ other input tokens, where $q<<n$. There are $2$ variants, normal where every position can have different $q$ elements which are causal to that position and a simpler variant, where every position has the same $q$ dependent tokens. With this task setup, the paper shows that a single layer Transformer with $\ge q$ heads can learn this task with sample complexity independent of the length of the sequence. RNNs can learn the simple variant in a similar fashion, but for the standard variant it requires polynomial (in length of sequence) number of samples. Feedforward networks have linear dependence (in length of sequence) to learn even simple-qSTR. Thus together these results establishes a clear separation between Transformers vis a vis RNNs and vis a vis Feedforward networks.

**Questions:**

- I am asking the following 2 questions to understand better why this specific setup was chosen. I think the current draft of the paper does not quite make it clear why q-STR is an important problem.
  - What would happen if the number of query positions that matter are also dynamic i.e. the task is not q-STR, but is $\le q$STR ?
  - q-STR is not causal, and the output at position $i$ can depend on future tokens as well. What’s the intuition on what will happen, if a causal setup is considered ?
- Regarding Assumption 2, the paragraph above (lines 132 - 134) explained the intuition behind why the assumption was required. Can you consider adding a similar paragraph, some more lines for Assumption 1 as well ? The current draft makes it hard to understand the motivation behind adding Assumption 1.

**Ethical Concerns:**

["NO or VERY MINOR ethics concerns only"]

**Final Justification:**

I read the authors responses, and understood some of the things I missed (before the clarifications given by the authors). Conditioned on the authors including some of the improvements in writing and including some of the discussions mentioned in the rebuttal, I think the paper can be quite solid and would constitute a good contribution to the field of theoretical analysis of modern neural architectures.

**Limitations:**

The limitations from the authors perspective is not discussed properly. The authors mention in the checklist that they discuss them throughout the paper and in Section 6, but I don’t think it is quite obvious what the authors consider as what the limitations are from their point of view. So, it would be helpful if it can be mentioned in a consolidated section what are the limitations for this work.

**Quality:**

4

**Strengths And Weaknesses:**

**Strengths**

- Technically quite strong. I went through all the proofs and it looks correct, no unnecessary assumptions are made to establish the results. The assumptions that are made are explained at the start, and are all reasonable.
- Quite comprehensive, establishing the results across Transformers, RNNs and FFNs, the negative results are also established for example are for a highly general class of RNNs that covers all architectures used in practise.
- Adequate empirical validation made for all the points and they match with what the theory predicts.

**Weaknesses**

- One might argue, that one doesn’t learn something as new compared to what was known before with these results. While the results seem accurate, they aren’t surprising, the fact that Transformers are better than RNNs and RNNs are better than FFNs seems intuitive for the task at hand.
- The paper is a bit difficult to read, and it took me multiple careful reads to ensure that I am not missing anything, and perhaps I still did. While the overall idea is quite clear, as mentioned in the earlier point, the overall idea was never in question, the exact specifics of how much better is one architecture better than the other which is the most interesting, and grasping that point from the paper is not that straightforward.
- The relevance of why q-STR is important, where is the general form useful compared to the simpler variant ? A discussion section is needed for explaining such things, currently it seems it has no practical relevance.

---

> ### Author Rebuttal · Authors · 2025-07-29
>
> We thank the reviewer for their thoughtful evaluation and positive feedback, and address their questions and concerns below.
>
> ---
> * **The fact that Transformers are better than RNNs and RNNs are better than FFNs seems intuitive for the task at hand**:
>
>     We would like to highlight that our contribution is exactly to mathematically formalize this separation in a quantifiable way. To our knowledge, *we are the first to theoretically prove a sample complexity separation between Transformers/RNNs/FFNs*, and we further believe that the proof techniques we develop to establish these separations can be useful in subsequent studies of other/combined architectures.
>
> ---
> * **The paper is a bit difficult to read**:
>
>     Thank you for this feedback. Table 1 and Theorem 1 summarize exactly how much Transformers are better than RNNs (on all of qSTR) and RNNs are better than FFNs (on simple-qSTR). We will try to improve the readability of the paper before the final version, and would be happy to incorporate any further specific feedback on improving readability.
>
> ---
> * **Why is qSTR relevant? Where is the general form useful compared to the simpler variant?**:
>
>     We believe this is an important question, also raised by Reviewer J2MZ, and plan to incorporate such an example to highlight the relevance of qSTR in practice.
>
>     > **Input string**: For my vacation this summer, I’m considering either Rome, `Paris`, or **Tokyo**. If I go to Rome, I would like to see their historic sites, if I go to Paris, I want to visit their `art museums`, and if I end up in Tokyo, I want to try their **cuisine**. Can you tell me how much my `second` option would cost? Would it cost more than the **third** option?
>
>     Here, the token `second` determines that the important tokens to process are `Paris` and `art museums`, and the token **third** determines that the important tokens are **Tokyo** and **cuisine**. Namely, the task involves both retrieving a few important tokens from all tokens, and processing them.
>
>     We remark that our qSTR model is of course a simplification of this setting, and it most importantly serves as a theoretical benchmark used by prior works, see e.g. [SHT23, WWHL24].
>
>     *The difference between qSTR and simple-qSTR* lies in the fact that the former is a sequence-to-sequence mapping, while the latter is a sequence-to-scalar mapping (i.e. all elements in the output sequence are identical). One way to illustrate this difference is to view qSTR as answering multiple questions about the prompt where the number of questions and thus the output of the model also grows with the input length, while viewing simple-qSTR as answering a single question, thus having an effectively constant-length output. We will better emphasize this difference in the final version.
>
> ---
> * **What would happen if the number of queries is also dynamic?**:
>
>     This is an interesting question. In this case, one would need to encode a list of indices of different lengths, which is less straightforward. Further, one would need to define how a unique function $g$ can act on sequences of different lengths. However, once these technical aspects are resolved, one might be able to show that the fundamental learning mechanism of Transformers for this problem remains unchanged.
>
> ---
> * **What happens if qSTR is made causal?**:
>
>     We note that qSTR not being causal is exactly why we use bidirectional RNNs. We expect that if one considers a causal restriction of qSTR, then Transformers with causal masks and single-directional RNNs can learn it.
>
> ---
> * **What is the intuition behind Assumption 1?**:
>
>     Assumption 1 is needed for purely technical reasons (to establish concentration bounds), and is very mild compared to typical data assumptions in the literature. It essentially means that $\Vert \boldsymbol{x}\Vert$ is subGaussian, and $y$ is sub-Weibull (even milder than sub-exponential). In practice these assumptions are usually satisfied as these quantities are typically bounded. We will further clarify this in the revised version.
>
> ---
> * **Discussion of limitations**:
>
>     Thank you for pointing this out. We will better highlight the limitations of our work in the final version. Specifically, our main limitation is the dependence of the lower bounds to ERM instead of being algorithm-free, and it would be interesting to see techniques developed to obtain minimax lower bounds for RNNs/FFNs over larger classes of algorithms. While another limitation can be the lack of studying optimization dynamics for Transformers, we believe this part should be more doable as the optimization dynamics of similar problems have been analyzed in the literature [WWHL24].
>
> ---
>
> We would be happy to answer any follow-up questions.
>
> References:
>
> [SHT23] C. Sanford et al. “Representational strengths and limitations of transformers”. NeurIPS 2023.
>
> [WWHL24] Z. Wang et al. “Transformers provably learn sparse token selection while fully-connected nets cannot.” ICML 2024.

---

> > ### Comment · Reviewer_Dqgk · 2025-08-01
> >
> > I thank the authors for their response. With the commitment to improve some of the writeup as well as including a discussion about the relevance of qSTR (both its variants), I think the paper would be quite strong. I am happy with the response, and would therefore like to increase my score.

---

### Official Review · Reviewer_Sbp1 · 2025-06-22

**Clarity:** 3
**Significance:** 3
**Originality:** 4
**Rating:** 4
**Confidence:** 3

**Summary:**

This paper introduces a novel modeling task termed q-Sparse Token Regression (qSTR) and demonstrates that Transformers exhibit a sample complexity advantage over traditional architectures (RNN, FFN) when applied to this setting.
Key Findings:
1. A single-layer Transformer can effectively leverage its attention mechanism to retrieve relevant tokens at each position, thereby adapting to dynamic sparsity. As long as the number of attention heads satisfies H ≥ q, it can successfully learn the qSTR model with sample complexity nearly independent of the sequence length N.

2. Recurrent Neural Networks (RNNs) exhibit a sample complexity lower bound of Ω(N^c) for learning general qSTR tasks, for some constant c > 0, regardless of model size.

3. Feedforward Neural Networks (FFNs), even with unbounded model size, require at least Ω(Nd) samples to learn the qSTR task, where d is the input token dimension.

**Questions:**

Could you comment on whether your theoretical results (especially the sample complexity separation) might extend to deeper Transformer architectures? Are there specific technical barriers to doing so?

**Ethical Concerns:**

["NO or VERY MINOR ethics concerns only"]

**Final Justification:**

After reading the rebuttal and other reviewers’ comments, my positive view remains unchanged. I believe the paper meets the bar for acceptance, and no issues raised alter my initial assessment.

**Limitations:**

Yes

**Paper Formatting Concerns:**

Use larger, consistent fonts in figures

**Quality:**

3

**Strengths And Weaknesses:**

Strengths
1. A novel perspective based on statistical learning theory.
2. Introduction of the qSTR task, which serves as a well-designed abstraction that captures dynamic sparsity and long-range dependencies.

Weaknesses
1. While theoretical clarity necessitates simplifications, the Transformer model studied is restricted to a single-layer architecture, which may limit the applicability of the results to practical multi-layer models.
2. The empirical experiments, though illustrative, are limited in scope and conducted on synthetic tasks. They do not evaluate performance on real-world datasets or complex modeling scenarios, leaving open questions about practical relevance.

---

> ### Author Rebuttal · Authors · 2025-07-29
>
> We thank the reviewer for their thoughtful evaluation and positive feedback, and address their questions and concerns below.
>
> ---
> * **The Transformer model studied is restricted to a single-layer architecture / How to extend to deeper Transformers**:
>
>     When considering the $q$STR model, it would be relatively straightforward to prove that a multi-layer Transformer can also learn sample-efficiently, where the subsequent layers simply learn an identity map.
>
>     We however agree that an interesting open problem is to find data models where there exists a depth separation, and one needs deep Transformers to learn sample-efficiently. There are certain tasks for which Transformer depth lower bounds are known [SHT24, CPW25]. However, these lower bound arguments are based on communication complexity and hold for finite-bit precision. It would be an interesting direction for future work to find representational lower bounds that eventually characterize sample complexity.
>
> ---
> * **The empirical experiments, though illustrative, are limited in scope and conducted on synthetic tasks**:
>
>     Indeed, our main contributions are theoretical and we present the numerical experiments only to validate our theory, and to demonstrate that our theoretical constructions for Transformers match the result of optimization algorithms, thus our ERM upper bounds should extend to gradient-based optimization.
>
>     We agree that it is important to have real-world examples where one can empirically verify Transformers outperform RNNs/FFNs on similar recovery problems. We remark however that the empirical separation between Transformers and RNNs/FFNs on this type of information retrieval tasks has already been well-studied in the literature [JBKM24]. We will better highlight these empirical separations in the revised version of our manuscript.
>
> ---
> * *Paper Formatting Concerns*:
>
>     Thank you for your suggestions.
>
> ---
>
> We would be happy to answer any follow-up questions.
>
> References:
>
> [SHT24] C. Sanford et al. “Transformers, parallel computation, and logarithmic depth.” ICML 2024.
>
> [CPW25] L. Chen et al. “Theoretical limitations of multi-layer Transformer.” FOCS 2025.
>
> [JBKM24] S. Jelassi et al. “Repeat After Me: Transformers are Better than State Space Models at Copying.” ICML 2024.

---

> > ### Comment · Reviewer_Sbp1 · 2025-08-05
> >
> > I appreciate the authors' commitment to better highlighting these empirical separations in the revised manuscript. However, after careful consideration, I have decided to keep the score unchanged.

---

### Official Review · Reviewer_iF3B · 2025-06-30

**Clarity:** 3
**Significance:** 3
**Originality:** 3
**Rating:** 5
**Confidence:** 1

**Summary:**

This paper compares Transformers, feedforward, and recurrent networks in qSTR or simple-qSTR setups, where the ability of inferring dynamic sparsity is required.
As a result of comparison and statistical analysis, it is proven that the sample complexity of a single-layer Transformer is almost independent of the context length, while recurrent networks require the samples at least dependent to the context length.
The (simple-)qSTR model family assumes a sequence-to-sequence setup, where the input is composed of the sequence of (x_i, t_i) where x_i corresponds to the value to be retrieved and t_i corresponds to the index for retrieval. The simple variant indicates the case where t_i value is fixed for every i.
In analyses, Empirical Risk Minimization (ERM) is used for finding bounds.
The results indicate that, (1) the sample complexity of Transformers depends on the context length N up to log factors, (2) RNNs cannot learn general qSTR albeit it can learn simple-qSTR, (3) FFNs require at least N samples to learn simple-qSTR.

**Questions:**

* Do you expect RNNs with attention mechanism will address some of the issues noted in the paper?
* Do you think the result of Figure 2, that the standard optimization of Transformer will match the theoretical construction, even for general multi-head cases?

**Ethical Concerns:**

["NO or VERY MINOR ethics concerns only"]

**Final Justification:**

I checked the authors’ response and other reviews, and want to keep my score.

**Limitations:**

Explicit mentioning of the limitation about the potential gap between theoretical analyses given by ERM and the empirical training done by gradient descent algorithm would be helpful for readers not knowledgeable to the field (like me).

**Paper Formatting Concerns:**

* Citation format: afaik the formatting guideline suggests using author/year or numeric formats, so I wonder the style like [SHT23] is okay.

**Quality:**

3

**Strengths And Weaknesses:**

Strengths
* Sound statistical analyses
* The organization and flow of the paper is smooth

Weaknesses
* Empirical experiments are mainly done on 1STR and simple-1STR setups
* Some factors for logical flow requires readers to refer to appendices.

---

> ### Author Rebuttal · Authors · 2025-07-29
>
> We thank the reviewer for their thoughtful evaluation and positive feedback, and address their questions and concerns below.
>
> ---
> * **Experiments are done on 1-STR**:
>
>     Indeed, we choose 1-STR as the simplest baseline and still observe the theoretical separation in practice. We believe that this separation would be even more pronounced for larger values of $q$, and plan to include such experiments in the final version of the manuscript.
>
> ---
> * **Some factors for logical flow requires readers to refer to appendices**:
>
>     Thank you for this feedback. We will try to improve the readability of our manuscript before the final version.
>
> ---
> * **Do you expect RNNs with attention mechanism will address some of the issues noted in the paper?**:
>
>     Yes, we believe an interesting future direction is to theoretically study the capabilities of hybrid architectures that can achieve some notion of optimal tradeoff by combining some global and local information.
>
> ---
> * **Will the result of Figure 2, the standard optimization of Transformer, match the theoretical construction, even for general multi-head cases?**:
>
>     This is an interesting question and we will perform additional experiments before the final version of our manuscript to see if this phenomenon holds for $q \geq 1$ where we need to use multiple heads.
>
> ---
> * **Explicit mentioning of the limitation about the potential gap between theoretical analyses given by ERM and the empirical training done by gradient descent**:
>
>     We thank the reviewer for this suggestion which will help readers better understand the current state of theoretical results on Transformers and our contributions, we will clarify this gap in the final version.
>
> ---
> * *Paper Formatting Concerns*:
>
>     Thank you for expressing your concern. We adopt the `amsalpha` citation style from `natbib` that is quite common among theoretical works in NeurIPS, specifically to be consistent with this literature.
>
> ---
> We would be happy to answer any follow-up questions.

---

### Official Review · Reviewer_J2MZ · 2025-07-01

**Clarity:** 3
**Significance:** 3
**Originality:** 3
**Rating:** 4
**Confidence:** 4

**Summary:**

This paper investigates the statistical advantages of Transformers over FFNs and RNNs from a sample complexity perspective. The authors introduce the qSTR model, where each output depends on a sparse, dynamically defined subset of the input tokens. They prove that Transformers, with enough attention heads, can learn such models with sample complexity nearly independent of the input length, unlike RNNs and FFNs. The work includes formal theorems and empirical validation, contributing to our theoretical understanding of Transformer advantages in sequence modeling.

**Questions:**

See Weaknesses.

**Ethical Concerns:**

["NO or VERY MINOR ethics concerns only"]

**Final Justification:**

The response and detailed clarifications have addressed my concerns. I will maintain my positive score.

**Limitations:**

See Weaknesses.

**Quality:**

3

**Strengths And Weaknesses:**

**Strengths.**
- The theoretical results establish a clear separation in sample complexity between transformers and RNNs/FFNs. The insights are conceptually intuitive.
- The synthetic experiments match the theoretical predictions regarding how sample complexity scales with sequence length.

**Weaknesses.**
- While the qSTR model is theoretically insightful, it may be overly simplified compared to real-world language tasks, where dependency structures are often more intricate and adaptively sparse (e.g., token-dependent).
- It remains unclear whether Transformers trained via standard optimization algorithms like Adam can effectively discover the constructive attention patterns in the theoretical analysis.
- How sensitive are the conclusions to the embedding dimension? For instance, if we increase the embedding dimension, can we reduce the number of heads required to achieve comparable sample complexity?

---

> ### Author Rebuttal · Authors · 2025-07-29
>
> We thank the reviewer for their thoughtful evaluation and positive feedback, and address their questions and concerns below.
>
> ---
> * **The qSTR model may be overly simplified compared to real-world language tasks.**
>
>     We agree with the reviewer that this model is a simplification of real-world dependencies. Our motivation to view this model was to use it as a theoretical benchmark following the works of [SHT23] and [WWHL24]. However, we will provide the following example to better demonstrate the relevance of qSTR in real-world language modeling:
>
>    > **Input string**: *For my vacation this summer, I’m considering either Rome, `Paris`, or **Tokyo**. If I go to Rome, I would like to see their historic sites, if I go to Paris, I want to visit their `art museums`, and if I end up in Tokyo, I want to try their **cuisine**. Can you tell me how much my `second` option would cost? Would it cost more than the **third** option?*
>
>     Here, the token `second` determines that the important tokens to process are `Paris` and `art museums`, and the token **third** determines that the important tokens are **Tokyo** and **cuisine**. Namely, the task involves both retrieving a few important tokens from all tokens, and processing them.
>
>     We hope this example clarifies the relevance of the qSTR model, and plan to include such an example in the final version of our paper.
>
> ---
> * **It remains unclear whether Transformers trained via standard optimization algorithms can effectively discover the constructive attention patterns in the theoretical analysis**:
>
>     Indeed, this is an important problem. We would like to highlight that we *provide some empirical evidence in Figure 2 that our attention construction is indeed learned by AdamW*. Further, [WWHL24] have studied the optimization aspect and were able to theoretically prove that gradient-based optimization recovers the same construction. However, their analysis is in the population limit, and we leave the analysis of finite-sample gradient-based optimization to future work, which we believe should be within reach given the current theoretical tools.
>
> ---
> * **How sensitive are the conclusions to the embedding dimension?**:
>
>     The lower bound on the number of heads holds for any embedding dimension, and is in fact an approximation lower bound, meaning that even with infinite embedding dimension and infinite number of samples, the Transformer still cannot learn the qSTR model unless $H \geq \Omega(q)$.
>
>     On the other hand, a larger embedding dimension may be detrimental to generalization and increase the number of training samples needed. However, depending on the way the embedding is defined, one might be able to find tailored generalization arguments that avoid a worse dependency. In particular, one should avoid using one-hot encoding for positions, as the embedding dimension would grow linearly with the sequence length which is not ideal for this setting.
>
> ---
>
> We would be happy to answer any follow-up questions.
>
> References:
>
> [SHT23] C. Sanford et al. “Representational strengths and limitations of transformers”. NeurIPS 2023.
>
> [WWHL24] Z. Wang et al. “Transformers provably learn sparse token selection while fully-connected nets cannot.” ICML 2024.

---

> > ### Comment · Reviewer_J2MZ · 2025-08-06
> >
> > Thank you to the authors for the response and detailed clarifications. I have no further questions, and I would like to congratulate you on the good work.

---

### Note · Authors · 2025-08-12

We would like to take this opportunity to thank the reviewers and the area chair for their effort in evaluating our work, and their motivating comments that helped improve our paper. We are grateful that our paper was well received by all reviewers, who appreciated the strength and breadth of our technical contributions in establishing a provable hierarchy of statistical complexity across different architectures.

Following the rebuttal discussions, in the final version we will:
* Better explain the relevance of the qSTR model to practical language-modeling scenarios.
* Use additional space to explain the broadness of Assumption 1 and improve the readability of the paper.
* Perform numerical experiments with $q > 1$ to see if a larger $q$ will further increase the gap between the studied architectures.
* Add a more explicit discussion of the limitations, especially moving towards algorithm-free lower bounds that can be an interesting future direction.

We hope our results can contribute to a more fundamental understanding of differences between popular architectures on sparse recovery tasks, and our proof techniques encourage similar principled and mathematical studies of statistical separations in modern deep learning.

---

### Decision · Program_Chairs · 2025-09-17

**Decision:**

Accept (poster)

**Comment:**

The authors introduce a theoretical model task called q-Sparse Token Regression,
and show that this task establishes a sample complexity separation between transformers and RNNs: a single layer Transformer with at least q heads can learn this task with sample complexity depending only logarithmically of the length of the sequence, whereas RNNs need polynomially many samples in the length of the sequence. A simplified version of the task leads to a separation between RNNs and MLPs, where MLPs always need linearly many samples in terms of the sequence length, whereas RNNs need only logarithmically many. Initial reviews of the paper were positive, and increased further after clarifications and additional results during the rebuttal and discussion phase. I concur with the reviewers and recommend acceptance. The authors should carefully incorporate feedback from the reviewers into the final revision.